# An assessment of the climatological representativeness of IAGOS-CARIBIC trace gas measurements using EMAC model simulations

Johannes Eckstein[1], Roland Ruhnke[1], Andreas Zahn[1], Marco Neumaier[1], Ole Kirner[2], and Peter Braesicke[1]

[1] Karlsruhe Institute of Technology (KIT), Institute of Meteorology and Climate Research (IMK), Herrmann-von-Helmholtz-Platz 1, 76344 Eggenstein-Leopoldshafen, Germany
[2] Karlsruhe Institute of Technology (KIT), Steinbuch Centre for Computing (SCC), Herrmann-von-Helmholtz-Platz 1, 76344 Eggenstein-Leopoldshafen, Germany

*Correspondence to:* Johannes Eckstein (johannes.eckstein@kit.edu)

**Abstract.** Measurement data from the long-term passenger aircraft project IAGOS-CARIBIC are often used to derive climatologies of trace gases in the upper troposphere and lower stratosphere (UTLS). We investigate to what extent such climatologies are representative of the true state of the atmosphere. Climatologies are considered relative to the tropopause in mid-latitudes ($35°$N to $75°$N) for trace gases with different atmospheric lifetimes. Using the chemistry-climate model EMAC, we sample the modelled trace gases along CARIBIC flight tracks. Representativeness is then assessed by comparing the CARIBIC sampled model data to the full climatological model state. Three statistical methods are applied for the investigation of representativeness: the Kolmogorov-Smirnov test and two scores based on (i) the variability and (ii) relative differences.

Two requirements for any score describing representativeness are essential: Representativeness is expected to increase (i) with the number of samples and (ii) with decreasing variability of the species considered. Based on these two requirements, we investigate the suitability of the different statistical measures for investigating representativeness. The Kolmogorov-Smirnov test is very strict and does not identify any trace gas climatology as representative – not even of long lived trace gases. In contrast, the two scores based on either variability or relative differences show the expected behaviour and thus appear applicable for investigating representativeness. For the final analysis of climatological representativeness, we use the relative differences score and calculate a representativeness uncertainty for each trace gas in percent.

In order to justify the transfer of conclusions about representativeness of individual trace gases from the model to measurements, we compare the trace gas variability between model and measurements. We find that the model reaches $50$-$100\%$ of the measurement variability. The tendency of the model to underestimate the variability is caused by the relatively coarse spatial and temporal model resolution.

In conclusion, we provide representativeness uncertainties for several species for tropopause referenced climatologies. Long-lived species like $CO_2$ have low uncertainties ($\leq 0.4\%$), while shorter-lived species like $O_3$ have larger uncertainties (10-15 %). Finally, we translate the representativeness score into a number of flights that are necessary to achieve a certain degree of

representativeness. For example, increasing the number of flights from 334 to 1000 would reduce the uncertainty in CO to a mere $1\%$, while the uncertainty for shorter lived species like NO would drop from $80\%$ to $10\%$.

# 1 Introduction

The UTLS (upper troposphere/lower stratosphere) is dynamically and chemically very complex and shows strong gradients in temperature, humidity and in many trace gases (Gettelman et al., 2011). As the mid and upper troposphere have a strong influence on the atmospheric greenhouse effect, the UTLS plays an important role in our climate system (Riese et al., 2012). To characterize processes and evaluate the performance of chemistry-transport models in this area, spatially well resolved data collected on a global scale are required.

Aircraft are a suitable platform to carry out these measurements as they are able to probe in situ and at a high frequency. Measurements taken by commercial aircraft projects like IAGOS (In-Service Aircraft for a Global Observing System, Petzold et al. (2015)) and CONTRAIL (Comprehensive Observation Network for Trace gases by Airliner, Matsueda et al. (2008)) generate more continuous and regular datasets than research aircraft on sporadic campaigns and are therefore commonly given the attribute representative. But what is meant by this adjective?

Ramsey and Hewitt (2005) give a general introduction to representativeness, coming from soil sciences. As they state, the adjective representative has no meaning of its own, so a definition has to be given and 'it must be asked "representative of what?"'

In the area of meteorology, Nappo et al. (1982) give the following definition: 'Representativeness is the extent to which a set of measurements taken in a space-time domain reflects the actual conditions in the same or different space-time domain taken on a scale appropriate for a specific application.' Representativeness in their understanding 'is an exact condition, i.e., an observation is or is not representative.' Only if 'a set of criteria for representativeness is established, analytical and statistical methods can be used to estimate how well the criteria are met.'

The mathematical definition given by Nappo et al. (1982) is mostly applied to data collected in the boundary layer, where it is used to answer the question whether a flux tower station is representative of the area in which it is positioned (e.g. by Schmid (1997), Laj et al. (2009) or Henne et al. (2010)). This can also be analysed by means of a cluster analysis with backward trajectories (e.g. by Henne et al. (2008) or Balzani Lööv et al. (2008)). By this method, source regions for measured trace gases can be found and the type and origin of air masses contributing to an observed air mass determined, i.e. the airmass the data are representative of. Köppe et al. (2009) apply this method to aircraft data from the project IAGOS-CARIBIC (Civil Aircraft for the Regular Investigation of the Atmosphere Based on an Instrument container, being part of IAGOS).

Lary (2004) and Stiller (2010) discuss the representativeness error in the field of data assimilation. Lary (2004) uses representativeness uncertainty as a synonym for variability within a grid cell, Stiller (2010) discusses the sampling error, which is considered to be part of the representativeness uncertainty. Larsen et al. (2014) study the representativeness of one dimensional measurements taken along the flight track of an aircraft to the three dimensional field that is being probed. But as they consider single flight tracks, their methods and definitions do not apply here.

The study of Schutgens et al. (2016) is more related to this study. They consider the sampling error on a global scale, comparing normal model means to means of model data collocated to satellite measurements. They find that this sampling error reaches $20 - 60\%$ of the model error (difference between observations and collocated model values).

We have been motivated by Kunz et al. (2008). They analysed whether the dataset of the aircraft campaign SPURT (SPURenstofftransport in der Tropopausenregion - trace gas transport in the tropopause region, Engel et al. (2006)) is representative of the larger MOZAIC dataset (Measurements of OZone, water vapour, carbon monoxide and nitrogen oxides by in-service AIrbus airCraft, the precursor of IAGOS-core). Kunz et al. (2008) investigate distributions of two substances ($O_3$ and $H_2O$) in two atmospheric compartments (upper troposphere and lower stratosphere). They find that the smaller SPURT dataset is representative on every time scale of the larger MOZAIC set for $O_3$, while this is not the case for $H_2O$. While SPURT $O_3$ data can be used for climatological investigations, the variability of $H_2O$ is too large to be fully captured by SPURT on the interseasonal time scales.

This is similar to what is done in this study: We investigate the representativeness of data for different trace gases from IAGOS-CARIBIC (see Sec. 2.1) for a climatology in the UTLS. Possible mathematical definitions of the word representativeness are first discussed with the help of this data. Then, its representativeness following these definitions is investigated. By using data from the chemistry-climate model EMAC (see Sec. 2.2) along the flight tracks of IAGOS-CARIBIC and comparing this to a larger sample taken from the model, it becomes possible to investigate the representativeness of the smaller of the two model datasets. We also assess whether the complexity of the model is similar to that portrayed by the measurements, using the variability as a measure for the complexity. We find that the variability of the model is high enough and therefore quantify the representativeness of IAGOS-CARIBIC measurement data for a climatology in the UTLS by using the two model datasets alone.

In Sec. 2, more details on the data from IAGOS-CARIBIC and the model run will be given. The general concept and definition of representativeness is discussed in Sec. 3. This section also gives details on sampling the model and on the variability, which is used to group results by species. The statistical methods are then explained in Sec. 4, namely the Kolmogorov-Smirnov test, a variability analysis following the general idea of Kunz et al. (2008) and Rohrer and Berresheim (2006) and the relative difference of two climatologies. We then discuss the variability of the model data in comparison to that of the measurements in Sec. 5. The application of the methods to the different model samples is described in Sec. 6. After showing the result of each of the three methods seperately, Sec. 6.4 discusses the representativeness of the IAGOS-CARIBIC measurement data, while Sec. 6.5 answers the question how many flights are necessary to achieve representativeness. Sec. 7 summarizes and concludes.

## 2  Model and data

### 2.1  The observational IAGOS-CARIBIC dataset

Within IAGOS-CARIBIC (CARIBIC for short), an instrumented container is mounted in the cargo bay of a Lufthansa passenger aircraft during typically four intercontinental flights per month, flying from Frankfurt, Germany (Munich, Germany, since August, 2014), see also Brenninkmeijer et al. (2007) and www.caribic-atmospheric.com.

During each CARIBIC flight, about 100 trace trace gas and aerosol parameters are measured. Some are measured continuously with a frequency between $5\,\mathrm{s}^{-1}$ and $0.2\,\mathrm{min}^{-1}$) and available from the database binned to $10\,\mathrm{s}$. Others (e.g. non-methane hydrocarbons) are taken from up to 32 air samples collected per flight. The substances considered in this study are $NO_y$, $H_2O$, $O_3$, $CO_2$, $NO$, $(CH_3)_2CO$ (acetone), $CO$ and $CH_4$ from continuous measurements and $N_2O$, $C_2H_6$ and $C_3H_8$ from air sam-

ples. $NO_y$ is the sum of all reactive nitrogen species, measured by catalytic conversion to $NO$ (Brenninkmeijer et al., 2007). Data of $N_2O$, $CH_4$ and $CO_2$ were detrended by subtracting the mean of each year from the values of that year and adding the overall mean.

The data of all flights from the year 2005 (beginning of the second phase of CARIBIC) to the end of December, 2013 (end of the model run) are considered in this study. This dataset will be referred to as MEAS$_{\mathrm{CARIBIC}}$.

As this study investigates representativeness using model data, the geolocation of the CARIBIC measurements at $10\,\mathrm{s}$ resolution is used. In a second step, the gaps of the CARIBIC measurements and height information (due to technical problems etc.) are mapped onto their representation in the model data to infer the representativeness of the measurement data.

## 2.2 The chemistry-climate model EMAC

EMAC (ECHAM5/MESSy Atmospheric Chemistry model; Jöckel et al. (2006)) is a combination of the general circulation
model ECHAM5 (Roeckner et al., 2006) and different submodels combined through the Modular Earth Submodel System (MESSy, Jöckel et al. (2005)). We use here a model configuration with 39 vertical levels reaching up to $80\,\mathrm{km}$ and a horizontal resolution of T42 (roughly $2.8°$ horizontal resolution).

The model integration used in this study simulated the time between January 1994 and December 2013, with data output every eleven hours. Meteorology is nudged up to $1\,\mathrm{hPa}$ using divergence, vorticity, ground pressure and temperature from
six-hourly ERA-Interim reanalysis. It includes the extensive EVAL-Chemistry using the kinetics for chemistry and photolysis of Sander et al. (2011). This set of equations has been designed to simulate tropospheric and stratospheric chemistry equally well.

Boundary conditions for greenhouse gases (latitude dependent monthly means) are taken from Meinshausen et al. (2011) and continued to 2013 from the RCP 6.0 scenario (Moss et al., 2010). Boundary conditions for ozone depleting substances
(CFCs and halons) are from the WMO-A1 scenario (WMO, 2010). Emissions for $NO_x$, $CO$, and non-methane volatile organic compounds are taken from the EDGAR data base (http://edgar.jrc.ec.europa.eu/index.php).

The setup of the model in this study is similar to that made for the run RC1SD-base-08 of the Earth System Chemistry integrated Modelling (ESCiMo) initiative, presented by Jöckel et al. (2016). It differs in vertical resolution (47 versus 39 levels), but horizontal resolution, nudging and the chemistry are the same. The study by Jöckel et al. (2016) gives a detailed
description and presents first validation results.

Hegglin et al. (2010) performed an extensive inter-model comparison including EMAC with the same horizontal resolution as the setup for this study. Dynamical as well as chemical metrics have been used in this study, focussing on the UTLS. Overall, they find EMAC performs well within the range of the models that were tested. The reader is referred to the study for further details.

The substances from the model used in this study are the same as those from measurements. $NO_y$, which is simulated in its components, is summed up from N, NO, $NO_2$, $NO_3$, $N_2O_5$ (counted twice because measurements of $NO_y$ are taken by catalytic conversion), $HNO_4$, $HNO_3$, HONO, HNO, PAN, $ClNO_2$, $ClNO_3$, $BrNO_2$ and $BrNO_3$. Data of $N_2O$, $CH_4$ and $CO_2$ were detrended, using the same method applied to the measurements.

## 3   Defining representativeness

As noted above and specified by Nappo et al. (1982) and Ramsey and Hewitt (2005), the word representative is meaningful only if accompanied by an object. Ramsey and Hewitt (2005) raise three questions to be answered in order to address representativeness: 1. For what parameter is the sample data to be seen as representative: e.g. the mean, a trend or an area? 2. Of which population are the sample data to be seen as representative? 3. To which degree are the data to be seen as representative? To assess the representativeness of CARIBIC data, these three questions have to be answered as well.

### 3.1   Representative for what parameter?

First, it is crucial to define what we anticipate the CARIBIC data to be representative of, since 'the same set of measurements may be deemed representative for some purpose but not other' (Nappo et al., 1982). In this study, we investigate whether the CARIBIC data can be used to construct a climatology in the UTLS. We consider monthly binned data in the height of $\pm 4.25\,\mathrm{km}$ around the dynamical tropopause defined at the pressure at $3.5\,\mathrm{PVU}$ and in mid-latitudes with $75°\mathrm{N} < \varphi < 35°\mathrm{N}$.

In order to reference data to the tropopause, we use the geometric height in kilometers relative to the tropopause (HrelTP) at each datapoint. For the measurements, this height is provided by the meteorological support of CARIBIC by KNMI (Koninklijk Nederlands Meteorologisch Instituut) (http://www.knmi.nl/samenw/campaign_support/CARIBIC/), who use data from ECMWF (European Centre for Mendium-range Weather Forecast) for their calculation.

From model output, the height relative to the tropopause (HrelTP) can be calculated, as the pressure value of the dynamical tropopause is known at each location, as well as the temperature and pressure profile. This HrelTP value calculated from the model data along the flight tracks of CARIBIC compares well with interpolated values from ECMWF provided by KNMI (Pearson correlation coefficient of $\rho = 0.97$), which is expected as the meteorology of the model is nudged using ERA-Interim data. The distribution of all values of HrelTP from the model is shown in Figure 1, showing a maximum right at the tropopause. Data were used within $\pm 4.25\,\mathrm{km}$ around the tropopause in steps of $0.5\,\mathrm{km}$.

Even though all data of trace gases (be it from model or measurements) are sorted into bins of HrelTP, it is important to keep in mind the limits in pressure. These are inherent in the CARIBIC dataset, as the aircraft flies on constant flight levels with $180\,\mathrm{hPa} < p < 280\,\mathrm{hPa}$. In addition, we explicitly limit pressure to this range in order to exclude data from ascents and descents of the aircraft. But since data are considered relative to the tropopause, these limits are no longer visible directly from the resulting climatology, even though they can influence it strongly. The reason is that aircraft flying at constant pressure can measure far above (below) the tropopause only if the tropopause is located at high (low) pressure. The properties of many trace substances are not only a function of their distance to the tropopause, but also of pressure. The limits in pressure inherent in

the sample therefore also influence the climatology. They have to be considered and should be explicitly stated. This effect is illustrated in Appendix A1 with the help of the methods developed in this study.

In addition to limiting in HrelTP and $p$, it is necessary to apply a limit in latitude $\varphi$. We limit the data by including only mid-latitudes with $75°N < \varphi < 35°N$. Tropical data with $\varphi < 35°N$ are excluded because of the considerably higher dynamical tropopause. Data with $\varphi > 75°N$ are excluded because of the different chemistry in far northern latitudes, which leads to considerably different mixing ratios for some some species that should not be combined with data from lower latitudes in one climatology. In addition, this latitudinal band is well covered by CARIBIC measurements. Other regions or latitudinal bands can be investigated using the same approach.

Like the limit in pressure, CARIBIC data are also limited in longitude, as the Pacific Ocean is never probed. The effect of this limit on the climatology is discussed in Appendix A2.

As a summary, we can specify more closely the question (Representative for what parameter?) asked in the beginning: Is a climatology compiled from CARIBIC data representative of the tropopause region in mid-latitudes?

## 3.2 Representative of which population?

When assessing the representativeness of the sample made up by all CARIBIC measurements (called $\textbf{MEAS}_{\textbf{CARIBIC}}$, see Sec. 2.1), the population is the atmosphere around the tropopause and its composition. For many of the species measured by CARIBIC, there is no other project that takes such multi-tracer in-situ meaurements as regularly at the same spatial and temporal resolution. IAGOS-core and CONTRAIL sample with much higher frequency, but take measurements of only few substances while satellites do not resolve the small scale structures necessary to disentangle the dynamics around the tropopause. The population is therefore not accessible by the measurement platforms currently available.

This is the reason why the representativeness of the CARIBIC data are investigated by comparing the model data along CARIBIC flight tracks to two larger samples taken from the model. These larger datasets are considered the population, in reference to which the representativeness of the smaller dataset (model along CARIBIC paths) is assessed. Three datasets were created from the model output: the model along CARIBIC paths and two random model samples. All are presented in the following paragraphs, a summary being given in Table 1 and Figure 1.

$\textbf{MOD}^{\textbf{regular}}_{\textbf{CARIBIC}}$: For the dataset $MOD^{regular}_{CARIBIC}$, the model output was interpolated linearly in latitude, longitude, logarithm of pressure and time to the position of the CARIBIC aircraft, using the location at a resolution of $10\,s$ for all species, independent of the time resolution in $MEAS_{CARIBIC}$. Figure 1 shows the flight paths considered in this study. Since CARIBIC also measures temperature (at $10\,s$ resolution), the high pearson correlation coefficient of $\rho = 0.97$ of modelled to measured temperature can serve as an indication that this interpolation leads to reasonable results, despite the coarser resolution in time and space of the model output.

$\textbf{MOD}^{\textbf{sampled}}_{\textbf{CARIBIC}}$: The measurement frequency for some species in $MEAS_{CARIBIC}$ is lower (e.g. those taken by whole air samples), all species contain gaps because of instrument problems at some point and some of the species considered by the model datasets are not measured at all. Sometimes, it is interesting to consider $MOD^{regular}_{CARIBIC}$ reduced to the exact number of measure-

**Table 1.** Summary of the specifications defining the three datasets $\text{MOD}_{\text{CARIBIC}}^{\text{regular}}$, $\text{MOD}_{\text{RANDPATH}}$ and $\text{MOD}_{\text{RANDLOC}}$.

| dataset | EMAC on | total sets | per month | duration | p distribution |
|---|---|---|---|---|---|
| $\text{MOD}_{\text{CARIBIC}}^{\text{regular}}$ | CARIBIC paths (2005-13) | 334 | up to 4 in 3 days | 8-10h | flight levels show up, $\bar{p} = 223.42\,\text{hPa}$ $\sigma(p) = 18.94\,\text{hPa}$ |
| $\text{MOD}_{\text{RANDPATH}}$ | random paths | 1296 | 12 in 28 days | 24h | adjusted gaussian, $\bar{p} = 223.42\,\text{hPa}$ $\sigma(p) = 18.94\,\text{hPa}$ |
| $\text{MOD}_{\text{RANDLOC}}$ | random location | 864 | 8 in 28 days | 24h | uniform, $\min(p) = 10\,\text{hPa}$ $\max(p) = 500\,\text{hPa}$ |

ment points, i.e. reduced by all these measurement gaps. The model dataset along CARIBIC paths that has the same gaps as $\text{MEAS}_{\text{CARIBIC}}$ will be referred to as $\text{MOD}_{\text{CARIBIC}}^{\text{sampled}}$.

As is visible in Figure 1 (central column), only three of the model levels lay in the pressure range sampled by CARIBIC. To have comparable statistics, $\text{MOD}_{\text{CARIBIC}}^{\text{regular}}$ was to two random model samples.

**$\text{MOD}_{\text{RANDPATH}}$**: The dataset referred to as $\text{MOD}_{\text{RANDPATH}}$ is a larger set of flight paths used to sample the model. This set was mainly used to investigate the representativeness of $\text{MOD}_{\text{CARIBIC}}^{\text{regular}}$. From the year 2005 to the end of 2013, 12 random flight paths were generated per month (1296 in total, evenly spaced in each month's first 28 days) and the model fields interpolated onto these paths. The starting point was randomly chosen in the northern hemisphere, as well as the direction taken by the aircraft. The speed was set to $885.1\,\text{km}\,\text{h}^{-1}$, the median of the speed of the true CARIBIC aircraft. The flights start at $0:00\,\text{UTC}$ and

sample the model for $24\,\text{h}$ in $10\,\text{s}$ intervals. They are reflected at the north pole and at the equator and reverse the sign of the increment in latitude direction once during flight. The first 100 of these paths are displayed in Figure 1.

The pressure was kept constant for each of the random flights, reproducing the statistics of the pressure distribution for CARIBIC as a whole. For this, a normal distribution centered around $223.42\,\text{hPa}$ with a standard deviation of $18.94\,\text{hPa}$ was used to choose the pressure value for each of the random flights. All pressure values of $p < 180\,\text{hPa}$ or $p > 280\,\text{hPa}$ were

redistributed evenly between $200\,\text{hPa}$ and $250\,\text{hPa}$ to exclude unrealistically high or low values and sharpen the maximum.

**$\text{MOD}_{\text{RANDPATH}}^{3}$**: The dependecy of representativeness on the number of flights is an important part of this study. Each of the random paths was divided into three parts, resulting in 3888 eight hour flights, the duration of a typical intercontinental flight with CARIBIC. Representativeness was then calculated with the different methods for $\text{MOD}_{\text{RANDPATH}}$ and these subsamples, increasing their size by including more of the 3888 shorter random flights. This dataset of randomized shorter flights will be

referred to as $\text{MOD}_{\text{RANDPATH}}^{3}$.

**$\text{MOD}_{\text{RANDLOC}}$**: For this sample, latitude and longitude were randomly drawn in the northern hemisphere (not aligned along a route) and the definition of the pressure distribution widened, drawing pressure from a uniform distribution from $500\,\text{hPa}$ to

10 hPa for each flight. Again, the datasets start at 0:00 UTC and the separate points are 10 s apart, collecting 8640 samples on a sampling day. Eight of these sets are distributed evenly in each month, summing to a total of 864 sets of this type. This set was used to test whether $\text{MOD}_{\text{CARIBIC}}^{\text{regular}}$ is representative of a climatology around the tropopause only within its pressure limits or also when expanding these limits.

5    As is visible in Figure 1, the distribution in HrelTP is very similar for $\text{MOD}_{\text{RANDPATH}}$ and $\text{MOD}_{\text{RANDLOC}}$ even though the pressure is prescribed in very different ways (mean of 0.79 km and 0.64 km respectively). The distribution of $\text{MOD}_{\text{CARIBIC}}^{\text{regular}}$ is different (mean of 0.26 km), which is due to the larger amount of data from southern latitudes (not shown). The different regional sampling is one of the reasons why climatologies from $\text{MOD}_{\text{CARIBIC}}^{\text{regular}}$ and $\text{MOD}_{\text{RANDPATH}}$ differ and this difference also affects the distribution in HrelTP.

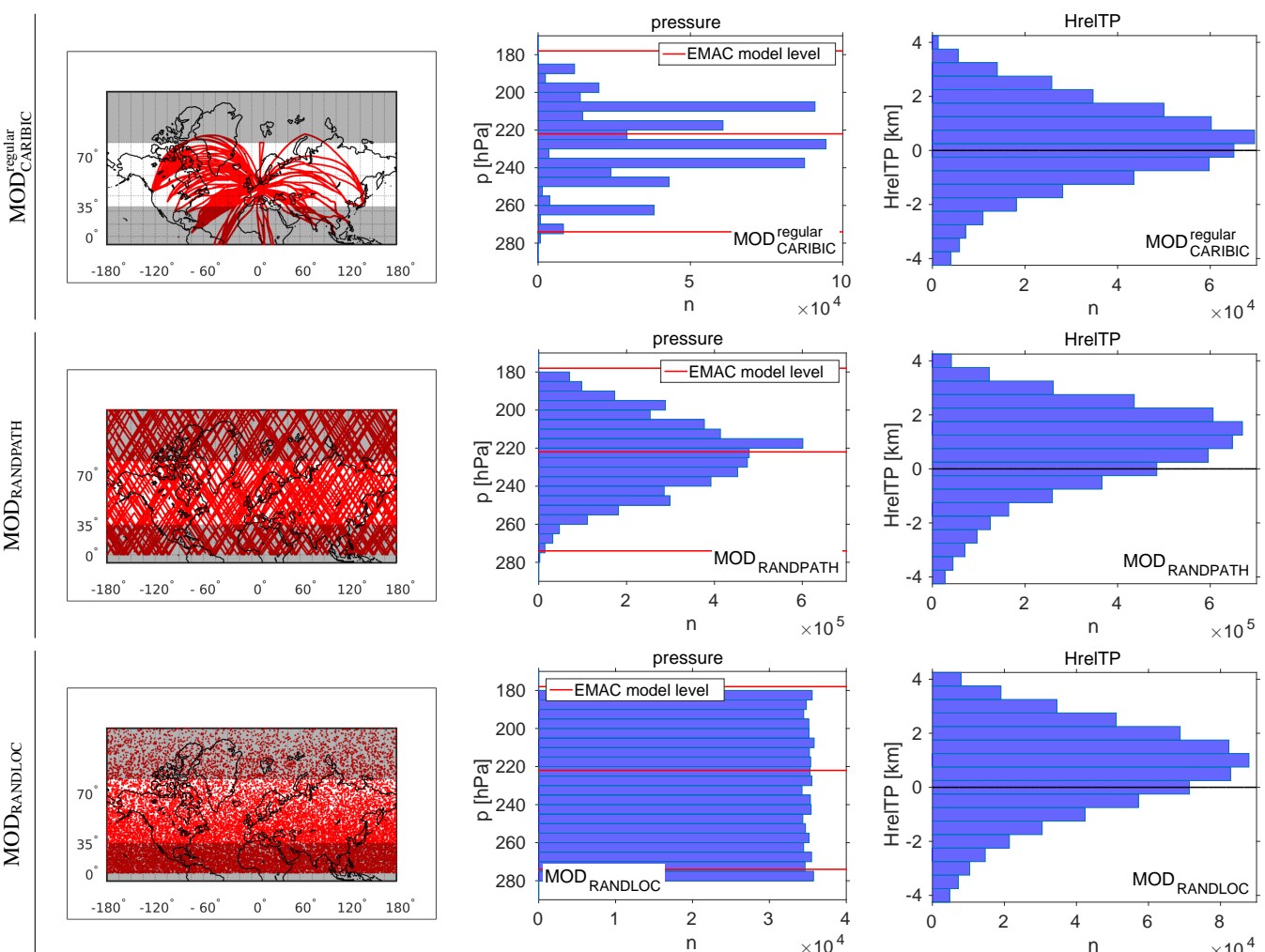

**Figure 1.** Flight path distribution (left), distribution of probed pressures ($p$, center) and height relative to the dynamical tropopause (HrelTP, right) for the three datasets $\text{MOD}_{\text{CARIBIC}}^{\text{regular}}$ (top), $\text{MOD}_{\text{RANDPATH}}$ (center) and $\text{MOD}_{\text{RANDLOC}}$ (bottom). Only parts of the paths of $\text{MOD}_{\text{RANDPATH}}$ and $\text{MOD}_{\text{RANDLOC}}$ are shown in the left column.

### 3.3 Confidence limits of representativeness

When defining representativeness, one more question remains: What are the confidence limits of the representativeness?

Three definitions for representativeness are discussed and applied in this study: The Kolmogorov-Smirnov test, the variability analysis following Kunz et al. (2008) and the relative difference of two climatologies. The first method gives a yes-no answer within a chosen statistical confidence level. The other two approaches are formulated in such a way as to return a score. By (arbitrarily) setting a value for the score, the representative cases can be discriminated from the non-representative cases (see Sec. 4 and Sec. 6), the score corresponding to a confidence level.

There are two more requirements that we define as having to be met by representativeness in general:

1. Representativeness has to increase with the number of samples (flights in the case of this study).

2. Representativeness has to decrease with increasing variability of the underlying distribution.

These two assumptions are implicitly also made by Kunz et al. (2008), as they investigate the representativeness of a smaller for a larger dataset and for two species of different variability. The measure for variability we use in this study is explained in the following section.

### 3.4 Defining a measure for variability

Representativeness is expected to differ for different species because of their atmospheric variability or atmospheric lifetime. This is part of the definition of representativeness given in Section 3.3. Kunz et al. (2008) also find that $O_3$ and $H_2O$ are different in their representativeness and attribute this to the variability. It is therefore reasonable to consider results for representativeness relative to the variability of a species. In this study, we use the relative standard deviation $\sigma_r$ as a measure for variability. It is calculated following Equation 1 using the mean $\mu$ and standard deviation $\sigma$ of each species.

$$\sigma_r = \frac{\sigma}{\mu} \tag{1}$$

Figure 2 shows the sorted values of $\sigma_r$ for the species considered in this study, using the full time series to calculate $\sigma_r$. It is worthwhile to note that in defining variability in this way, we closely follow Junge (1974), who showed that under certain constraints, the relationship

$$\sigma_r = \frac{\sigma}{\mu} = a \cdot \tau^{-b} \tag{2}$$

holds, which links variability and lifetime $\tau$ using two species-dependent constants $a$ and $b$. This relationship has frequently been called Junge relationship in the past (e.g. by Stroebe et al. (2006) or MacLeod et al. (2013)). And indeed, as visible in Figure 2, longer lived species like $CO_2$ or $N_2O$ show lower variability, while shorter lived species show higher variability.

It is important to note that the values determined from $MEAS_{CARIBIC}$ are affected by the measurement frequency in case of data sampled by whole air samples ($N_2O$, $C_2H_6$ and $C_3H_8$) and by gaps due to instrument problems. But the influence of these gaps is small, as can be seen by the small differences of the two values for $MOD_{CARIBIC}^{regular}$ and $MOD_{CARIBIC}^{sampled}$. $MEAS_{CARIBIC}$ has

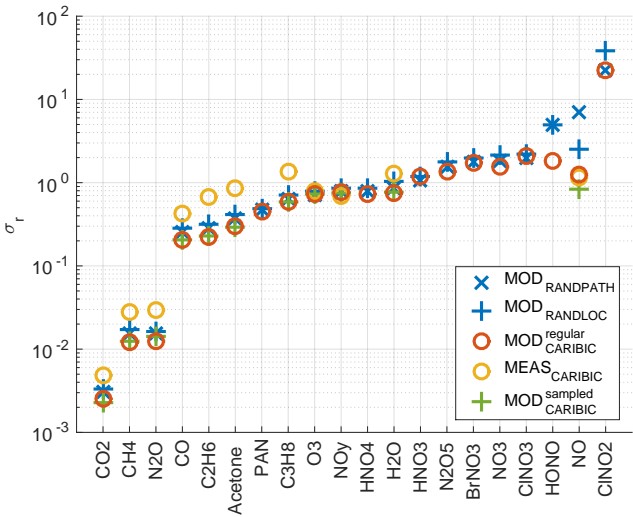

**Figure 2.** Variability $\sigma_r$ calculated for different datasets using Equation 1. The species are sorted by $\sigma_r$, species with low variability listed to the left, using the values from MOD$_{\text{RANDPATH}}$ for sorting. Note that $\log_{10}(\sigma_r) = \tau^*$, see Eq. 3.

a slightly higher variability than the model datasets for most species. The relationship of model and measurement variability is discussed in more detail in Section 5. The model datasets are very similar, despite their different sampling patterns. They only differ for short-lived species (to the right in Figure 2), which have a strong daily cycle, e.g NO.

In Sec. 3.3, we defined representativeness as having to decrease with increasing variability. Because we want to emphasize the relationship of $\sigma_r$ with $\tau$ and in order to differentiate this variability (calculated from the complete time series) clearly from other similar terms, we use $\tau^*$ defined in Equation 3 to test the relationship of representativeness and variability.

$$\tau^* = \log_{10}(\sigma_r) = \log_{10}(a) - b \cdot \log_{10}(\tau) \tag{3}$$

Sec. 4.2 will take a closer look at variability. It will be discussed how variability depends on the time scale for which it is calculated. The values shown in Figure 2 and used for the calculation of $\tau^*$ use the full time series, and thereby the overall variability. If shorter time scales had been considered, the values for $\sigma_r$ in Figure 2 would change, but not the order of the species that follows from the values.

So including these thoughts on variability in the question formulated at the end of Section 3.1, we can specify more closely the question we answer in this study: For which species is a climatology compiled from CARIBIC data representative of the tropopause region in mid-latitudes?

## 4 Statistical methods

We use three different methods to evaluate representativeness: the Kolmogorov-Smirnov test, the variability analysis and relative differences.

### 4.1 Kolmogorov-Smirnov test

The Kolmogorov-Smirnov two-sample test is a non-parametric statistical test that is used to examine whether two datasets have been taken from the same distribution (e.g. Sachs and Hedderich (2009)). It considers all types of differences in the sample distributions that can be apparent in the mean, the standard deviation, the kurtosis, etc. The test statistic is the maximum absolute difference $\hat{D}$ in the cumulative empirical distribution functions $\hat{F}_x$ of the two samples $x$:

$$\hat{D} = \max|\hat{F}_1 - \hat{F}_2| \tag{4}$$

The discriminating values $D_\alpha$ have been derived depending on the accepted confidence limit $\alpha$. In this study, the two empirical distribution functions $\hat{F}_i$ were taken from $\text{MOD}_{\text{CARIBIC}}^{\text{regular}}$ and $\text{MOD}_{\text{RANDPATH}}$ in each height bin and month. In addition to the Kolmogorov-Smirnov test, we also applied the Mann-Whitney test for the mean and Levene's and the Brown-Forsythe test for variance (see again Sachs and Hedderich (2009)). All results of applying these tests are presented in Sec. 6.1.

### 4.2 Variability analysis

The variability analysis follows Rohrer and Berresheim (2006) and Kunz et al. (2008). Rohrer and Berresheim (2006) introduced a variance analysis for ground-based observations, Kunz et al. (2008) then applied it to aircraft data. A timeseries of data is subsequently divided into ever shorter time slices of increasing number and the variance is calculated for the data within each time slice. By taking the mean over the whole number of slices and doing this for all divisions in time, a line is calculated, which is characteristic for the development of variance in time.

Instead of considering variance in each time slice, we use the relative standard deviation $\sigma_r = \frac{\sigma}{\mu}$, which is the definition of variability following Junge (1974). It is calculated in each time slice and the mean gives the value for the corresponding time scale. In the following, time scale therefore refers to the length of the interval in time in which the variability is calculated. By scaling the standard deviation $\sigma$ with the mean $\mu$, different species become comparable. Being a combination of variability as defined by Junge (1974) and the variance analysis introduced by Rohrer and Berresheim (2006), this method is called variability analysis in the following paragraphs.

Figure 3 shows the variability analysis for CO just below the tropopause for $\text{MOD}_{\text{CARIBIC}}^{\text{regular}}$, $\text{MOD}_{\text{RANDPATH}}$ and $\text{MOD}_{\text{RANDLOC}}$. The time scale changes from about $5\,\text{min}$ to $5\,\text{a}$ along the logarithmically spaced abscissa. As CO is a medium long-lived trace gas with an atmospheric lifetime of 2-3 months and a pronounced annual cycle, the mean variability increases up to time scales of $1\,\text{a}$. The variability of $\text{MOD}_{\text{RANDPATH}}$ and $\text{MOD}_{\text{RANDLOC}}$ is larger than that of $\text{MOD}_{\text{CARIBIC}}^{\text{regular}}$ on almost all time scales. For time scales of $30\,\text{d}$ and more, however, the lines of all three datasets run in parallel, showing an increase up to $1\,\text{a}$, from when on the variability does not increase. This is consistent with the annual cycle of CO, which is also the cause for the relative

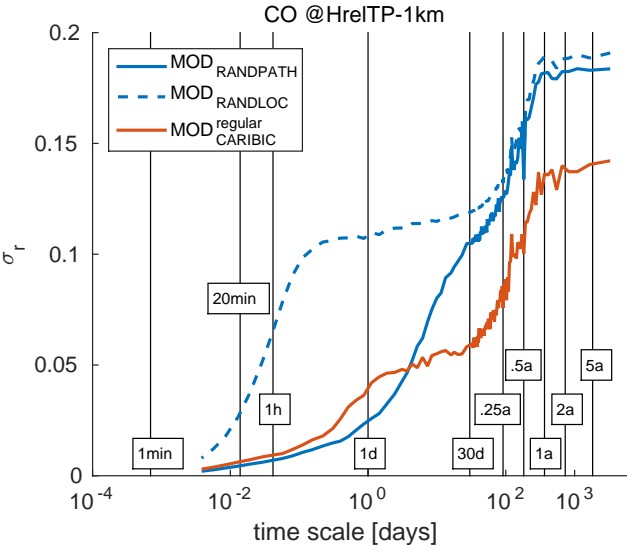

**Figure 3.** Variability analysis calculated for CO for MOD$_{\text{RANDPATH}}$, MOD$_{\text{RANDLOC}}$ and MOD$_{\text{CARIBIC}}^{\text{regular}}$ at HrelTP $= -1\,\text{km}$ (one kilometer below the tropopause). The time scales used to calculate R$_{\text{var}}$ using Equation 5 are indicated by vertical lines.

decrease sharply at $0.5\,\text{a}$ and $1.5\,\text{a}$. For time scales below $30\,\text{d}$, the distribution of flights in one month dominates the variability analysis. MOD$_{\text{CARIBIC}}^{\text{regular}}$ includes only up to four flights on consecutive days, the mean variability does not decrease when going to time scales between $30\,\text{d}$ and $4\,\text{d}$, while in MOD$_{\text{RANDPATH}}$, continuosly less data are included in each time slice, leading to a continuous drop in the variability. For time scales of less than $1\,\text{d}$, the data come from a single flight, showing another

drop in variability that is linked to using data from geographic regions that are ever more close in the case of MOD$_{\text{CARIBIC}}^{\text{regular}}$ and MOD$_{\text{RANDPATH}}$. Since the variability analysis is so closely linked to the distribution in time and space, the variability analysis of MOD$_{\text{RANDLOC}}$ shows an almost constant value for time scales shorter than $30\,\text{d}$ until time scales shorter than one day are reached, from when on the variability also drops.

Kunz et al. (2008) used the variance analysis to investigate whether the smaller SPURT dataset represents the variance

present in MOZAIC dataset. Following this thinking, we consider the variability as one possible criterion to judge the representativeness of one dataset for another. A score R$_{\text{var}}^{t,h}$ describing the representativeness is defined from the difference of the values of the variability analysis, using the following equation:

$$\text{R}_{\text{var}}^{t,h} = \log_{10}\left(\left|\overline{\left[\frac{\sigma_1^{t,h}}{\mu_1^{t,h}}\right]} - \overline{\left[\frac{\sigma_2^{t,h}}{\mu_2^{t,h}}\right]}\right|\right) \tag{5}$$

where $\sigma_x^{t,h}$ stands for the standard deviation and at $\mu_x^{t,h}$ for the mean in time scale $t$ and height $h$ of the datasets $x$. The overbar

implies that the mean over all time slices corresponding to the time scale $t$ of $\sigma/\mu$ are used. Considering Figure 3, the score can be interpreted as the absolute value of the difference of the two lines at certain time scales $t$.

Decreasing values of $R_{var}^{t,h}$ mean better representativeness, the value always being negative. Depending on $t$, the representativeness in different time scales can be evaluated. We used time scales of $30\,d$, $0.25\,a$, $0.5\,a$, $1\,a$, $2\,a$ and $5\,a$ to calculate $R_{var}^{t,h}$. When applying this method to all height bins, a profile in $R_{var}^t$ is calculated for each species. This is one possible definition for representativeness. Yet it has to pass the two requirements of being related to number of samples and variability outlined in
Sec. 3.3. The results of testing this will be presented in Sec. 6.2.

### 4.3   Relative differences

The third approach to assess representativeness is to analyze the relative differences between the climatologies from two differently large datasets. The procedure is summarized in Equation 6:

$$R_{rel}^h = \log_{10}\left(\frac{1}{12}\sum_{m=1}^{12}\frac{|\mu_1^{m,h} - \mu_2^{m,h}|}{\mu_2^{m,h}}\right) \tag{6}$$

which was applied to each height bin $h$. $\mu_x^{m,h}$ stands for the mean of the data in the month $m$ and in height bin $h$ of the datasets $x$. The logarithm to the basis 10 was applied to the mean relative difference profile to end up with a profile in $R_{rel}$, similar to the score $R_{var}^t$ calculated from the variability analysis. Contrary to the Kolmogorov-Smirnov test or the variability analysis, this test statistic does not contain any information on the underlying distribution, because it uses only the mean in each bin.

Figure 4 shows an example of relative differences between CO from $MOD_{CARIBIC}^{regular}$ and the larger dataset $MOD_{RANDPATH}$.
The differences are small, mostly below an absolute value of 0.15. $R_{rel}$ is defined (in Equation 6) as the logarithm to the base 10 of the mean over all months (not shown). The score increases towards the top and bottom in Figure 4 due to less data there. Like for $R_{var}^t$, decreasing values in $R_{rel}$ mean better representativeness. And like $R_{var}^t$, $R_{rel}$ has to be tested for passing the requirements of being related to number of samples and variability (see Sec. 3.3) in order to be acceptable as a score for representativeness. The results of testing this will be discussed in Sec. 6.3.

Other than just as a score, the value of $R_{rel}$ can be understood as the average uncertainty for assuming the climatology of $MOD_{CARIBIC}^{regular}$ as a full model climatology. This is more obvious if taken to the power of 10, in which case the uncertainty will take values between 0 and 1. Use of this will be made in Section 6.4.

## 5   Model and measurement variability

Representativeness was assessed using only model data in this study, yet the final goal was to investigate the representativeness
of $MEAS_{CARIBIC}$. $MOD_{CARIBIC}^{regular}$ and $MOD_{CARIBIC}^{sampled}$ are used as a placeholder for $MEAS_{CARIBIC}$ and compared to other model datasets ($MOD_{RANDPATH}$ and $MOD_{RANDLOC}$) in the analysis. The results derived from these model datasets will be interpreted for $MEAS_{CARIBIC}$ in Sec. 6. This means that conclusions drawn from model data alone will be applied to measurements.

To justify this reasoning, it is important to investigate the differences between the model and the real atmosphere. It is not crucial that the model reproduces the exact values of the measurements, but rather that the complexity for each species in the
model is similar to the real complexity. This will be investigated in the following two sections. The variability of $MOD_{CARIBIC}^{sampled}$ will be used as an indicator of its complexity and compared to the variability of $MEAS_{CARIBIC}$. Similar to Equation 1, we use

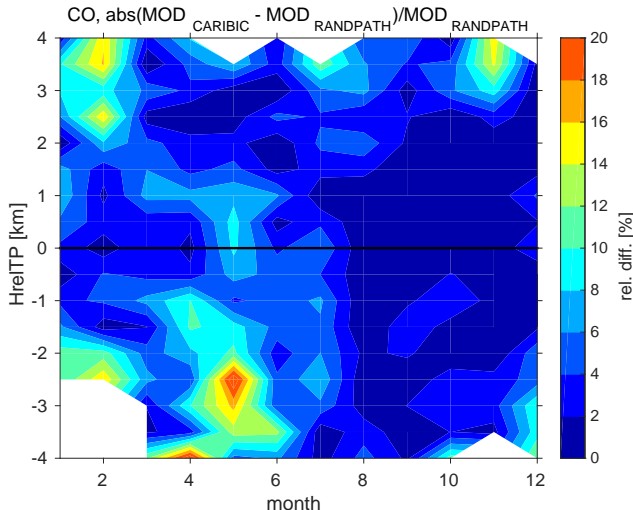

**Figure 4.** Relative differences of CO for $MOD_{CARIBIC}^{regular}$ and $MOD_{RANDPATH}$. This is the basis used to calculate $R_{rel}$.

the relative standard deviation $\sigma_r = \sigma/\mu$ as a measure for variability when comparing model and measurements. Variability of a certain time scale, e.g. $20\,min$, will be referred to as $20\,min$ variability in the following, accordingly for other time scales.

### 5.1 Influence of short time scales on the climatological mean

All model datasets have been created from gridded datafiles with a certain resolution ($2.8°$ or about $200\,km$, see Sec. 2.2).

Considering the median airspeed of the CARIBIC aircraft of $885.1\,km\,h^{-1}$, this model resolution corresponds to a time scale of about $20\,min$. $MEAS_{CARIBIC}$ has a time resoltution of up to $10\,s$, depending on the instrument. Model data has been linearly interpolated to this high $10\,s$ resolution, but this does not introduce the variability that is present in the measurements. The $20\,min$ variability is therefore always larger in $MEAS_{CARIBIC}$ than in $MOD_{CARIBIC}^{sampled}$. To what extent this small scale variability influences the climatological values is investigated here.

By reducing the $20\,min$ variability in $MEAS_{CARIBIC}$ to that of $MOD_{CARIBIC}^{sampled}$, it is possible to determine the influence of the small scale variability on the climatological mean values. The reduction in variability was done separately for each species and height to account for differences in terms of model complexity between the species. In order to reduce the variability in the time series, they were smoothed out, the method is presented in App. B. The smoothing number used in this method indicates how much variability has been removed. The $20\,min$ variability of $MEAS_{CARIBIC}$ was then calculated for several smoothing

numbers.

Figure 5 (left panel, solid lines) shows how the $20\,min$ variability drops for all species if the data are smoothed progressively (increasing the smoothing number). The leftmost point for each species corresponds to the full $20\,min$ variability, while this variability drops to zero if the time intervals considered in smoothing become much longer than $20\,min$. The dashed lines show

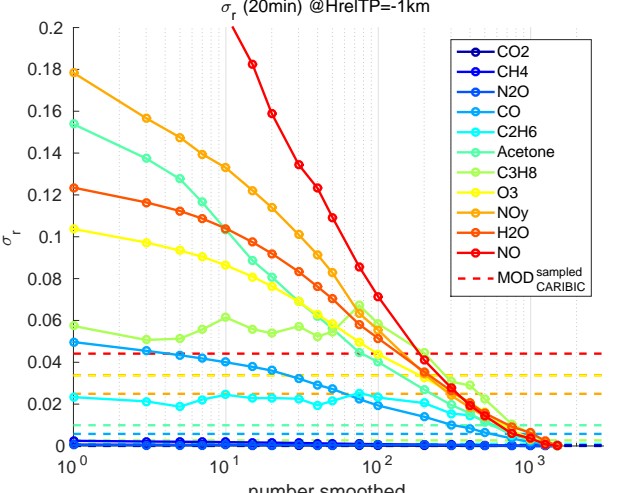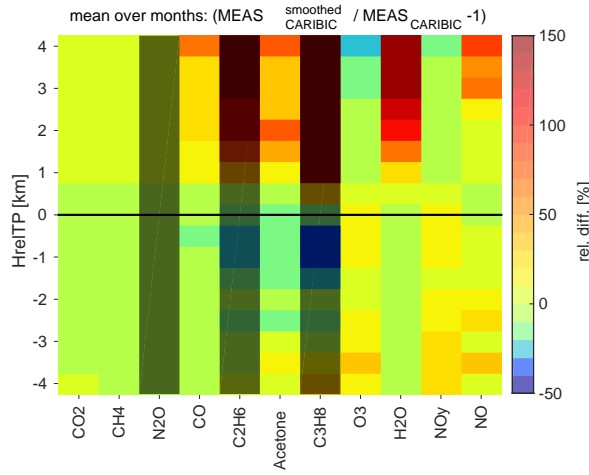

**Figure 5.** Left panel: $20\,\text{min}$ variability of i) $\text{MEAS}_{\text{CARIBIC}}$, that has been smoothed out to an increasing degree, indicated by an increasing smoothing number (solid lines) and of ii) $\text{MOD}_{\text{CARIBIC}}^{\text{sampled}}$ (dashed lines), both for $\text{HrelTP} = -1\,\text{km}$. The crosspoint of the dashed and corresponding full line indicate the smoothing number that is needed to reproduce the $20\,\text{min}$ variability of $\text{MOD}_{\text{CARIBIC}}^{\text{sampled}}$. Right panel: Mean relative differences of $\text{MEAS}_{\text{CARIBIC}}^{\text{smoothed}}$ and $\text{MEAS}_{\text{CARIBIC}}$. $\text{MEAS}_{\text{CARIBIC}}^{\text{smoothed}}$ has been smoothed to have the same $20\,\text{min}$ variability as $\text{MOD}_{\text{CARIBIC}}^{\text{sampled}}$, using the smoothing number from the left hand panel. The relative differences correspond to the error in the climatologies of $\text{MOD}_{\text{CARIBIC}}^{\text{sampled}}$ due to the coarse model resolution. $N_2O$, $C_2H_6$ and $C_3H_8$ are measured by air samples with a low measurement frequency and therefore not considered here.

the full model variability, which was not smoothed out. The crosspoints of the full and corresponding dashed line indicate the smoothing numbers for which $\text{MEAS}_{\text{CARIBIC}}$ has the same $20\,\text{min}$ variability as $\text{MOD}_{\text{CARIBIC}}^{\text{sampled}}$. $\text{MEAS}_{\text{CARIBIC}}$ in which each species has been smoothed to this point will be referred to as $\text{MEAS}_{\text{CARIBIC}}^{\text{smoothed}}$.

Climatological mean values of $\text{MEAS}_{\text{CARIBIC}}^{\text{smoothed}}$ were then compared to mean values from $\text{MEAS}_{\text{CARIBIC}}$ with the full variabil-
ity, thereby determining the influence of the reduced $20\,\text{min}$ variability. A similar influence is expected by the coarse model resolution, which by definition has the same $20\,\text{min}$ variability as $\text{MEAS}_{\text{CARIBIC}}^{\text{smoothed}}$.

The mean relative difference of the climatologies for different species between $\text{MEAS}_{\text{CARIBIC}}^{\text{smoothed}}$ and $\text{MEAS}_{\text{CARIBIC}}$ is displayed in Figure 5 (right panel). The differences depend strongly on the species. Those species that are measured by air samples ($N_2O$, $C_2H_6$ and $C_3H_8$) have been shaded in grey, since they contain very little data far above and below the tropopause and are
therefore not considered in this section.

The mean relative differences are smaller than $1\%$ for the long lived species to the left and reach $10\text{-}20\%$ for the other species. Largest values appear where the mixing ratios of the species are small and vertical gradients are strong, i.e. in stratospheric CO, acetone or $H_2O$ and tropospheric $O_3$. E.g. $H_2O$ has very low stratospheric mixing ratios, that are reached in small-scale intrusions of stratospheric air encountered during flight. If these small-scale structures are smoothed out, the mean
values become larger and the difference of $\text{MEAS}_{\text{CARIBIC}}^{\text{smoothed}}$ and $\text{MEAS}_{\text{CARIBIC}}$ is large and positive.

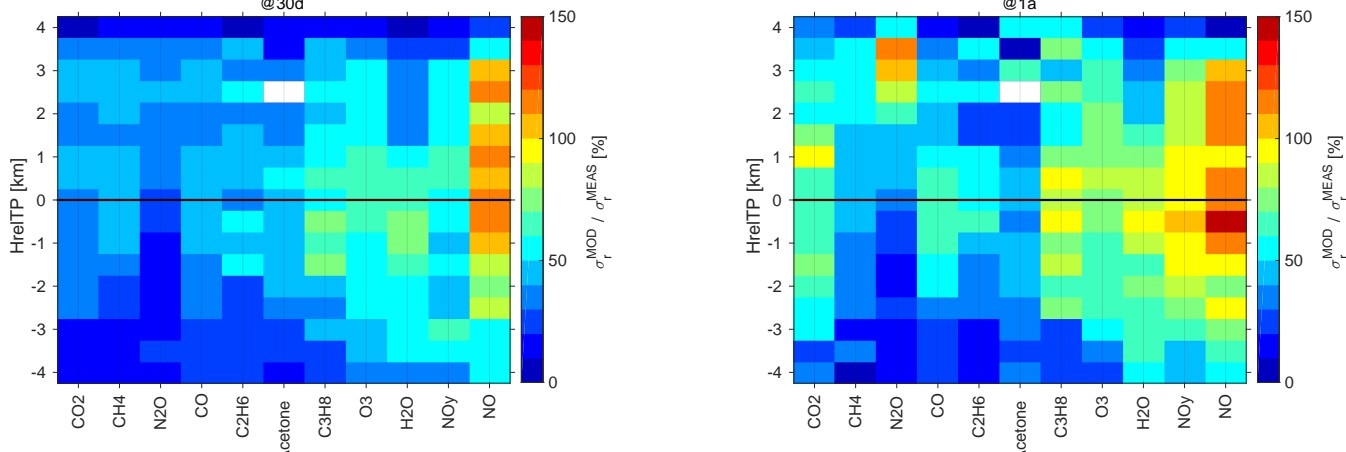

**Figure 6.** $\sigma_r^{\mathrm{MOD}}/\sigma_r^{\mathrm{MEAS}}$ given in percent for time scales of $30\,\mathrm{d}$ (left) and $1\,\mathrm{a}$ (right), where MOD stands for $\mathrm{MOD}_{\mathrm{CARIBIC}}^{\mathrm{sampled}}$ and MEAS stands for $\mathrm{MEAS}_{\mathrm{CARIBIC}}^{\mathrm{smoothed}}$. Values greater than $50\,\%$ indicate the high model complexity.

The relative differences show the influence of a lower variability that is equal to that of $\mathrm{MOD}_{\mathrm{CARIBIC}}^{\mathrm{sampled}}$. This therefore shows that the coarse model resolution does in principle not lead to very large errors in climatological mean values. Nevertheless, the model could have other defiencies in the description of the different species. These are made visible in the following section by comparing model and measurement variability directly.

## 5.2  Comparing model and measurement variability

In this section, the variability of $\mathrm{MOD}_{\mathrm{CARIBIC}}^{\mathrm{sampled}}$ is compared directly to that of $\mathrm{MEAS}_{\mathrm{CARIBIC}}^{\mathrm{smoothed}}$. For this dataset, $\mathrm{MEAS}_{\mathrm{CARIBIC}}$ has been altered in such a way to reproduce the $20\,\mathrm{min}$ variability of $\mathrm{MOD}_{\mathrm{CARIBIC}}^{\mathrm{sampled}}$, see the preceeding section. As this study argues completely within the model world, it is important that the model has similar values for the variability, which is used as an indicator of the underlying complexity. If the model cannot reproduce the measurement variability at all, it is not plausible why conclusions on representativeness drawn from model data should also be true for the real atmosphere.

As has been discussed in Sec. 4.2, variability depends on the time scale for which it is considered. In order to evaluate the model performance, we compare $\sigma_r$ on time scales of $30\,\mathrm{d}$ and $1\,\mathrm{a}$. $30\,\mathrm{d}$ variability includes data from typically 4 flights, so this is a measure for the atmospheric variabilty on the global, large scale dynamics. $1\,\mathrm{a}$ variability gives a good impression of the annual cycle, as it includes data from many flights and different years. Figure 6 shows $\sigma_r^{\mathrm{MOD}}/\sigma_r^{\mathrm{MEAS}}$ for time scales of $30\,\mathrm{d}$ (left) and $1\,\mathrm{a}$ (right), using the datasets $\mathrm{MOD}_{\mathrm{CARIBIC}}^{\mathrm{sampled}}$ and $\mathrm{MEAS}_{\mathrm{CARIBIC}}^{\mathrm{smoothed}}$

Figure 6 shows that the variability in the measurements reached by the model differs between species. In general, the variability reached for shorter lived species better fits that of the measurements. Short-lived species also undergo a more complex chemistry in the model, which adds variability. The $30\,\mathrm{d}$ variability shown in Figure 6 (left) reveals to what extent the model is able to capture variability related to the large scale dynamics. Most species reach 40-80 %. NO is very short lived and strongly determined by its daily cycle, which is the reason why the variability in the model reaches higher values.

The time scale of $1\,\mathrm{a}$ shows the variability that represents seasonality. The model does a better job for this time scale than for $30\,\mathrm{d}$, short lived species and $CO_2$ reaching well over $60\,\%$ of the variability, approaching $100\,\%$ for some species. Here again, the model chemistry increases the variability for shorter lived species to the right. There are species that are not as well represented, while this also depends on the height considered (e.g. high values for stratospheric $N_2O$).

The model variability is influenced by many factors including the dynamics, the representation of the chemistry and of the sources included in the model. The limited horizontal and vertical resolution also plays a role, even though $MEAS_{CARIBIC}^{smoothed}$ is used as a reference for the comparison. If compared to the original $MEAS_{CARIBIC}$, the percentages of variability reached by the model drop by $10$-$20\,\%$ (not shown). It is beyond the scope of this paper to further disentangle what causes the deficiencies of the model and what leads to the differences between the species.

As is shown in Figure 6, the model reaches more than $50\,\%$ of the variability of the measurements, depending on the species and time scale. In general, the model variability can be increased by using a run with a higher resolution, because a decrease in spatial resolution requires a decrease in the time step of the integration. The variability of the measurements in each bin of HrelTP (height relative to the tropopause) is influenced by the choice of reference for HrelTP. For this study, HrelTP has been derived from model output fields from ECMWF at a resolution of $1°$ ($\approx 110\,\mathrm{km}$), while the measurement data have a much

higher resolution ($\approx 2.5\,\mathrm{km}$, see Sec. 2.1). The highly variable measurements are then sorted into bins of coarsely resolved HrelTP, artificially increasing the variability of the measurements in each bin of HrelTP. This also affects $MEAS_{CARIBIC}^{smoothed}$. Considering these complementing thoughts on the model and measurement variability, the fraction of variability reached by the model (more than $50\,\%$) justifies the application of the representativeness evaluated from the model to $MEAS_{CARIBIC}$.

## 6    Results

Here, we first present the results of the application of the Kolmogorov-Smirnov test (Sec. 6.1), the variability analysis (Sec. 6.2) and the relative difference (Sec. 6.3) to $MOD_{CARIBIC}^{regular}$ and $MOD_{RANDPATH}$. All have to be related to the number of flights and the variability of the species as discussed in Section 3.3. These methods have also been applied to data not from an atmospheric model but from a random number generator, leading to equivalent results. These are presented as supplementary material to the article. Sec. 6.4 interprets the results by species as a representativeness uncertainty. Finally, Sec. 6.5 answers the question

of how many flights are necessary to achieve a certain degree of representativeness. In addition, Appendix A discusses the influence of the limitations in longitude and in pressure which are inherent in the CARIBIC dataset.

### 6.1    Applying the Kolmogorov-Smirnov test

The application of the Kolmogorov-Smirnov test to $MOD_{CARIBIC}^{regular}$ and $MOD_{RANDPATH}$ yields a first important result. Independent of the trace gas and height considered, the result is always negative (not shown). This means that the data in each bin of

$MOD_{CARIBIC}^{regular}$ are not representative of the corresponding bin in $MOD_{RANDPATH}$ when defining representativeness by a positive result of the Kolmogorov-Smirnov test. This is also true if the data are not binned in months but only in HrelTP. The result also stays the same for all values of the confidence limit $\alpha$ (using values of 0.001, 0.01, 0.05, 0.1 and 0.2).

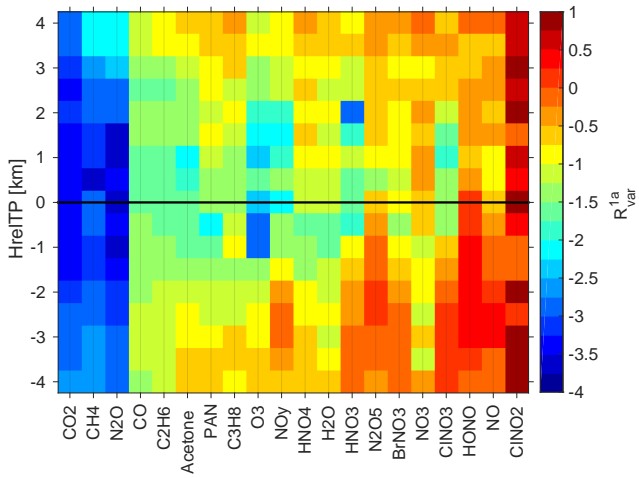

**Figure 7.** $R_{var}$ calculated according to Equation 5 for a time scale of $1\,a$ for all species in all height bins, using $MOD_{CARIBIC}^{regular}$ and $MOD_{RANDPATH}$. Low values indicate small differences in variability.

A similar finding for aircraft data have already been reported by Kunz et al. (2008). On the one hand side this could mean that $MOD_{CARIBIC}^{regular}$ is simply not representative of $MOD_{RANDPATH}$. But if the other methods presented here are considered, the conclusion seems more appropriate that the Kolmogorov-Smirnov test is simply not the appropriate way to answer the question. It can be considered as too strict for the type of data and the question considered here. This is also the result of a sensitivity
study, which is discussed as supplementary material to this text.

In addition to binning into twelve months (January to December), we have also tested $MOD_{CARIBIC}^{regular}$ and $MOD_{RANDPATH}$ when first binning into separate months (108 months in nine years) and then using this monthly mean data to compile a climatology. For this monthly mean data, the Kolmogorov-Smirnov test does give a positive result in some heights and months. But no meaningful pattern could be determined from the results. Especially, the result does not depend on $\tau^*$ (not shown). The same
is true for the Mann-Whitney test for the mean and Levene's and the Brown-Forsythe test for variance. They give no positive result for data binned directly into months. The result is positive for some months and heights if data are first binned into separate months the monthly mean data used for testing. The postive results seem randomly distributed and no relationship to $\tau^*$ could be found. These tests therefore also seem not to be suitable for answering the question of representativeness.

### 6.2 Applying the variability analysis

This section presents the results of the application of the variability analysis to $MOD_{CARIBIC}^{regular}$ and $MOD_{RANDPATH}$. Equation 5 was applied for different time scales ($30\,d$, $0.25\,a$, $0.5\,a$, $1\,a$, $2\,a$ and $5\,a$) to calculate $R_{var}$. The results are exemplarily discussed for a time scale of $1\,a$, shown in Figure 7, in which the results are sorted using the values of $\tau^*$ displayed in Figure 2.

$R_{var}$ shows a strong dependancy on $\tau^*$. This is visible from Figure 7, in which the results are sorted with decreasing values of $\tau^*$ (from Figure 2), i.e. with increasingly higher atmospheric variabilty from left to right. The Pearson correlation coefficient $\rho$ of $R_{var}$ and $\tau^*$ is high, $|\rho| > 0.9$ in all height bins, independent of the time scale. $R_{var}$ also shows a strong relationship to the number of samples: The amount of data in both $MOD_{CARIBIC}^{regular}$ and $MOD_{RANDPATH}$ decreases below and above the tropopause, and $R_{var}$ follows suit for practically all species.

The relation of $R_{var}$ and the number of flights was also tested by using $MOD_{RANDPATH}^3$ defined in Sec. 3.3. $R_{var}$ was correlated with the number of flights for each species and height. When investigating a linear relationship, the Pearson correlation coefficient was approximately $|\rho| \approx 0.75$ for the time scale of $5\,a$, increasing continously when considering shorter time scales to $|\rho| \approx 0.95$ for the time scale of $30\,d$. Considering a logarithmic relationship inreases the goodness of fit for longer time scales, while it decreases that for shorter time scales ($|\rho| \approx 0.85$ for both $5\,a$ and $30\,d$).

$R_{var}$ therefore passes the requirements of being inversely related to $\tau^*$ and directly to the number of included data points and flights. Figure 7 can therefore be used to judge upon the representativeness of $MOD_{CARIBIC}^{regular}$ for $MOD_{RANDPATH}$.

This shows that by using the relative standard deviation (Equation 5) instead of the variance analysis applied by Kunz et al. (2008), the difference in variability can be used to infer representativeness. Rohrer and Berresheim (2006) originally introduced the variance analysis to investigate the sources and time scales of variability in a dataset and for this it remains a valid method. In order to infer representativeness, it is more appropriate to use the relative standard deviation in the analysis instead of the absolute variance.

## 6.3 Relative differences

$R_{rel}$ was calculated for each species in each height bin according to Equation 6, results are presented in Figure 8.

Figure 8 shows how low variability (decreasing to the left, values taken from Figure 2), is linked with good representativeness (low values in $R_{rel}$). $R_{rel}$ decreases linearly with increasing variability $\tau^*$ with a high Pearson correlation coefficient greater than 0.95 for all height bins (not shown). As visibile in Figure 8, $R_{rel}$ also decreases with the number of data points, which maximizes just around the tropopause and decreases above and below it (see Figure 1).

This dependance on the number of data points was also tested by using $MOD_{RANDPATH}^3$ described in Sec. 3.3. The Pearson correlation coefficient $\rho$ between the number of shorter random flights and $R_{rel}$ was $\rho \approx 0.95$ for all species in all heights. Less variable species like $CO_2$ show a better relationship with the logarithm of the number of flights. This underlines how $R_{rel}$ is well correlated with the number of measurements.

Using $R_{rel}$ as a measure passes both conditions: It is directly proportional to the number of flights and indirectly to the variability. In addition to Figure 7, Figure 8 can therefore be used to judge upon the representativeness of $MOD_{CARIBIC}^{regular}$ for $MOD_{RANDPATH}$. $R_{rel}$ can be transformed into a relative difference in percent, by taking $R_{rel}$ to the power of ten. A score of -2 stands for a mean relative difference of $1\,\%$.

The score that discriminates representative from the non-representative case has to be arbitrarily chosen (see Nappo et al. (1982) and Ramsey and Hewitt (2005)). This score gives the uncertainty within which the data are considered representative. If a score of -2 is defined as representative (corresponding to $1\,\%$ mean relative difference), then representative species and

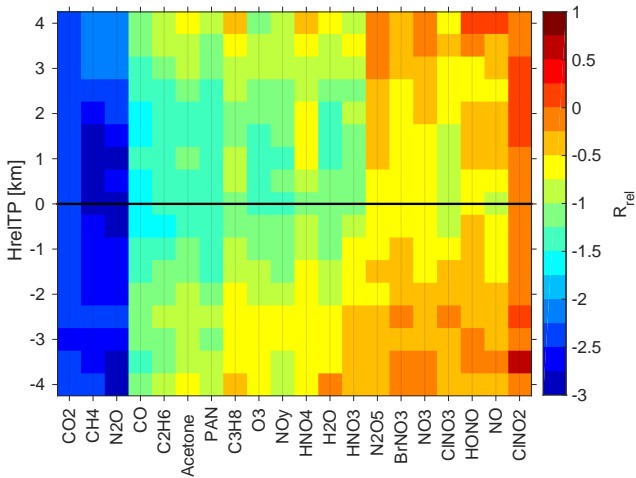

**Figure 8.** $R_{rel}$ calculated according to Equation 6 for all species in all height bins, using $MOD_{CARIBIC}^{regular}$ and $MOD_{RANDPATH}$. Low values indicate small differences in climatological mean values.

heights can now be seperated from those species that are not representative using the results from Figure 8. But the score of -2 is arbitrary. If it is reduced to -1.5 (roughly $3\%$ relative difference), $MOD_{CARIBIC}^{regular}$ can be seen as representative for many more species.

## 6.4 Representativeness uncertainty of the CARIBIC measurement data

The last sections have shown $R_{rel}$ (see Equation 6) and $R_{var}$ (see Equation 5) to be adequate scores to describe representativeness. After reconsidering the question we asked in the Section 3.1 (Is a climatology compiled from CARIBIC data representative of the tropopause region in mid-latitudes?), we will use $R_{rel}$ in the following. It is more intuitive (compared to $R_{var}$) as it describes the difference to a larger dataset, e.g. in percent. A further discussion of $R_{var}$ is beyond the scope of this paper. As noted in Sec. 4.3, $R_{rel}$ is also comprehensible as an uncertainty for using the smaller dataset to compile a climatology and will

be called representativeness uncertainty correspondingly.

In order to asses the uncertainty for accepting CARIBIC measurement data to create a climatology, model data have to contain the same amount of data as $MEAS_{CARIBIC}$, which is why $MOD_{CARIBIC}^{sampled}$ (see Sec. 2) will be used in the following. In addition, $MOD_{RANDLOC}$ (see Table 1) was used as reference, as it has a random sampling pattern and represents the full model state, independent of the sampling pressure. The limits in pressure where again set to $180\,hPa < p < 280\,hPa$. The resulting

$R_{rel}$ is shown in Figure 9. Using different wording, $R_{rel}$ in this formulation can also be considered the sampling error of the measurements.

This result - deduced from model data only - is also valid for the real world if the complexity of the model is sufficiently high for each species. This has been shown by comparing the variability of $MOD_{CARIBIC}^{sampled}$ and $MEAS_{CARIBIC}^{smoothed}$ for different time

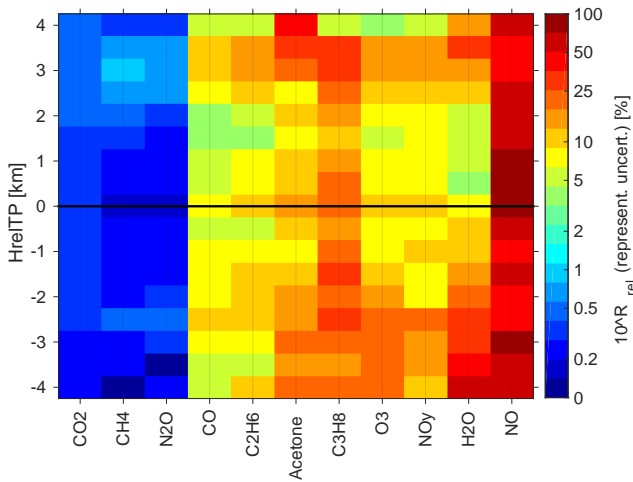

**Figure 9.** Representativeness uncertainty for using the CARIBIC data (that is 334 long-distance flights, see Table 1) to compile a climatology: $10^{R_{rel}}$ calculated from $MOD_{RANDLOC}$ and $MOD_{CARIBIC}^{sampled}$. Low values indicate small representativeness uncertainties. $N_2O$, $C_2H_6$ and $C_3H_8$ are measured from air samples, which increases the uncertainty, especially for $C_3H_8$.

scales (see Sec. 5). The discussion of the following paragraphs is therefore also valid for the real atmosphere, even though results have been derived from model data alone. Figure 9 answers the question we asked in Sec. 3.2: For which species is a climatology compiled from CARIBIC data representative of the tropopause region in mid-latitudes?

When considering the representativeness uncertainty of a climatology, it is also important to consider the annual cycle of a species, e.g. $10\%$ can be much for a species that is more or less constant, while it is much for a species with a strong seasonality. The following paragraphs discuss representativeness by species, not explicitly considering the seasonal variations for each species. The monthly resolved climatologies of $CO$, $CO_2$ and $O_3$ will be discussed exemplarily at the end of this section.

Many of the species that sum up to $NO_y$ in the model are not actually measured by CARIBIC and therefore are not displayed in Figure 9. In general, the representativeness uncertainty is lowest where there are most measurements, which is just around the tropopause (see Figure 1). This effect overlays the physical reasons for the different uncertainties for the considered species.

$NO$ has the highest uncertainty of $90\%$. We propose two possible reasons: On the one hand, there are many gaps in the observations. But $NO$ is also emitted by aircraft in the UTLS (Stevenson et al., 2004), and since CARIBIC flies in the flight corridors heavily frequented by commercial aircraft, it is unrealistic to assume a climatology of these species to be representative of the UTLS on a whole.

$H_2O$ shows a strong gradient in its representativeness uncertainty, which is directly linked to the strong gradient in variability. The dry stratosphere can be described by relatively few measurements, which is why the uncertainty is low, only reaching $25\%$

at most. The humid and variable troposphere influenced by daily meteorology has a higher uncertainty, reaching more than $60\%$.

NO$_y$, being a pseudo-species made up of many substances, is more difficult to disassemble. The variabilty of many components is higher in the troposphere, where the uncertainty is $30\%$ at its maximum. Above, it is smaller than $10\%$ and the climatology therefore quite trustworthy.

It is interesting to note that $C_2H_6$ and $C_3H_8$, both collected in whole air samples still reach uncertainties comparable to those of other species in their range of $\tau^*$. This is due to the fact that these are moderately long-lived species for which only a smaller number of measurements are needed for a representative climatology. The climatology of $C_3H_8$ comes with an uncertainty of up to $25\%$, while that of $C_2H_6$ is better with an uncertainty of less than $10\%$.

The climatology of $O_3$ is very trustworthy, the uncertainty being smaller than $10\%$ for most height bins. The higher values in the tropospheric bins should not raise much concern, as $O_3$ increases strongly with height in the UTLS and an uncertainty of $15\%$ will be practically unnoticable compared to the vertical increase.

This is not true for acetone, where the gradient is just opposite to $O_3$. The climatology is trustable with an uncertainty only up to $10\%$ in upper levels, while it increases to $20\%$ in the lower heights, where the influence of spatially and temporally variable sources at the ground is stronger.

The climatology of CO is very good, the uncertainty in stratospheric height bins being less than $5\%$. The troposphere, again stronger under the influence of sources, has a higher uncertainty reaching up to $10\%$.

The long-lived trace gases $CH_4$, $N_2O$ and $CO_2$ (all detrended as described in Sec. 2.1) all have representativeness uncertainties of less than $0.4\%$, which is lower than their seasonal variability. This is interesting especially for $N_2O$, which is measured only in the whole air samples.

As example and summary, the representativeness uncertainty will be applied to climatologies of CO, $CO_2$ and $O_3$, shown in Figure 10. CO is shown for $MOD_{CARIBIC}^{sampled}$ (top left, panel A), $MOD_{RANDLOC}$ (top right, panel B) and CARIBIC measurements ($MEAS_{CARIBIC}$, center left, panel C). The white space in these figures has three possible reasons: the aircraft could have never flown in that bin, there could be measurement gaps in CO or a gap in HrelTP. The measurement gaps of CO and HrelTP from $MEAS_{CARIBIC}$ have been mapped onto $MOD_{CARIBIC}^{sampled}$, but HrelTP differs slightly and therefore also the white space. The representation of CO in the model, comparing top and center left figure (panels A and C), is similar to measurements (in the troposphere more so than in the stratosphere), but was not subject of this study. We compared the top row ($MOD_{CARIBIC}^{sampled}$ and $MOD_{RANDLOC}$, panels A and B) and found that $R_{rel}$ is a good descriptor for the representativeness of one for the other. By accepting the result from the model to be valid also for measurements, we can now use the score calculated from the two model samples to determine the representativeness uncertainty of $MEAS_{CARIBIC}$.

By again defining $R_{rel} = -1$ ($10\%$ uncertainty, one third of the seasonal variation) as the limit for representativeness, the climatology of $MEAS_{CARIBIC}$ (Figure 10, center left, panel C) was shaded in grey where it is not representative. The representativeness uncertainty shown in Figure 9 only serves as a first indication of the expected uncertainty when resolving monthwise. The center right panel (panel D) displays the standard deviation of CO from $MOD_{RANDLOC}$. By comparing the center panels (C and D), it becomes evident that the variability specific to CO is one of the reasons for the higher representativeness uncer-

tainty in spring, while it cannot explain all the features. The number of flights is a different reason, which explains the higher uncertainty in January, the month with the least flights (not shown).

The limit of $10\%$ should not be applied in general and has to be adapted to the species under consideration. This becomes evident by the bottom row in Figure 10 (panels E and F), which shows climatologies of $CO_2$ and O3. $CO_2$ shows a small annual variation around a high background value. So $10\%$ uncertainty could be easily reached by a single measurement, which would certainly not be representative of the whole year. The shading for $CO_2$ in Figure 10 was set at a threshold of $0.3\%$, again just above one third of the seasonal variation. The high values in spring in the upper troposphere show an even lower uncertainty, the uncertainty of all data being less than $0.7\%$ (not shown). The opposite is true for O3, for which the threshold was set to $15\%$ uncertainty (around one fourth of the seasonal variation). Many tropospheric values in spring or at times of high gradients in the stratosphere at the beginning and end of spring have an uncertainty higher than these $15\%$.

As the results in Figure 9 are sorted by the variability of the species and this is linked to their lifetime in following Junge (1974), conclusions are possible for species even if they have not been explicitly considered in this study. This is true for $SF_6$, for example, which is measured in whole air samples by CARIBIC but was set to 0 in the model run and could therefore not be included in this study. As it is long-lived in both troposphere and stratosphere (Ravishankara et al., 1993), a climatology from CARIBIC $SF_6$ measurements can be considered to be representative even though it is measured only by whole air samples.

Two limitations are inherent in the CARIBIC data: the Pacific Ocean is never sampled and the pressure is limited to flight levels. The influence of both these limitations is discussed in Appendix A.

## 6.5 Number of flights for representativeness

One last question remains to be answered: For those substances not representative yet, how often does one have to fly in order to achieve a representative climatology?

This question can be answered with the help of $MOD^3_{RANDPATH}$. Figure 11 shows the representativeness uncertainty for some species and different numbers of flights. As has been discussed in Section 6.4, the yearly variation of a species is one of the factors that determines the threshold of the uncertainty with which the species can be considered to be representative.

E.g., for (detrended) $CO_2$, the mean value of $MOD_{RANDLOC}$ is $385.7\,\mathrm{ppmv}$ with a yearly variation of 2.5 to $3.5\,\mathrm{ppmv}$. A representativeness uncertainty of at least $0.5\%$ has therefore to be set as the minimum threshold for $CO_2$. This can be reached with only few flights, much less than those included in $MOD^{sampled}_{CARIBIC}$, indicated by the dashed line in Figure 11 at 334 flights.

For $O_3$, on the other hand, the yearly cycle proposes an uncertainty of $50\%$ or more. While this is the minimum value to reproduce the yearly cycle at all, it may still not be sufficient for the application. With the number of CARIBIC flights, the uncertainty in $O_3$ is low already ($< 5\%$ in this height), while the uncertainty is continuosly reduced if the number of flights increases.

As is indicated by Figure 11, highly variable species like NO need many flights in order for their climatologies to reach low uncertainties. Even 1000 flights, approximately ten more years of flying the CARIBIC observatory, will not reduce the uncertainty below $10\%$.

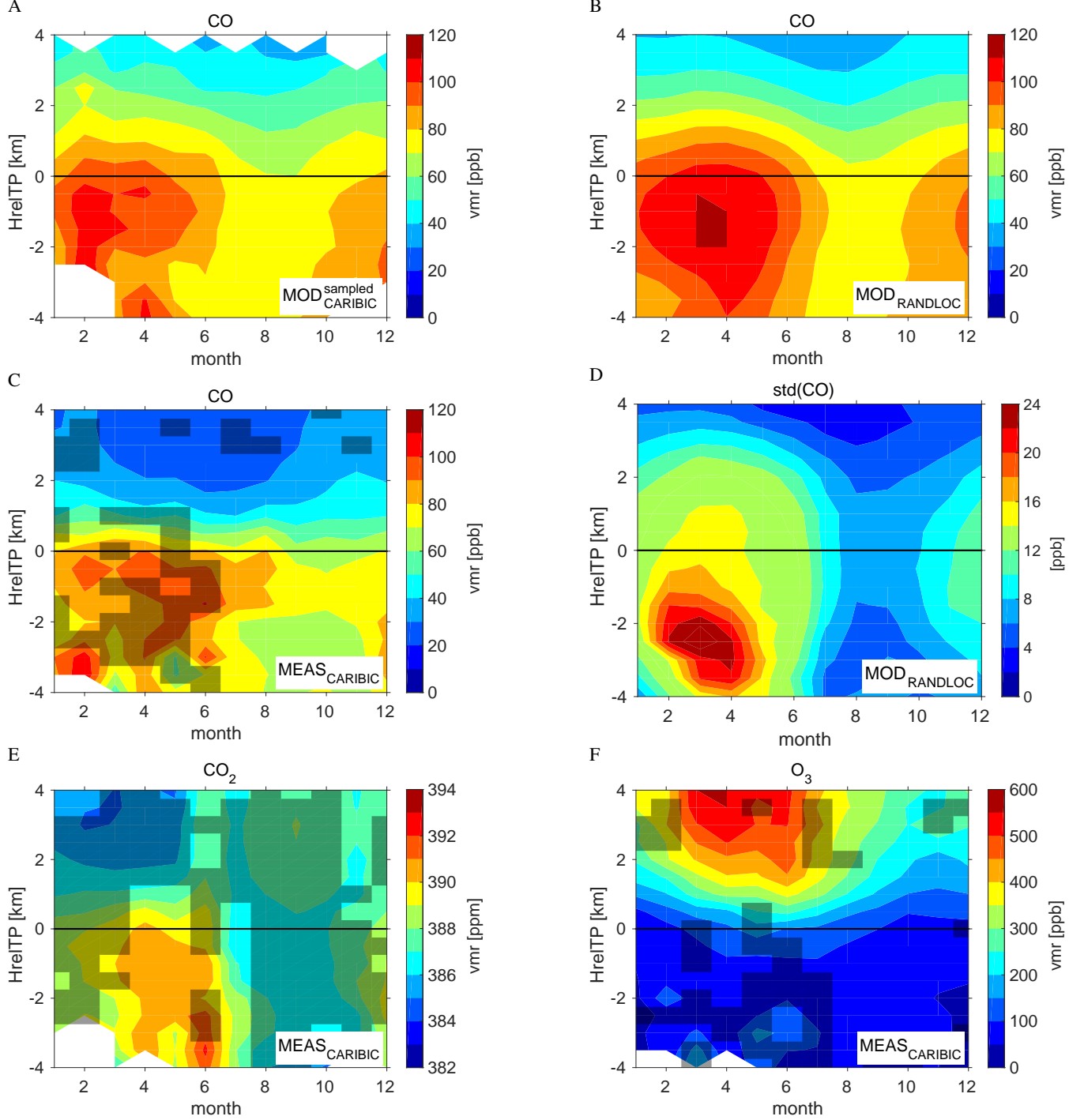

**Figure 10.** Climatology of CO, built from $\text{MOD}^{\text{sampled}}_{\text{CARIBIC}}$ (panel A), $\text{MOD}_{\text{RANDLOC}}$ (panel B) and the CARIBIC measurements ($\text{MEAS}_{\text{CARIBIC}}$, panel C). Areas of $10\hat{\ }R_{\text{rel}} > 0.1$, calculated from the top row, were used to shade non-representative areas in the climatology of $\text{MEAS}_{\text{CARIBIC}}$ in grey. Panel D displays the $1\sigma$ standard deviation of CO from $\text{MOD}_{\text{RANDLOC}}$. The bottom row (panels E and F) displays climatologies from $\text{MEAS}_{\text{CARIBIC}}$ of $CO_2$ (left) and O3, shaded with $10\hat{\ }R_{\text{rel}} > 0.003$ and $10\hat{\ }R_{\text{rel}} > 0.15$, respectively.

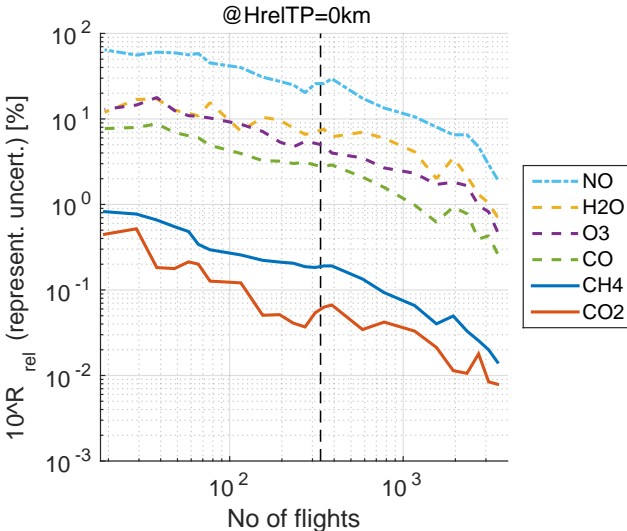

**Figure 11.** Representativeness uncertainty for different numbers of flights for some species. The number of flights in MEAS$_{\text{CARIBIC}}$ is indicated by the vertical dashed line. Other species can be deduced from their value of $\tau^*$ with the help of Figure 2.

Other species that are not included in Figure 11 can be deduced from their value of $\tau^*$ with the help of Figure 2. Those species measured in air samples need even more CARIBIC flights than indicated by the number in Figure **??**, as the measurement frequency is much lower.

## 7  Conclusions

We describe and assess the degree of climatological representativeness of data from the passenger aircraft project IAGOS-CARIBIC. After a general discussion of the concept of representativeness, we apply general rules to investigate whether climatologies from IAGOS-CARIBIC trace gas measurements can be seen as representative. We answer the specific question: For which species is a climatology compiled from CARIBIC data representative of the tropopause region in mid-latitudes?

In order to answer this question, four datasets were created from a nudged model run of the chemistry-climate model

EMAC. Two datasets sample the model at the geolocation of CARIBIC measurement data (MOD$_{\text{CARIBIC}}^{\text{regular}}$ and MOD$_{\text{CARIBIC}}^{\text{sampled}}$). These datasets are contrasted to the much larger datasets MOD$_{\text{RANDPATH}}$ (random flight tracks with similar properties as those of MOD$_{\text{CARIBIC}}^{\text{regular}}$) and MOD$_{\text{RANDLOC}}$ (random locations).

As a first step, we demonstrate that these model datasets are appropriate to answer our question, which asks for the representativeness of CARIBIC measurement data. In order to justify the validity of the conclusions drawn from model data to

the measurements, we compare model and measurement variability, using the variability as an indication of the models ability to reproduce changes in space and time. To compare like with like, variability on scales smaller than the model resolution is

removed from the measurements. With this prerequisite the model reproduces 50-100% of the variability of the measurements, depending on time scale, height relative to the tropopause and species. This is sufficient to transfer our results from the model world to the real atmosphere considering the coarse resolution of the model and of the data used for binning the measurements into height relative to the tropopause.

Three methods to describe representativeness are developed and applied: (i) the Kolmogorov-Smirnov test (and the Mann-Whitney, Brown-Forsythe and Levene's test), (ii) variability analysis following Kunz et al. (2008) and (iii) a test interpreting the relative difference between two datasets. Two fundamental requirements are essential for representativeness: its increase (i) with the number of measurements and (ii) with decreasing atmospheric variability of the species, which is related to atmospheric lifetime following Junge (1974). By formulating the variability analysis and relative differences as scores ($R_{var}$ and $R_{rel}$

respectively), we demonstrate that they pass these two requirements, while the statistical tests are all too strict. $R_{rel}$ (describing the representativeness of a climatology) is better suited for answering the question and is therefore used in the remaining analysis.

The score $R_{rel}$ is easily converted to a representativeness uncertainty in percent and this measure is used in the discussion. The results show that $CO_2$, $N_2O$ and $CH_4$ have very low uncertainties (below 0.4%). CO, $C_2H_6$, and $O_3$ reach higher values

(5% - 20%), but can still be used to compile representative climatologies around the tropopause. $NO_y$ and $H_2O$ are only usable in the lower stratosphere (uncertainties of 5% to 8% there, higher elsewhere), while NO and $C_3H_8$ cannot be used for a representative climatology (uncertainties of 25% and more). Naturally, the interpretation of results strongly depends on the chosen threshold uncertainty and should depend on the seasonal variability of the species under consideration. This is demonstrated by setting different limits for climatologies of $CO_2$, CO and $O_3$.

In addition, the uncertainty can be translated into a number of flights necessary to achieve representativeness. This is demonstrated for some species by showing the relationship of the number of flights and the representativeness uncertainty. For long-lived species like $CO_2$ and $CH_4$, the 334 IAGOS-CARIBIC flights used in this study already provide enough data, while short-lived species like NO need around 1000 flights to reduce the uncertainty to 10%, sufficient to reproduce the strong annual cycle.

The general concept of using two sets of model data to calculate the representativeness is easily applicable to other questions. One model dataset should mirror the measurements, the other should be much larger, taking into account certain statistical properties of the measurement dataset, so that the two datasets become comparable.

Questioning the representativeness of sampled data is important. Patterns might occur when sorting or averaging sparsely sampled data, but these patterns are not necessarily meaningful. We discuss and show a way to address this problem of repre-

sentativeness by using model data. By help of the methods presented here, representativeness is given a sound mathematical description, returning an uncertainty characterizing the specific dataset.

## Appendix A:  Limitations in longitude and pressure

$MEAS_{CARIBIC}$ is limited in longitude (the Pacific Ocean is never sampled) and pressure (as all civil aircraft, CARIBIC flies at a certain pressure level). Both limitations influence the climatologies calculated from the dataset. They are discussed in the following sections.

### A1    Limitation in pressure: Aircraft tropopause pressure bias

By calculating $R_{rel}$ using $MOD_{CARIBIC}^{regular}$ and $MOD_{RANDLOC}$, an important fact can be illustrated about data collected with instruments on civil aircraft. As the aircraft flies at constant pressure levels, data are also taken at these pressure altitudes only. If data are then resorted into heights relative to the tropopause (HrelTP), this limit in pressure is no longer visible. Nevertheless, it influences the results as the volume mixing rations of many trace substances are not only a function of their distance to the tropopause, but also of pressure.

The effect on the climatological values can be illustrated by calculating $R_{rel}$ (see Equation 4) using $MOD_{RANDLOC}$ and $MOD_{CARIBIC}^{regular}$ within $10\,hPa < p < 500\,hPa$. Figure 12 shows the results (right panel). For comparison, the left panel of Figure 12 shows $R_{rel}$ of the same datasets when setting $180\,hPa < p < 280\,hPa$, the range at which CARIBIC measures. The representativeness uncertainty is much higher in almost all heights on the right hand side ($10\,hPa < p < 500\,hPa$), except just above the tropopause, where $MOD_{CARIBIC}^{regular}$ contains most data. Only the long lived species $CO_2$, $N_2O$ and $CH_4$ retain their low uncertainties. For the more variable species to the right of the figure, the representativeness uncertainty increases strongly, especially in the troposphere, where the variability increases if data taken at higher pressure are included.

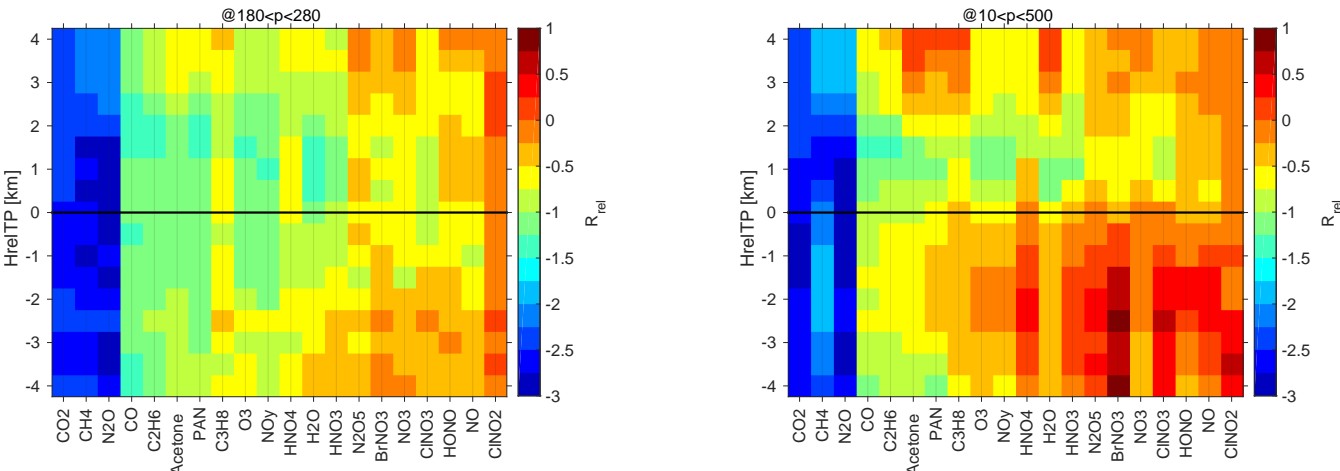

**Figure 12.** $R_{rel}$ calculated from $MOD_{CARIBIC}^{regular}$ and $MOD_{RANDLOC}$ with the range of $p$ set to $180\,hPa < p < 280\,hPa$ (left) and $10\,hPa < p < 500\,hPa$ (right). Low values indicate small climatological differences. The difference between the two panels shows the influence of expanding the limits in $p$ when calculating the climatological mean values with HrelTP used as a vertical coordinate.

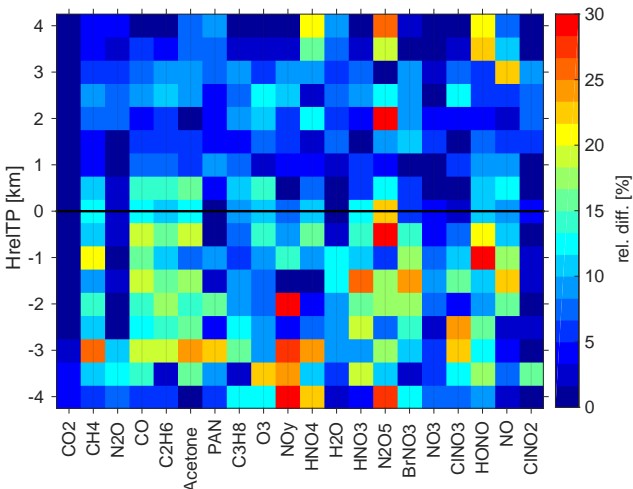

**Figure 13.** $|R_{rel}^A/R_{rel}^B - 1|$, given in percent. This is the fraction of the representativeness uncertainty introduced in $R_{rel}$ calculated from $MOD_{CARIBIC}^{regular}$ and $MOD_{RANDLOC}$ by including the Pacific ocean in $MOD_{RANDLOC}$, even though it is not sampled by $MOD_{CARIBIC}^{regular}$. Both, $text R_{rel}^A$ and $text R_{rel}^B$ have been calculed from $MOD_{CARIBIC}^{regular}$ and $MOD_{RANDLOC}$, excluding the Pacific in $MOD_{RANDLOC}$ in the calculation of $text R_{rel}^B$.

The strong increase in representativeness uncertainty is always present in measurement data from commercial aircraft, which can only collect data high above the tropopause when the tropopause is at high pressure and far below when it is at low pressure values. This bias is naturally contained in all data measured at constant pressure and then sorted relative to the tropopause and should be kept in mind when examining climatologies from corresponding platforms.

## A2   Limitation in longitude: The influence of the Pacific Ocean

As visible in Fig 1, there are no CARIBIC measurements over the Pacific Ocean, while $MOD_{RANDLOC}$ and $MOD_{RANDPATH}$ also cover the Pacific. The uncertainty introduced by taking the Pacific into account in $MOD_{RANDLOC}$ is investigated by calculating $R_{rel}$ from $MOD_{CARIBIC}^{regular}$ and $MOD_{RANDLOC}$ in two different setups. $R_{rel}$ is calculated from full $MOD_{RANDLOC}$ and $MOD_{CARIBIC}^{regular}$ (denoted by $R_{rel}^A$) and compared to $R_{rel}$ calculated with $MOD_{RANDLOC}$ limited in longitude $\lambda$ to $120°W < \lambda < 120°E$ (denoted by $R_{rel}^B$). The result is shown in Figure 13 as relative differences $|R_{rel}^A/R_{rel}^B - 1|$ between the two uncertainties. The relative differences show the share of the uncertainty inherent in $MOD_{CARIBIC}^{regular}$ because the Pacific is included in the reference dataset $MOD_{RANDLOC}$.

The importance of the Pacific depends on the species under consideration and whether the stratosphere or troposphere are considered. The influence on stratospheric values is very small for all species. In addition, those heights with less data (top and bottom) are most strongly influenced if the Pacific is not considered. For the long-lived species $CO_2$ and $N_2O$, the uncertainty increases only little (less than 3%) if the Pacific is included in the reference climatology of $MOD_{RANDLOC}$. But tropospheric

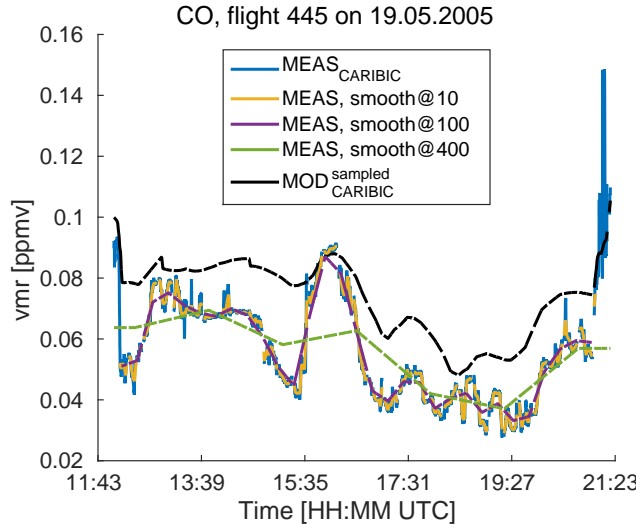

**Figure 14.** Timeseries of CO for flight 445 from Frankfurt to Tokyo. Shown is the time series of the interpolated model data and of the measurements. Measurements have been smoothed three times. The number indicates the length of the smoothing interval $N$.

$CH_4$ is more influenced by surface values. Interestingly, $ClNO_2$ is also not affected, which clearly shows that the effect does not depend on lifetime, but on the source regions and the chemistry. Acetone, CO and $C_2H_6$ are air pollutants with strong sources in Asia. Parts of these sources are excluded if the Pacific is not considered, which is why the inclusion of the Pacific in $MOD_{RANDLOC}$ is responsible for 15-20% of the total uncertainty. The situation is similar for $HNO_3$, $N_2O_5$, $BrNO_3$ and HONO. For the other species, the uncertainty introduced by the Pacific is smaller.

### Appendix B: Method of smoothing

This section shortly describes the method of smoothing used for creating the dataset $MEAS_{CARIBIC}^{smoothed}$.

Each species and each flight is considered separately. For smoothing a certain interval of the time series (consisting of a certain number of data points $N$), the time series is first cut into the corresponding number of pieces and the mean value of the $N$ datapoints calculated within each piece. In a second step, these mean values are associated with the center of each piece of the time series. Then, a linear interpolation is performed between the central points. The corresponding mean value is applied directly from the beginning of the flight to the center of the first interval and from the center of the last interval to the end of the flight. Finally, the gaps in the original time series are mapped onto the smoothed data. The original and the resulting smoothed time series are shown in Figure 14 for three different lengths of the smoothing interval $N$.

*Acknowledgements.* The authors would like to thank Andreas Engel for his work as editor and two anonymous referees, whose comments and the discussions they spawned improved the manuscript substantially. We would also like to thank Markus Hermann for his ongoing interest and support.

We thank all the members of the IAGOS-CARIBIC team, especially those who operate the CARIBIC container and Peter van Velthoven of KNMI who provides meteorological support. The collaboration with Lufthansa and Lufthansa Technik and the financial support from the German Ministry for Education and Science (grant 01LK1223C) are gratefully acknowledged. The CARIBIC measurement data analyzed in this paper can be accessed by signing the CARIBIC data protocol to be downloaded at http://www.caribic-atmospheric.com/.

This work was partially performed on the computational resource bwUniCluster funded by the Ministry of Science, Research and Arts and the Universities of the State of Baden-Württemberg, Germany, within the framework program bwHPC.

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
