# Peer review of "An assessment of the climatological representativeness of IAGOS-CARIBIC trace gas measurements using EMAC model simulations"

_Atmospheric Chemistry and Physics, 2016_

## Referee Comment (RC1) · Anonymous Referee #1 · 29 May 2016

The study by Eckstein et al. uses the EMAC model to identify the representativeness of CARIBIC aircraft measurements of the "state of the atmosphere" for different atmospheric trace gases. For this, the model is interpolated to aircraft flight tracks and time of the sampling and different statistics are applied to identify whether the interpolated profiles are characteristic of climatological averages.

The paper is well written and organized and the figures are appropriate. However, the question arises if the model can be used as an appropriate tool for the question. I think this question has not been addressed sufficiently in the paper. How well can data from a course model resolution be representative of the state of the atmosphere as described here? The representation of the model climatology vs. flight track interpolation

should depend on the models spatial and temporal resolution. If the grid or time span is too large (likely the case for global models), the model would not be able to represent the variability of the observations. A test would require to average the observations to the same model grid and then compare the variability.

Furthermore, I do not see any evaluation of the model. How well does the model represent the atmosphere? Especially water vapor is a gas that many models are not able to simulate appropriately, which is also the case for NOx and NOy. A discussion on how much this study depends on the performance of the model to represent chemical tracer should be added.

Finally, little has been done to identify reasons for differences between the flight track comparison and the global comparison, based on the atmospheric character of different trace gases dependent on the region for instance. Depending on region, airmasses experience more pollution, convection, stratosphere/troposphere exchange. The Pacific experiences a lot of pollution from South East Asia in some seasons than the Atlantic. Since CARIBIC data do not cover the Pacific, what implication does that have of the representation of the data compared to a global average? I would suggest, plotting a lon/lat map for a certain altitude level, say 1 km below the tropopause. This may help explain why some tracers are representative and why others may be not. Certainty 35-70 degrees is a very large region that covers a lot of different airmasses reaching from the tropics to the polar regions.

Detailed comments:

Page 3, Line 9. The assumption that species in the model show a similar variability has not been supported. A climatology of trace gases from the course model resolution is expected to show a much smaller variability than the observations. Wouldn't you expect a different result if you would run with a high model resolution spatial and temporal?

Page 4, line 15: Why is N2O5 counted twice, please explain.

[Figure]

Page 5: Line 6: is it +-4km (as stated above) or +- 4.25km?

Page 5, Line 17ff: Constraining the data to 35-75 degree N is not really removing different characteristics of tropical or polar airmasses and you would expect a larger variability. Earlier studies discussed differences in the characteristics of UTLS airmasses depending on the location with the jet stream and therefore with the height of the tropopause, which strongly varies with season. I think, constraining the comparison to 35-75 degrees N because of a good coverage of aircraft data would the better argument. There should be some discussion on the variability of the considered region.

Page 5, Line 23, if you define mid-latitude as 35-75deg, then please specify that here.

Page 6: Line 6-7: The temperature comparison for the data is taken from meteorological analysis. Are those the same that were used to nudge the model? That would explain the high correlation coefficient. Please clarify.

Page 7, Line 7-8: HrelTP does not look very similar to me. Distributions in the lower two rows in Figure 1 are more often above the TP than the flight track interpolated data. What implications will this have for the analysis?

Page 7: Line 18. The text describes that the variability of the model data if interpolated to the flight track is only 40-70% of the actual observed data. Further, it is discussed that the variability in the model cannot capture the small scale variability of the data. Then the assumption is made that the variability of the model is similar for all species. I do not follow this conclusion. Why is this the case?

Page 9: Line 19: How does the model represent CO2, N2O and CH4? If those are prescribed as fixed boundary conditions, certainly the model would not identify the variability that exists in the real data.

Page 13: I am not surprised about the different characteristics, since the different coverage of CARIBIC compared to the random distribution is very different, Figure 1 left column, the flight track sample more tropical air masses (being more concentrated

in the south). Furthermore, the Pacific with different characteristic of tracers are not sampled by the CARIBIC data set. It would help to see for example a figure of CO at the altitude considered for example 1 km below the tropopause.

A discussion on differences of the sampling location due to chemical characteristics that are different depending on sampling tropical or polar air masses, or characteristic longitudinal variability in different tracers would be helpful.

Page 17: typo line 2 "while it is can be much"

Page 17: Line 10: models usually have a poor representation of NO and NO2, especially in the UTLS it depends on lightning. Also convection is influencing NOx and can strongly vary with location, which is usually not well represented in models. Couldn't this be the reason why there is a larger uncertainty?

Line 14: How is the model representing H2O in the stratosphere?

Line 20; C2H6 and C3H8 are considered short-lived species with lifetimes of a few weeks or so.

Section 5.5 I think, the question should be changes for extended to: What would be a better regional coverage improve the statistic? This could be easily addressed within this paper, since one could extend the coverage over the pacific region, but keep the number of flights the same.

Conclusions: Page 21: Line 14: Sentence is unclear.

––––––––––––––––––––––––––––––––

---

## Referee Comment (RC2) · Anonymous Referee #2 · 19 Jun 2016

The manuscript "An assessment of the climatological representativeness of IAGOS-CARIBIC trace gas measurements using EMAC model simulations" aims at the characterisation and analysis of representativeness of the IAGOS-CARIBIC trace gas climatology for the upper troposphere/lower stratosphere (UTLS) as obtained from observations onboard commercial aircraft. Representativeness is analysed applying different sampling strategies, including the IAGOS-CARIBIC sampling, to a global scale chemistry transport model and evaluating the obtained populations by means of statistical tests and descriptors. The findings of this study are certainly relevant to the IAGOS-CARIBIC programme as a whole, since they challenge the use of IAGOS-CARIBIC data in the climatological sense. In general, the manuscript is well written and tries to

justify the applied methodologies. However, there are several areas where the choice of methods seems to be rather arbitrary and needs either additional justification or analysis to corroborate the conclusions reached in the manuscript. After these issues are addressed this contribution should be well suited for publication in Atmospheric Chemistry and Physics.

**Major comments**

**Global scale chemistry transport model**

There are two major concerns about using the EMAC model as a reference state of the atmosphere. First, the model description in the text is insufficient. It needs to be mentioned how the model was validated against other independent observations. For which species did the model perform well and for which not? Where is the model insufficient to reproduce variability on the scale given by the model resolution? This is especially important since one may suspect that the model will have difficulties reproducing vertical trace gas gradients in the UTLS region. Second, as shown in Figure 1 the model has only 3 levels in the UTLS region and output was only available every 12-hour. Therefore, the model misses large parts of the real variability (see also the CARIBIC comparison). How can it be justified that the model can still be assessed to analyse representativeness?

**Sampling strategy**

Several choices seem to be arbitrary. I especially don't understand why the temporal domain is not sampled as a whole. Both sampling patterns RANDPATH and RANDLOC only sample 12 and 8 days per month, respectively. It would seem more appropriate to sample daily but on the other hand with a more realistic pattern that resembles that of the CARIBIC flights (i.e., on great arcs between major airports in the northern hemisphere, leaving out transpacific flights, since this region is never covered by CARIBIC). In that case the RANDPATH sampling could be viewed as the maximal achievable sampling pattern by commercial aircraft and RANDLOC could still be seen as sampling the
northern hemispheric UTLS region as a whole.

**Selected statistical measures**

Again there seem to be arbitrary choices concerning the statistical estimators and tests. If the Komogorov-Smirnov test turned out to be too strict because it requires similarity of the whole distribution, why did you not select other statistical tests that only evaluate one statistical parameter at a time (e.g., Mann-Whitby test for the mean and Levene's or Brown–Forsythe test for variance, all are non-parametric tests suited for atmospheric trace gas observations). Furthermore, the results need to be discussed together with observed seasonality of the trace species as is mentioned by the authors themselves on page 17, line 1, but than dropped without further reasoning 3 lines later. The relative difference does not contain much information in itself and as stated correctly depends on the lifetime of a species.

**Minor comments**

P1,L11: "formulated above". Not clear from the context where this was formulated

P3,L28ff: Although no details on the measurement techniques are needed here, it would still be interesting to learn something about the overall uncertainties of the measurements and how these compare to the later discussion of representativeness.

P4,L10: Model output every "eleven hours"? Did you mean 12 hours?

P4,L9ff: Additional information on emissions used in EMAC and vertical resolution in the UTLS region would be useful here.

Section 3.1: It should be more prominently mentioned in the first paragraph of this section that you restrict the analysis to the latitude region 35N to 75N. Details follow towards the end of the section and can remain there, but it would be good to make this important detail clear from the beginning. It should also be stated in the abstract.

Table 1: For RANDPATH it is an adjusted Gaussian distribution, as mentioned in the

text.

Table 1 and elsewhere: "Uniform" or "rectangular" distribution should be used instead of "even".

P6,L6f: The good correlation for temperature is not a big surprise, given the strong vertical stratification in the UTLS and the assumably large number of measurements. Since this is one of the few pieces of model validation mentioned, one could add a scatter plot to the supplement.

P6,L9f: It is not clear to me why the limited vertical model resolution is the reason you cannot compare CARIBIC directly to EMAC. The random sampling is still done using vertical interpolation to specific pressure levels. Would't the same argument apply to the random sampling strategy as well and could one not simply drop it and do the analysis of representativeness on discreet model levels instead?

P6,L20f: Why did you choose these cut-off values instead of simply using the standard deviation as a criterion (i.e., redistribute values outside +/- 2 sigma). I don't assume this would change much, but would seem statistically more sound. Alternatively, one could have sampled directly from the observed CARIBIC distribution.

P7,L7ff: I don't agree with the statement that the distribution "is very similar for all datasets". There is a strong offset to higher HrelTP in both random sampling strategies. What is the actual mean HrelTP for all three samples?

p7,L11f: This requires some further justification (see major comment above). Without being aware of the details of Jöckel et al 2015, it seems a bit hard to believe that the model performs equally well for the very different set of species analysed here. There should be additional discussion of the species for which this may not be justified.

p9,l19: How was the mean $tau_star$ $calculated$? $As$ $the$ $mean$ $over$ $all$ $monthly$ $tau_star$ $or$ $as$ $a$ $tau_star$ $of$ $the$ $mean$ $mu$ $and$ $sigma$?

Figure2: It would be interesting to add CARIBIC observed $tau_star$ $in$ $the$ $figure$ $(where$ $available)$.

Figure3 and others: The y-axis if often titled "variability". It would be useful to give a more concrete title, since the manuscript is dealing with all kinds of variability. This could reduce confusion. In this specific case I assume this is relative standard deviation?

p12,l18: "The differences are small, mostly below an absolute value of 0.15." But this means that the absolute difference between both samples is 1.4 times larger than the value of the reference (or am I mistaken). I am not sure that I would call this small! In general using the log scaled relative difference seems a bit odd and only confuses. Why not use the relative difference as is?

p12,l29f: "A similar analysis has also been performed with data from a random number generator, leading to equivalent results." Are you referring to the RANDLOC sample here?

p13,l13: At least repeat the result of the sensitivity study here. The supplement should not be a paper on its own.

p16,l5: Not clear which correlation is referred to here.

p16,l6: What is an "uncertainty error"? I think the use of representativeness uncertainty would in general work better.

p16,l8-13: This description is completely confusing. I don't understand what is done and why. Please improve the description.

p17,l10ff: Since the discussion on NOx is along the EMAC results, it would be interesting to know how NOx sources in the UTLS are treated in the model. Does the model include a realistic representation of lightning NOx? Has this been analysed in previous studies?

p17,l33f: The representativeness uncertainty of 5lived trace gases is huge considering their atmospheric abundance. It is much larger than their seasonal variability. This aspect needs to be considered in the analysis and discussed along with the results.
p18,l20ff: Finally there is some discussion using species specific thresholds, but again these thresholds are chosen without any justification. They should be related to seasonal variability.

p18,l32: Was it ever shown that $R_{rel}$ "increases linearly"? Maybe an increasing relationship, but linearly?

Figures: It would be easier to follow the discussion of the figures if sub-panels would be labelled by letters (which is Copernicus style). For example discussion of Figure 8 on page 18.

Figure 1 in supplement: Please explain black line in legend and add fit as additional line to the plot.Indicate which species are behind each point. Is the given fit applied to log(meas) and log(model)? Is it just my impression or does the model actually capture less of the variability for species that have a small relative variability? How could this be explained? I would have expected the opposite.

**Technical comments:**

P1,L2: It is "representative of" not "representative for".

P5,L6 and elsewhere: "Data" is always plural. Change to "Data were used ..."

---

## Author Comment (AC1) · 9 Sep 2016

**Reply to Referee Comments**

Dear Referees,

thank you for your comments and remarks to the manuscript. Reconsidering the points you addressed have substantially improved the manuscript.

We have included a new chapter (now chapter 5), which we devote to the comparison of the model and measurements. We compare the variability in the two datasets in order justify the conclusion of the study, in which we apply the results from a model study to measurements.

In addition, numerous other and small changes have been made. We no longer include NO2, as there are very few measurements. The paper now includes an appendix, which describes the influence of the Pacific ocean and on limits in pressure on the results.

In the following, we present a point by point reply to your suggestions, followed by the marked up manuscript. As a reference and for better readability, we have also added the manuscript as such.

Best regards,

Johannes Eckstein

**Referee 1**

Major Comments

1. *However, the question arises if the model can be used as an appropriate tool for the question. I think this question has not been addressed sufficiently in the paper. How well can data from a course model resolution be representative of the state of the atmosphere as described here? The representation of the model climatology vs. flight track interpolation should depend on the models spatial and temporal resolution. If the grid or time span is too large (likely the case for global models), the model would not be able to represent the variability of the observations. A test would require to average the observations to the same model grid and then compare the variability.*

   As noted above, we have now included a separate section (Sec. 5) that treats this question. We show the influence of the small scale variability on climatological mean values and discuss the differences between model and measurement variability on longer time scales.

2. *Furthermore, I do not see any evaluation of the model. How well does the model represent the atmosphere? Especially water vapor is a gas that many models are not able to simulate appropriately, which is also the case for NOx and NOy. A discussion on how much this study depends on the performance of the model to represent chemical tracer should be added.*

   The new section also covers the differences between species. A detailed validation is beyond the scope of this study, as we use the model as a tool for a different purpose. The section describing the model has been expanded, including more references. A validation of the model is not the focus of this study, but described by Hegglin (JGR), 2010.

3. *Finally, little has been done to identify reasons for differences between the flight track comparison and the global comparison, based on the atmospheric character of different trace gases dependent on the region for instance. Depending on region, airmasses experience more pollution, convection, stratosphere/troposphere exchange. The Pacific experiences a lot of pollution from South East Asia in some seasons than the Atlantic. Since CARIBIC data do not cover the Pacific, what implication does that have of the representation of the data compared to a global average? I would suggest, plotting a lon/lat map for a certain altitude level, say 1 km below the tropopause. This may help explain why some tracers are representative and why others may be not. Certainty 35-70 degrees is a very large region that covers a lot of different airmasses reaching from the tropics to the polar regions.*

   We have included a section in the appendix that assesses the influence of the Pacific (Sec. A1). The influence on climatological mean values is stronger for those species determined by source regions in Asia.

Minor Comments

1. *Page 3, Line 9. The assumption that species in the model show a similar variability has not been supported. A climatology of trace gases from the course model resolution is expected to show a much smaller variability than the observations. Wouldn't you expect a different result if you would run with a high model resolution spatial and temporal?*

   We now include a new section (Sec. 5) which treats the differences in model and measurement variability. Setting up or running a model run with a higher resolution is beyond the scope of this paper.

2. *Page 4, line 15: Why is N2O5 counted twice, please explain.*

   N2O5 is measured by catalytic conversion to NO. One N2O5 molecule yields two NO molecules, this is why every N has to be counted. This is explained in the manuscript.

3. *Page 5: Line 6: is it +-4km (as stated above) or +- 4.25km?*

   It has been corrected here. It is +-4.25km, but heights are labeled with their centers, which corresponds to +-4km.

4. *Page 5, Line 17ff: Constraining the data to 35-75 degree N is not really removing different characteristics of tropical or polar airmasses and you would expect a larger variability. Earlier studies discussed differences in the characteristics of UTLS airmasses depending on the location with the jet stream and therefore with the height ofthe tropopause, which strongly varies with season. I think, constraining the comparison to 35-75 degrees N because of a good coverage of aircraft data would the better argument. There should be some discussion on the variability of the considered region.*

   True, the good coverage was also an argument that we now state in the text. The latitudinal limit is for sure not sufficient to exclude all influence of lower higher or lower latitudes, but is a first approximation. We do discuss data relative to the local tropopause, as all fields are presented in HrelTP.

5. *Page 5, Line 23, if you define mid-latitude as 35-75deg, then please specify that here.*

   We have added a comment specifying this.

6. *Page 6: Line 6-7: The temperature comparison for the data is taken from meteorological analysis. Are those the same that were used to nudge the model? That would explain the high correlation coefficient. Please clarify.*

   Temperature measured by CARIBIC is not considered for ERA-Interim, which was used for nudging the model.

7. *Page 7, Line 7-8: HrelTP does not look very similar to me. Distributions in the lower two rows in Figure 1 are more often above the TP than the flight track interpolated data. What implications will this have for the analysis?*

   We have reformulated the paragraph and revised our judgment. The reasoning is different: Both, the distribution in HrelTP and the different climatologies, are influenced by the sampling pattern. So the differences that show up in HrelTP do not imply differences in the climatologies, but both are influenced for the same reasons.

8. *Page 7: Line 18. The text describes that the variability of the model data if interpolated to the flight track is only 40-70% of the actual observed data. Further, it is discussed that the variability in the model cannot capture the small scale variability of the data. Then the assumption is made that the variability of the model is similar for all species. I do not follow this conclusion. Why is this the case?*

   This paragraph has been completely revised and a new section now covers this subject.

9.  *Page 9: Line 19: How does the model represent CO2, N2O and CH4? If those are prescribed as fixed boundary conditions, certainly the model would not identify the variability that exists in the real data.*

    Boundary conditions are not fixed. For CO2, N2O and CH4, they are prescribed as latitude dependent monthly means. We have included a short paragraph in the text on the boundary conditions of chemical species.

10. *Page 13: I am not surprised about the different characteristics, since the different coverage of CARIBIC compared to the random distribution is very different, Figure 1 left column, the flight track sample more tropical air masses (being more concentrated in the south). Furthermore, the Pacific with different characteristic of tracers are not sampled by the CARIBIC data set. It would help to see for example a figure of CO at the altitude considered for example 1 km below the tropopause. A discussion on differences of the sampling location due to chemical characteristics that are different depending on sampling tropical or polar air masses, or characteristic longitudinal variability in different tracers would be helpful.*

    Whether the climatologies produced by the sampling pattern of CARIBIC are representative is just the question that we are investigating in this study. Regional differences are another, interesting subject, which is more difficult to investigate with CARIBIC data. As a first step, the influence of the Pacific ocean is included as part of the appendix of the paper.

11. *Page 17: typo line 2 "while it is can be much"*

    The typo has been corrected.

12. *Page 17: Line 10: models usually have a poor representation of NO and NO2, especially in the UTLS it depends on lightning. Also convection is influencing NOx and can strongly vary with location, which is usually not well represented in models. Couldn't this be the reason why there is a larger uncertainty?*

    NOx production resulting from lightning activity is included in the model (Grewe et al., 2001). The geographical restraint of CARIBIC flight routes to flight corridors and thereby to the regions with high VMR of NOx has the stronger influence on representativeness.

13. *Line 14: How is the model representing H2O in the stratosphere?*

    In the stratosphere and mesosphere the chemical H2O tendency (due to the methane oxidation) is calculated with the help of the chemical submodel MECCA (Sander et al., 2005).

14. *Line 20; C2H6 and C3H8 are considered short-lived species with lifetimes of a few weeks or so.*

    We have changed the description to moderately long-lived.

15. *Section 5.5 I think, the question should be changes for extended to: What would be a better regional coverage improve the statistic? This could be easily addressed within this paper, since one could extend the coverage over the pacific region, but keep the number of flights the same.*

    The influence of the Pacific is now covered in the appendix (Sec. A1). A more detailed study of the influence of different regions could be the subject future studies.

16. *Conclusions: Page 21: Line 14: Sentence is unclear.*

    The sentence has been reworded.

**Literature**

- Grewe, V., Brunner, D., Dameris, M., Grenfell, J. L., Hein, R., Shindell, D., and Staehelin, J.: Origin and variability of upper tropospheric nitrogen oxides and ozone at northern mid-latitudes, Atmos. Environ., 35, 3421–3433, 10.1016/S1352-2310(01)00134-0, 2001

- Hegglin, M. I., Gettelman, A., Hoor, P., Krichevsky, R., Manney, G. L., Pan, L. L., Son, S.-W., Stiller, G., Tilmes, S., Walker, K. A., Eyring, V., Shepherd, T. G., Waugh, D., Akiyoshi, H., Añel, J. A., Austin, J., Baumgaertner, A., Bekki, S., Braesicke, P., Brühl, C., Butchart, N., Chipperfield, M., Dameris, M., Dhomse, S., Frith, S., Garny, H., Hardiman, S. C., Jöckel, P., Kinnison, D. E., Lamarque, J. F., Mancini, E., Michou, M., Morgenstern, O., Nakamura, T., Olivié, D., Pawson, S., Pitari, G., Plummer, D. A., Pyle, J. A., Rozanov, E., Scinocca, J. F., Shibata, K., Smale, D., Teyssèdre, H., Tian, W., and Yamashita, Y.: Multimodel assessment of the upper troposphere and lower stratosphere: Extratropics, Journal of Geophysical Research: Atmospheres, 115, doi:10.1029/2010JD013884, 2010.

- Sander, R., Kerkweg, A., Jöckel, P., and Lelieveld, J.: Technical note: The new comprehensive atmospheric chemistry module MECCA, Atmos. Chem. Phys., 5, 445–450, doi:10.5194/acp-5-445-2005, 2005

[revised manuscript text omitted]
 20 min variability of $MOD_{CARIBIC}^{sampled}$. Right panel: Mean relative differences of $MEAS_{CARIBIC}^{smoothed}$ and $MEAS_{CARIBIC}$. $MEAS_{CARIBIC}^{smoothed}$ has been smoothed to have the same 20 min variability as $MOD_{CARIBIC}^{sampled}$, using the smoothing number from the left hand panel. The relative differences correspond to the error in the climatologies of $MOD_{CARIBIC}^{sampled}$ due to the coarse model resolution. $N_2O$, $C_2H_6$ and $C_3H_8$ are measured by air samples with a low measurement frequency and therefore not considered here.

model could have other deficiencies in the description of the different species. These are made visible in the following section by comparing model and measurement variability directly.

**5.2 Comparing model and measurement variability**

In this section, the variability of $MOD_{CARIBIC}^{sampled}$ is compared directly to that of $MEAS_{CARIBIC}^{smoothed}$. For this dataset, $MEAS_{CARIBIC}$
5 has been altered in such a way to reproduce the 20 min variability of $MOD_{CARIBIC}^{sampled}$, see the preceeding section. As this study argues completely within the model world, it is important that the model has similar values for the variability, which is used as an indicator of the underlying complexity. If the model cannot reproduce the measurement variability at all, it is not plausible why conclusions on representativeness drawn from model data should also be true for the real atmosphere.

As has been discussed in Sec. 4.2, variability depends on the time scale for which it is considered. In order to evaluate the
10 model performance, we compare $\sigma_r$ on time scales of 30 d and 1 a. 30 d variability includes data from typically 4 flights, so this is a measure for the atmospheric variabilty on the global, large scale dynamics. 1 a variability gives a good impression of the annual cycle, as it includes data from many flights and different years. Figure 6 shows $\sigma_r^{MOD}/\sigma_r^{MEAS}$ for time scales of 30 d (left) and 1 a (right), using the datasets $MOD_{CARIBIC}^{sampled}$ and $MEAS_{CARIBIC}^{smoothed}$.

[Figure]

**Figure 6.** $\sigma_r^{MOD}/\sigma_r^{MEAS}$ given in percent for time scales of $30\,\mathrm{d}$ (left) and $1\,\mathrm{a}$ (right), where MOD stands for $MOD_{CARIBIC}^{sampled}$ and MEAS stands for $MEAS_{CARIBIC}^{smoothed}$. Values greater than $50\,\%$ indicate the high model complexity.

Figure 6 shows that the variability in the measurements reached by the model differs between species. In general, the variability reached for shorter lived species better fits that of the measurements. Short-lived species also undergo a more complex chemistry in the model, which adds variability. The $30\,\mathrm{d}$ variability shown in Figure 6 (left) reveals to what extent the model is able to capture variability related to the large scale dynamics. Most species reach $40\text{-}80\,\%$. NO is very short lived and
5    strongly determined by its daily cycle, which is the reason why the variability in the model reaches higher values.

The time scale of $1\,\mathrm{a}$ shows the variability that represents seasonality. The model does a better job for this time scale than for $30\,\mathrm{d}$, short lived species and $CO_2$ reaching well over $60\,\%$ of the variability, approaching $100\,\%$ for some species. Here again, the model chemistry increases the variability for shorter lived species to the right. There are species that are not as well represented, while this also depends on the height considered (e.g. high values for stratospheric $N_2O$).

10    The model variability is influenced by many factors including the dynamics, the representation of the chemistry and of the sources included in the model. The limited horizontal and vertical resolution also plays a role, even though $MEAS_{CARIBIC}^{smoothed}$ is used as a reference for the comparison. If compared to the original $MEAS_{CARIBIC}$, the percentages of variability reached by the model drop by $10\text{-}20\,\%$ (not shown). It is beyond the scope of this paper to further disentangle what causes the deficiencies of the model and what leads to the differences between the species.

15    As is shown in Figure 6, the model reaches more than $50\,\%$ of the variability of the measurements. This ratio depends strongly on the species and is higher for longer time scales. This points at a high complexity of the model and justifies the assumption underlying this study: The representativeness evaluated from the model data alone 
[revised manuscript text omitted]
_{CARIBIC}$ (CARIBIC measurements) and $MOD_{CARIBIC}$ in each month. $\sigma_r^{MOD_{CARIBIC}}$ and $\sigma_r^{MEAS_{CARIBIC}}$ were calculated in each month. Figure ?? shows the correlation of $\sigma_r^{MOD_{CARIBIC}}$ and $\sigma_r^{MEAS_{CARIBIC}}$. Monthly variability $\sigma_r$ of $MOD_{CARIBIC}$ over $MEAS_{CARIBIC}$. Colorcoding corresponds to the variability $\tau^*$ of each species. Data closer to the tropopause is plotted as larger circles.

As discussed in the main text, $\sigma_r^{MOD_{CARIBIC}}$ reaches 40 to 70 % of $\sigma_r^{MEAS_{CARIBIC}}$ for all species. The correlation coefficient of the two is 0.81. This shows that the model variability is similar for all species, justifying the use of results from the model datasets for CARIBIC measurements$_{CARIBIC}^{regular}$.

**2 Calculating representativeness from random numbers**

All three methods to investigate representativeness (Kolmogorov-Smirnov test, variability analysis and relative differences) have also been applied to data created with a random number generator. The results of this study are  presented here.

To produce the random numbers, 20 sets of $10^8$ numbers were taken from a normal distribution. These 20 sets are referred to as species, well aware of the fact that they are purely artificial. From species to species, the standard deviation $\sigma$ was set to vary from $10^{-3}$ to $10^3$, values of the exponent  increasing linearly. 20 mean values $\mu$ (increasing from $10^4$ to $10^8$, with a linear increase in the exponent) where distributed randomly  to the 20 species. This results in 20 species with different values for $\sigma$ and $\mu$. The statistics of each species will be indexed by the number 2. For short, this dataset will be called RAND.

3000 samples were taken from each of the 20 species. For each sample, 20 numbers were first randomly drawn from each species. These new numbers and all those that had been drawn before then make up this one sample. So the size increases by 20 for each sample. This way, the relationship of the representativeness score with the sample size is directly accessible.  Samples are indexed by the number 1.

The variability $\tau^*$ of each species  is defined as in Equation  3 of the main text: $\tau^* = \log_{10}(\sigma_2/\mu_2)$. The two requirements set up in Section 3.3 for representativeness in general also have to hold here:

1. Representativeness has to increase with the number of samples.

2. Representativeness has to decrease with increasing variability of the underlying distribution.

With RAND defined in this way, it is possible to test representativeness using the variability analysis following Rohrer and Berresheim (2006) and Kunz et al. (2008) (see Section 4.2) and the relative differences (see Section 4.3). The Kolmogorov-Smirnov test was positive for very few samples (less than fifty numbers, independent of $\tau^*$) and will not be further discussed. Its behaviour with aircraft data was subject of a sensitivity study, the results of which are shown in Sec. 3 of this supplement.

**2.1 Variability analysis**

The  variability analysis (defined in Section 4.2 and Eq. 3) was applied in a simplified manner. As RAND is independent of time, $R_{var}$ is reduced to just a single value containing the absolute difference of variability of each species of RAND and the sample taken thereof: $R_{var} = |\nu_1 - \nu_2|$, where $\nu$ is the mean variability. Figure 1 shows a result. The exact result is a matter of chance, as a random number generator is used. Similar to using $MOD_{CARIBIC}^{regular}$  and $MOD_{RANDPATH}$, a strong dependance on $\tau^*$ and a weak dependance on the number of samples is visible.

Similar to $R_{var}$ when using $MOD_{CARIBIC}^{regular}$  and $MOD_{RANDPATH}$, the variability analysis using RAND meets the two requirements necessary for describing representativeness, which were described in Section 3.3 and above. This result supports the  finding that $R_{var}$ can be used as a statistic for describing representativeness.

[Figure]

**Figure 1.** Representativeness score $R_{var}$ applied to RAND. Vertical lines indicate the values of $\tau^*$ of each species.

**2.2 Relative differences**

Similar to $R_{var}$, $R_{rel}$ is reduced to a simple relative difference when using RAND: $R_{rel} = |\mu_1 - \mu_2|/\mu_2$, where $\mu$ is the mean of the sample (index 1) and of the whole subset (index 2). Figure 2 shows  a result when applying $R_{rel}$ to RAND. The dependance on $\tau^*$ is strong and linear. The result also depends on the number of samples, showing a slow increase with the number of samples. This dependance is sometimes disturbed by better values which are reached by chance when drawing from RAND.

Like for $MOD_{CARIBIC}^{regular}$  and $MOD_{RANDPATH}$, $R_{rel}$ passes both conditions for a valid description of representativeness: it depends on variability $\tau^*$ and on the number of samples. The latter is also being influenced by chance and generally much weaker.

The fact that $R_{rel}$ passes the two conditions for a description of representativeness can be understood with some theoretical considerations. The standard error of the mean is defined by

$$\sigma_{\overline{x}} = \frac{\sigma}{\sqrt{n}} \tag{1}$$

where $\sigma_{\overline{x}}$, the standard deviation of a sample, can be given by the following equation ($N$ being the number of samples):

$$\sigma_{\overline{x}} = \sqrt{\frac{1}{N}\sum_{i=1}^{N}(\overline{x}_i - \mu)^2} \tag{2}$$

For $N = 1$, this gives:

$$\sigma_{\overline{x}} = |\overline{x}_i - \mu| \tag{3}$$

[Figure]

**Figure 2.** Like Figure 1, but for $R_{rel}$.

Plugging Eq. 3 into Eq. 1 gives:

$$\frac{|\overline{x}_i - \mu|}{\mu} = \frac{\sigma}{\mu\sqrt{n}} = \frac{10^{1/\tau^*}}{\sqrt{n}} \tag{4}$$

and therefore

$$R_{rel} = \log_{10}\left(\frac{|\overline{x} - \mu|}{\mu}\right) = -0.5\log_{10}(n) + \frac{1}{\tau^*} \tag{5}$$

5 So ideally, $R_{rel}$ should depend inversely on $\tau^*$ and directly on the logarithm of the number of values. Figure 2 shows this is approximately true for RAND.

In the case of RAND, $R_{rel}$  and $R_{var}$ can both be used to describe representativeness as  they pass the two conditions. Theoretical considerations make the finding plausible for $R_{rel}$. RAND can be considered a theoretical abstraction of MOD. The finding here therefore strongly supports that of Sections 5.2 and 5.3, where $R_{rel}$ and $R_{var}$

10 have also been found to be good descriptors of representativeness when using $MOD_{\overline{CARIBIC}}^{regular}$ and $MOD_{RANDPATH}$ or $MOD_{RANDLOC}$. In the main text, we use $R_{rel}$ for final results, as it more suitable to answer the question of representativeness for a climatology.

**3 Sensitivity study on the Kolmogorov-Smirnov test**

When  applying the Kolmogorov-Smirnov test to $MOD_{\overline{CARIBIC}}^{regular}$, $MOD_{RANDPATH}$ or $MOD_{RANDLOC}$,

15  it returned almost only negative results. This indicates that $MOD_{\overline{CARIBIC}}^{regular}$ 
[revised manuscript text omitted]
 to the 20 species. This results in 20 species with different values for $\sigma$ and $\mu$. The statistics of each species will be indexed by the number 2. For short, this dataset will be called RAND.

3000 samples were taken from each of the 20 species. For each sample, 20 numbers were first randomly drawn from each species. These new numbers and all those that had been drawn before then make up this one sample. So the size increases by 20 for each sample. This way, the relationship of the representativeness score with the sample size is directly accessible. Samples are indexed by the number 1.

The variability $\tau^*$ of each species is defined as in Equation 3 of the main text: $\tau^* = \log_{10}(\sigma_2/\mu_2)$. The two requirements set up in Section 3.3 for representativeness in general also have to hold here:

1. Representativeness has to increase with the number of samples.

2. Representativeness has to decrease with increasing variability of the underlying distribution.

With RAND defined in this way, it is possible to test representativeness using the variability analysis following Rohrer and Berresheim (2006) and Kunz et al. (2008) (see Section 4.2) and the relative differences (see Section 4.3). The Kolmogorov-Smirnov test was positive for very few samples (less than fifty numbers, independent of $\tau^*$) and will not be further discussed. Its behaviour with aircraft data was subject of a sensitivity study, the results of which are shown in Sec. 3 of this supplement.

[Figure]

**Figure 1.** Representativeness score R$_{var}$ applied to RAND. Vertical lines indicate the values of $\tau^*$ of each species.

**2.1 Variability analysis**

The variability analysis (defined in Section 4.2 and Eq. 3) was applied in a simplified manner. As RAND is independent of time, R$_{var}$ is reduced to just a single value containing the absolute difference of variability of each species of RAND and the sample taken thereof: R$_{var} = |\nu_1 - \nu_2|$, where $\nu$ is the mean variability. Figure 1 shows a result. The exact result is a matter of

5  chance, as a random number generator is used. Similar to using MOD$_{CARIBIC}^{regular}$ and MOD$_{RANDPATH}$, a strong dependance on $\tau^*$ and a weak dependance on the number of samples is visible.

Similar to R$_{var}$ when using MOD$_{CARIBIC}^{regular}$ and MOD$_{RANDPATH}$, the variability analysis using RAND meets the two requirements necessary for describing representativeness, which were described in Section 3.3 and above. This result supports the finding that R$_{var}$ can be used as a statistic for describing representativeness.

10  ## 2.2 Relative differences

Similar to R$_{var}$, R$_{rel}$ is reduced to a simple relative difference when using RAND: R$_{rel} = |\mu_1 - \mu_2|/\mu_2$, where $\mu$ is the mean of the sample (index 1) and of the whole subset (index 2). Figure 2 shows a result when applying R$_{rel}$ to RAND. The dependance on $\tau^*$ is strong and linear. The result also depends on the number of samples, showing a slow increase with the number of samples. This dependance is sometimes disturbed by better values which are reached by chance when drawing from RAND.

15  Like for MOD$_{CARIBIC}^{regular}$ and MOD$_{RANDPATH}$, R$_{rel}$ passes both conditions for a valid description of representativeness: it depends on variability $\tau^*$ and on the number of samples. The latter is also being influenced by chance and generally much weaker.

[Figure]

**Figure 2.** Like Figure 1, but for $R_{rel}$.

The fact that $R_{rel}$ passes the two conditions for a description of representativeness can be understood with some theoretical considerations. The standard error of the mean is defined by

$$\sigma_{\overline{x}} = \frac{\sigma}{\sqrt{n}} \tag{1}$$

where $\sigma_{\overline{x}}$, the standard deviation of a sample, can be given by the following equation ($N$ being the number of samples):

$$\quad \sigma_{\overline{x}} = \sqrt{\frac{1}{N} \sum_{i=1}^{N} (\overline{x}_i - \mu)^2} \tag{2}$$

For $N = 1$, this gives:

$$\sigma_{\overline{x}} = |\overline{x}_i - \mu| \tag{3}$$

Plugging Eq. 3 into Eq. 1 gives:

$$\frac{|\overline{x}_i - \mu|}{\mu} = \frac{\sigma}{\mu\sqrt{n}} = \frac{10^{1/\tau^*}}{\sqrt{n}} \tag{4}$$

10 and therefore

$$R_{rel} = \log_{10}\left(\frac{|\overline{x} - \mu|}{\mu}\right) = -0.5\log_{10}(n) + \frac{1}{\tau^*} \tag{5}$$

So ideally, $R_{rel}$ should depend inversely on $\tau^*$ and directly on the logarithm of the number of values. Figure 2 shows this is approximately true for RAND.

[Figure]

**Figure 3.** Flightroutes to Vancoucer, Canada, where each flight has been cut into 20 pieces and randomly chosen 30% of those pieces have been plotted. These are tested against the whole data from flights to Vancouver to give one point in Figure 4.

In the case of RAND, $R_{rel}$ and $R_{var}$ can both be used to describe representativeness as they pass the two conditions. Theoretical considerations make the finding plausible for $R_{rel}$. RAND can be considered a theoretical abstraction of MOD. The finding here therefore strongly supports that of Sections 5.2 and 5.3, where $R_{rel}$ and $R_{var}$ have also been found to be good descriptors of representativeness when using $MOD_{CARIBIC}^{regular}$ and $MOD_{RANDPATH}$ or $MOD_{RANDLOC}$. In the main text, we use $R_{rel}$ for final results, as it more suitable to answer the question of representativeness for a climatology.

**3  Sensitivity study on the Kolmogorov-Smirnov test**

When applying the Kolmogorov-Smirnov test to $MOD_{CARIBIC}^{regular}$, $MOD_{RANDPATH}$ or $MOD_{RANDLOC}$, it returned almost only negative results. This indicates that $MOD_{CARIBIC}^{regular}$ is not representative of $MOD_{RANDPATH}$ in the definition of the Kolmogorov-Smirnov test. This behaviour was tested in a sensitivity study, the results of which are discussed here.

One of the most frequent destinations within the CARIBIC project is Vancouver, Canada (near $120\,^{\circ}$W, $45\,^{\circ}$N, see Figure 3), and only the subset of $MOD_{CARIBIC}^{regular}$ to this destination is considered in this example to minimize effects that may come from different flight routes. Parts of this reduced dataset were tested with the Kolmogorov-Smirnov test against the whole reduced dataset for all variables. To produce these partial datasets, each flight was cut into an increasing number of pieces (corresponding to a certain time) and different percentages of these pieces were used in testing. Figure 3 shows an example of applying this method, by cutting each flight into 20 pieces and taking 30% of these by showing the corresponding flightpaths.

[Figure]

**Figure 4.** The Kolmogorov-Smirnov test applied to the flights to Vancouver, Canada, of $\text{MOD}_{\text{CARIBIC}}^{\text{regular}}$ and subsets of these flights. Dotted lines indicate those lengths in time and those percentages that were tested. 0 stands for a passing the Kolmogorov-Smirnov test, 1 for not passing.

Data was not binned in months. When applying the Kolmogorov-Smirnov test without binning in months, the result is a profile in HrelTP for each variable. The result can then be diplayed in similar way to Figures 7 and 8. This matrix of height versus species was calculated for each combination of number of pieces and percent of pieces. In each combination, all the profiles of the different variables were averaged to end up with one value betwween 1 and 0 characterizing the result of the test
5   for this combination of number of pieces and percent of pieces. The result can then give an impression of the strictness of the Kolmogorov-Smirnov test.

Figure 4 shows the result of the study. Independent of the number of pieces, the result is positive if all pieces are considered, as the definition of the test prescribes. But only when removing short pieces (shorter than $20\,\text{min}$) is the result also positive for less pieces, even though 70% percent of the data is still needed. When removing whole flights (at the top of the plot), more the
10   90% of the data has to be taken into account to achieve a positive result of the Kolmogorov-Smirnov test. This result is very similar also for other error probabilities $\alpha$, taking values of 0.001, 0.01, 0.05 (in the figure), 0.1 and 0.2. The area of failing increases only slightly with the error probability. This showcases the strictness of the test. The Kolmogorov-Smirnov test does not seem suitable to test a dataset measured with aircraft for representativeness of a larger dataset.

---

## Author Comment (AC2) · 9 Sep 2016

**Reply to Referee Comments**

Dear Referees,

thank you for your comments and remarks to the manuscript. Reconsidering the points you addressed have substantially improved the manuscript.

We have included a new chapter (now chapter 5), which we devote to the comparison of the model and measurements. We compare the variability in the two datasets in order justify the conclusion of the study, in which we apply the results from a model study to measurements.

In addition, numerous other and small changes have been made. We no longer include NO2, as there are very few measurements. The paper now includes an appendix, which describes the influence of the Pacific ocean and on limits in pressure on the results.

In the following, we present a point by point reply to your suggestions, followed by the marked up manuscript. As a reference and for better readability, we have also added the manuscript as such.

Best regards,

Johannes Eckstein

**Referee 2**

Major Comments

1. *Global scale chemistry transport model*
   *There are two major concerns about using the EMAC model as a reference state of the atmosphere. First, the model description in the text is insufficient. It needs to be mentioned how the model was validated against other independent observations. For which species did the model perform well and for which not? Where is the model insufficient to reproduce variability on the scale given by the model resolution? This is especially important since one may suspect that the model will have difficulties reproducing vertical trace gas gradients in the UTLS region. Second, as shown in Figure 1 the model has only 3 levels in the UTLS region and output was only available every 12-hour. Therefore, the model misses large parts of the real variability (see also the CARIBIC comparison). How can it be justified that the model can still be assessed to analyse representativeness?*

   We have included a new section (Sec. 5) that covers these questions. The influence of the small-scale variability on climatological mean values is shown and the variability of the model compared to that of the measurements.

2. *Sampling strategy*
   *Several choices seem to be arbitrary. I especially don't understand why the temporal domain is not sampled as a whole. Both sampling patterns RANDPATH and RANDLOC only sample 12 and 8 days per month, respectively. It would seem more appropriate to sample daily but on the other hand with a more realistic pattern that resembles that of the CARIBIC flights (i.e., on great arcs between major airports in the northern hemisphere, leaving out transpacific flights, since this region is never covered by CARIBIC). In that case the RANDPATH sampling could be viewed as the maximal achievable sampling pattern by commercial aircraft and RANDLOC could still be seen as sampling the northern hemispheric UTLS region as a whole.*

   The alternative approach for creating RANDPATH proposed here may be more realistic, but the results would not be much different. The same is true for sampling the temporal domain. This is probable, as even RANDPATH and RANDLOC are very similar, despite the differences in their sampling statistics.

3. *Selected statistical measures*
   *Again there seem to be arbitrary choices concerning the statistical estimators and tests. If the Komogorov-Smirnov test turned out to be too strict because it requires similarity of the whole distribution, why did you not select other statistical tests that only evaluate one statistical parameter at a time (e.g., Mann-Whitby test for the mean and Levene's or Brown–Forsythe test for variance, all are non-parametric tests suited for atmospheric trace gas observations). Furthermore, the results need to be discussed together with observed seasonality of the trace species as is mentioned by the authors themselves on page 17, line 1, but than dropped without further reasoning 3 lines later. The relative difference does not contain much information in itself and as stated correctly depends on the lifetime of a species.*

   We have checked all the tests proposed here and find that they are also not much more useful. Short comments on this are included in the text. In discussing results, we now more explicitly state the need to consider the seasonality and do so when considering single climatologies.

Minor Comments

1. *P1,L11: "formulated above". Not clear from the context where this was formulated*

   The sentence has been reformulated.

2. *P3,L28ff: Although no details on the measurement techniques are needed here, it would still be interesting to learn something about the overall uncertainties of the measurements and how these compare to the later discussion of representativeness.*

   It is difficult to give an overall uncertainty of the measurements, as these are taken by man different instruments. For example, the accuracy of acetone measurements is typically +-15%, which is mainly determined by the accuracy of the calibration gas standard and the reproducibility of the calibration. The relative precision becomes smaller for higher mixing ratios. At 1000 pptV, it is ~+- 3%, but it becomes +-25% at 200 pptV. For O3, the precision is in the order of 0.3-1%. Since the instruments have such different characteristics, we have decided not include this factor in the analysis.

3. *P4,L10: Model output every "eleven hours"? Did you mean 12 hours?*

   Model output for this model run was saved every eleven hours in order to be able to reproduce mean daily cycles.

4. *P4,L9ff: Additional information on emissions used in EMAC and vertical resolution in the UTLS region would be useful here.*

   The vertical resolution is displayed in Figure 1 and corresponds to about 1.5km in the UTLS.

5. *Section 3.1: It should be more prominently mentioned in the first paragraph of this section that you restrict the analysis to the latitude region 35N to 75N. Details follow towards the end of the section and can remain there, but it would be good to make this important detail clear from the beginning. It should also be stated in the abstract.*

   We now state these limits at the beginning of the paragraph as well as in the abstract.

6. *Table 1: For RANDPATH it is an adjusted Gaussian distribution, as mentioned in the text.*

   Wording has been adopted.

7. *Table 1 and elsewhere: "Uniform" or "rectangular" distribution should be used instead of "even".*

   Wording has been adopted.

8. *P6,L6f: The good correlation for temperature is not a big surprise, given the strong vertical stratification in the UTLS and the assumably large number of measurements. Since this is one of the few pieces of model validation mentioned, one could add a scatter plot to the supplement.*

   A validation of the model is not the focus of this study. It has been done elsewhere, e.g. by Hegglin et al. (JGR), 2010. The text now gives a reference to this study.

9.  *P6,L9f: It is not clear to me why the limited vertical model resolution is the reason you cannot compare CARIBIC directly to EMAC. The random sampling is still done using vertical interpolation to specific pressure levels. Would't the same argument apply to the random sampling strategy as well and could one not simply drop it and do the analysis of representativeness on discreet model levels instead?*

    Using discrete model levels would introduce a strong bias to the results The pressure would be limited to certain values only, which - in addition - are close to the limits CARIBIC ever reaches. This influences the distribution in HrelTP, introducing complicated differences to the climatologies.

10. *P6,L20f: Why did you choose these cut-off values instead of simply using the standard deviation as a criterion (i.e., redistribute values outside +/- 2 sigma). I don't assume this would change much, but would seem statistically more sound. Alternatively, one could have sampled directly from the observed CARIBIC distribution.*

    We used these cut-off values as these correspond to the upper and lower limit of the CARIBIC measurements. The lower boundary was set to exclude ascents and descents of the aircraft.

11. *P7,L7ff: I don't agree with the statement that the distribution "is very similar for all datasets". There is a strong offset to higher HrelTP in both random sampling strategies. What is the actual mean HrelTP for all three samples?*

    The mean of HrelTP for the different datasets is now stated in the text. In addition, we have revised our judgment on the differences. The reasons for the differences are the same as those that lead to differences in the climatologies.

12. *p7,L11f: This requires some further justification (see major comment above). Without being aware of the details of Jöckel et al 2015, it seems a bit hard to believe that the model performs equally well for the very different set of species analysed here. There should be additional discussion of the species for which this may not be justified.*

    We have included a new, separate section (Sec. 5) that covers this subject, comparing model and measurement variability.

13. *p9,l19: How was the mean tau\* calculated? As the mean over all monthly tau\* or as tau\* of the mean mu and sigma?*

    This is now clearly stated in the text: As $\tau^*$ of the mean of $\mu$ and $\sigma$ using the whole time series.

14. *Figure2: It would be interesting to add CARIBIC observed tau\* in the figure (where available).*

    Figure 2 now includes all $\tau^*$ of all the relevant datasets.

15. *Figure3 and others: The y-axis if often titled "variability". It would be useful to give a more concrete title, since the manuscript is dealing with all kinds of variability. This could reduce confusion. In this specific case I assume this is relative standard deviation?*

    The y-axis is now titled $\sigma_r$, the relative standard deviation.

16. *p12,l18: "The differences are small, mostly below an absolute value of 0.15." But this means that the absolute difference between both samples is 1.4 times larger than the value of the reference (or am I mistaken). I am not sure that I would call this small! In general using the log scaled relative difference seems a bit odd and only confuses. Why not use the relative difference as is?*

    The value 0.15 refers to the absolute values of which the logarithm has not been taken. 0.15 means there is 15% percent difference between the fields.

17. *p12,l29f: "A similar analysis has also been performed with data from a random number generator, leading to equivalent results." Are you referring to the RANDLOC sample here?*

    We are referring to the study of data created with a random number generator. It is documented in the supplement to the paper. The sentence has been reworded.

18. *p13,l13: At least repeat the result of the sensitivity study here. The supplement should not be a paper on its own.*

    The reference in the following sentence has been made clearer.

19. *p16,l5: Not clear which correlation is referred to here.*

    The reference to the correlation coefficient (with the number of samples) has been removed.

20. *p16,l6: What is an "uncertainty error"? I think the use of representativeness uncertainty would in general work better.*

    The wording has been adopted.

21. *p16,l8-13: This description is completely confusing. I don't understand what is done and why. Please improve the description.*

    The wording has been changed and hopefully improved.

22. *p17,l10ff: Since the discussion on NOx is along the EMAC results, it would be interesting to know how NOx sources in the UTLS are treated in the model. Does the model include a realistic representation of lightning NOx? Has this been analysed in previous studies?*

    NOx production resulting from lightning activity is calculated with the help of the EMAC submodel LNOX (Grewe et al., 2001).

23. *p17,l33f: The representativeness uncertainty of 5 [% for long-] lived trace gases is huge considering their atmospheric abundance. It is much larger than their seasonal variability. This aspect needs to be considered in the analysis and discussed along with the results.*

    We now more clearly state the importance of the seasonal cycle. 'Less than 5%' has been reworded to be more meaningful for these substances. In addition, the color scale of the corresponding figure has been changed.

24. *p18,l20ff: Finally there is some discussion using species specific thresholds, but again these thresholds are chosen without any justification. They should be related to seasonal variability.*

    The thresholds are now related to seasonal variability.

25. *p18,l32: Was it ever shown that R_rel "increases linearly"? May be an increasing relationship, but linearly?*

> A linear relationship has been shown to exist by the good Pearson correlation coefficient in Section 6.3.

26. *Figures: It would be easier to follow the discussion of the figures if sub-panels would be labelled by letters (which is Copernicus style). For example discussion of Figure 8 on page 18.*

> We have included labels to the panels of Figure 8.

27. *Figure 1 in supplement: Please explain black line in legend and add fit as additional line to the plot. Indicate which species are behind each point. Is the given fit applied to log(meas) and log(model)? Is it just my impression or does the model actually capture less of the variability for species that have a small relative variability? How could this be explained? I would have expected the opposite.*

> This part of the supplement has been removed as it is now covered in the new section on model and measurement variability (Sec. 5).

**Technical Comments**

1. *P1,L2: It is "representative of" not "representative for".*

> This has been changed where 'representative for' stands for representative of a population.

2. *P5,L6 and elsewhere: "Data" is always plural. Change to "Data were*

> This has been changed.

**Literature**

- Grewe, V., Brunner, D., Dameris, M., Grenfell, J. L., Hein, R., Shindell, D., and Staehelin, J.: Origin and variability of upper tropospheric nitrogen oxides and ozone at northern mid-latitudes, Atmos. Environ., 35, 3421–3433, 10.1016/S1352-2310(01)00134-0, 2001

- Hegglin, M. I., Gettelman, A., Hoor, P., Krichevsky, R., Manney, G. L., Pan, L. L., Son, S.-W., Stiller, G., Tilmes, S., Walker, K. A., Eyring, V., Shepherd, T. G., Waugh, D., Akiyoshi, H., Añel, J. A., Austin, J., Baumgaertner, A., Bekki, S., Braesicke, P., Brühl, C., Butchart, N., Chipperfield, M., Dameris, M., Dhomse, S., Frith, S., Garny, H., Hardiman, S. C., Jöckel, P., Kinnison, D. E., Lamarque, J. F., Mancini, E., Michou, M., Morgenstern, O., Nakamura, T., Olivié, D., Pawson, S., Pitari, G., Plummer, D. A., Pyle, J. A., Rozanov, E., Scinocca, J. F., Shibata, K., Smale, D., Teyssèdre, H., Tian, W., and Yamashita, Y.: Multimodel assessment of the upper troposphere and lower stratosphere: Extratropics, Journal of Geophysical Research: Atmospheres, 115, doi:10.1029/2010JD013884, 2010.

- Sander, R., Kerkweg, A., Jöckel, P., and Lelieveld, J.: Technical note: The new comprehensive atmospheric chemistry module MECCA, Atmos. Chem. Phys., 5, 445–450, doi:10.5194/acp-5-445-2005, 2005

[revised manuscript text omitted]
_{\mathrm{var}}^{t}$, decreasing values in $R_{\mathrm{rel}}$ mean better representativeness. And like $R_{\mathrm{var}}^{t}$, $R_{\mathrm{rel}}$ has to be tested for passing

[Figure]

**Figure 4.** Relative differences of CO for MOD MOD$_{\mathrm{CARIBIC}}^{\mathrm{regular}}$ and MOD$_{\mathrm{RANDPATH}}$. This is the basis used to calculate R$_{\mathrm{rel}}$.

the requirements of being related to number of samples and variability (see Sec. 3.3) in order to be acceptable as a score for representativeness. The results of testing this will be discussed in Sec. 6.3.

Other than just as a score, the value of R$_{\mathrm{rel}}$ can be understood as the average uncertainty for assuming the climatology of MOD MOD$_{\mathrm{CARIBIC}}^{\mathrm{regular}}$ as a full model climatology. This is more obvious if taken to the power of 10, in which case the uncertainty

5 will take values between 0 and 1. Use of this will be made in Section 6.4.

**5 Model and measurement variability**

Representativeness was assessed using only model data in this study, yet the final goal was to investigate the representativeness of MEAS$_{\mathrm{CARIBIC}}$. MOD$_{\mathrm{CARIBIC}}^{\mathrm{regular}}$ and MOD$_{\mathrm{CARIBIC}}^{\mathrm{sampled}}$ are used as a placeholder for MEAS$_{\mathrm{CARIBIC}}$ and compared to other model datasets (MOD$_{\mathrm{RANDPATH}}$ and MOD$_{\mathrm{RANDLOC}}$) in the analysis. The results derived from these model datasets will be interpreted

[revised manuscript text omitted]

**2 Calculating representativeness from random numbers**

All three methods to investigate representativeness (Kolmogorov-Smirnov test, variability analysis and relative differences) have also been applied to data created with a random number generator. The results of this study are  presented here.

To produce the random numbers, 20 sets of $10^8$ numbers were taken from a normal distribution. These 20 sets are referred to as species, well aware of the fact that they are purely artificial. From species to species, the standard deviation $\sigma$ was set to vary from $10^{-3}$ to $10^3$, values of the exponent  increasing linearly. 20 mean values $\mu$ (increasing from $10^4$ to $10^8$, with a linear increase in the exponent) where distributed randomly  to the 20 species. This results in 20 species with different values for $\sigma$ and $\mu$. The statistics of each species will be indexed by the number 2. For short, this dataset will be called RAND.

3000 samples were taken from each of the 20 species. For each sample, 20 numbers were first randomly drawn from each species. These new numbers and all those that had been drawn before then make up this one sample. So the size increases by 20 for each sample. This way, the relationship of the representativeness score with the sample size is directly accessible.  Samples are indexed by the number 1.

The variability $\tau^*$ of each species  is defined as in Equation  3 of the main text: $\tau^* = \log_{10}(\sigma_2/\mu_2)$. The two requirements set up in Section 3.3 for representativeness in general also have to hold here:

1. Representativeness has to increase with the number of samples.

2. Representativeness has to decrease with increasing variability of the underlying distribution.

With RAND defined in this way, it is possible to test representativeness using the variability analysis following Rohrer and Berresheim (2006) and Kunz et al. (2008) (see Section 4.2) and the relative differences (see Section 4.3). The Kolmogorov-Smirnov test was positive for very few samples (less than fifty numbers, independent of $\tau^*$) and will not be further discussed. Its behaviour with aircraft data was subject of a sensitivity study, the results of which are shown in Sec. 3 of this supplement.

**2.1 Variability analysis**

The  variability analysis (defined in Section 4.2 and Eq. 3) was applied in a simplified manner. As RAND is independent of time, $R_{var}$ is reduced to just a single value containing the absolute difference of variability of each species of RAND and the sample taken thereof: $R_{var} = |\nu_1 - \nu_2|$, where $\nu$ is the mean variability. Figure 1 shows a result. The exact result is a matter of chance, as a random number generator is used. Similar to using MOD$_{\text{CARIBIC}}^{\text{regular}}$ and MOD$_{\text{RANDPATH}}$, a strong dependance on $\tau^*$ and a weak dependance on the number of samples is visible.

Similar to $R_{var}$ when using MOD$_{\text{CARIBIC}}^{\text{regular}}$ and MOD$_{\text{RANDPATH}}$, the variability analysis using RAND meets the two requirements necessary for describing representativeness, which were described in Section 3.3 and above. This result supports the  finding that $R_{var}$ can be used as a statistic for describing representativeness.

[Figure]

**Figure 1.** Representativeness score R$_{var}$ applied to RAND. Vertical lines indicate the values of $\tau^*$ of each species.

**2.2 Relative differences**

Similar to R$_{var}$, R$_{rel}$ is reduced to a simple relative difference when using RAND: R$_{rel} = |\mu_1 - \mu_2|/\mu_2$, where $\mu$ is the mean of the sample (index 1) and of the whole subset (index 2). Figure 2 shows  a result when applying R$_{rel}$ to RAND. The dependance on $\tau^*$ is strong and linear. The result also depends on the number of samples, showing a slow increase with the

5    number of samples. This dependance is sometimes disturbed by better values which are reached by chance when drawing from RAND.

Like for MOD $_{\text{CARIBIC}}^{\text{regular}}$ and MOD$_{\text{RANDPATH}}$, R$_{rel}$ passes both conditions for a valid description of representativeness: it depends on variability $\tau^*$ and on the number of samples. The latter is also being influenced by chance and generally much weaker.

10    The fact that R$_{rel}$ passes the two conditions for a description of representativeness can be understood with some theoretical considerations. The standard error of the mean is defined by

$$\sigma_{\overline{x}} = \frac{\sigma}{\sqrt{n}} \tag{1}$$

where $\sigma_{\overline{x}}$, the standard deviation of a sample, can be given by the following equation ($N$ being the number of samples):

$$\sigma_{\overline{x}} = \sqrt{\frac{1}{N} \sum_{i=1}^{N} (\overline{x}_i - \mu)^2} \tag{2}$$

15    For $N = 1$, this gives:

$$\sigma_{\overline{x}} = |\overline{x}_i - \mu| \tag{3}$$

[Figure]

**Figure 2.** Like Figure 1, but for $R_{rel}$.

Plugging Eq. 3 into Eq. 1 gives:

$$\frac{|\overline{x}_i - \mu|}{\mu} = \frac{\sigma}{\mu\sqrt{n}} = \frac{10^{1/\tau^*}}{\sqrt{n}} \tag{4}$$

and therefore

$$R_{rel} = \log_{10}\left(\frac{|\overline{x} - \mu|}{\mu}\right) = -0.5\log_{10}(n) + \frac{1}{\tau^*} \tag{5}$$

5 So ideally, $R_{rel}$ should depend inversely on $\tau^*$ and directly on the logarithm of the number of values. Figure 2 shows this is approximately true for RAND.

In the case of RAND, $R_{rel}$  and $R_{var}$ can both be used to describe representativeness as  they pass the two conditions. Theoretical considerations make the finding plausible for $R_{rel}$. RAND can be considered a theoretical abstraction of MOD. The finding here therefore strongly supports that of Sections 5.2 and 5.3, where $R_{rel}$ and $R_{var}$

10 have also been found to be good descriptors of representativeness when using $MOD_{\overline{CARIBIC}}^{regular}$ and $MOD_{RANDPATH}$ or $MOD_{RANDLOC}$. In the main text, we use $R_{rel}$ for final results, as it more suitable to answer the question of representativeness for a climatology.

**3 Sensitivity study on the Kolmogorov-Smirnov test**

When  applying the Kolmogorov-Smirnov test to $MOD_{\overline{CARIBIC}}^{regular}$, $MOD_{RANDPATH}$ or $MOD_{RANDLOC}$,

15  it returned almost only negative results. This indicates that $MOD_{\overline{CARIBIC}}^{regular}$ 
[revised manuscript text omitted]
 $20\,\text{min}$ variability is therefore always larger in $\text{MEAS}_{\text{CARIBIC}}$ than in $\text{MOD}_{\text{CARIBIC}}^{\text{sampled}}$. To what extent this small scale variability influences the climatological values is investigated here.

10  By reducing the $20\,\text{min}$ variability in $\text{MEAS}_{\text{CARIBIC}}$ to that of $\text{MOD}_{\text{CARIBIC}}^{\text{sampled}}$, it is possible to determine the influence of the small scale variability on the climatological mean values. The reduction in variability was done separately for each species and height to account for differences in terms of model complexity between the species. In order to reduce the variability in the time series, they were smoothed out, the method is presented in App. B. The smoothing number used in this method indicates how much variability has been removed. The $20\,\text{min}$ variability of $\text{MEAS}_{\text{CARIBIC}}$ was then calculated for several smoothing
15  numbers.

Figure 5 (left panel, solid lines) shows how the $20\,\text{min}$ variability drops for all species if the data are smoothed progressively (increasing the smoothing number). The leftmost point for each species corresponds to the full $20\,\text{min}$ variability, while this variability drops to zero if the time intervals considered in smoothing become much longer than $20\,\text{min}$. The dashed lines show

[Figure]

[Figure]

**Figure 5.** Left panel: $20\,\mathrm{min}$ variability of i) $\mathrm{MEAS_{CARIBIC}}$, that has been smoothed out to an increasing degree, indicated by an increasing smoothing number (solid lines) and of ii) $\mathrm{MOD_{CARIBIC}^{sampled}}$ (dashed lines), both for $\mathrm{HrelTP} = -1\,\mathrm{km}$. The crosspoint of the dashed and corresponding full line indicate the smoothing number that is needed to reproduce the $20\,\mathrm{min}$ variability of $\mathrm{MOD_{CARIBIC}^{sampled}}$. Right panel: Mean relative differences of $\mathrm{MEAS_{CARIBIC}^{smoothed}}$ and $\mathrm{MEAS_{CARIBIC}}$. $\mathrm{MEAS_{CARIBIC}^{smoothed}}$ has been smoothed to have the same $20\,\mathrm{min}$ variability as $\mathrm{MOD_{CARIBIC}^{sampled}}$, using the smoothing number from the left hand panel. The relative differences correspond to the error in the climatologies of $\mathrm{MOD_{CARIBIC}^{sampled}}$ due to the coarse model resolution. $N_2O$, $C_2H_6$ and $C_3H_8$ are measured by air samples with a low measurement frequency and therefore not considered here.

the full model variability, which was not smoothed out. The crosspoints of the full and corresponding dashed line indicate the smoothing numbers for which $\mathrm{MEAS_{CARIBIC}}$ has the same $20\,\mathrm{min}$ variability as $\mathrm{MOD_{CARIBIC}^{sampled}}$. $\mathrm{MEAS_{CARIBIC}}$ in which each species has been smoothed to this point will be referred to as $\mathrm{MEAS_{CARIBIC}^{smoothed}}$.

Climatological mean values of $\mathrm{MEAS_{CARIBIC}^{smoothed}}$ were then compared to mean values from $\mathrm{MEAS_{CARIBIC}}$ with the full variabil-
5    ity, thereby determining the influence of the reduced $20\,\mathrm{min}$ variability. A similar influence is expected by the coarse model resolution, which by definition has the same $20\,\mathrm{min}$ variability as $\mathrm{MEAS_{CARIBIC}^{smoothed}}$.

[revised manuscript text omitted]
 to the 20 species. This results in 20 species with different values for $\sigma$ and $\mu$. The statistics of each species will be indexed by the number 2. For short, this dataset will be called RAND.

3000 samples were taken from each of the 20 species. For each sample, 20 numbers were first randomly drawn from each species. These new numbers and all those that had been drawn before then make up this one sample. So the size increases by 20 for each sample. This way, the relationship of the representativeness score with the sample size is directly accessible. Samples are indexed by the number 1.

The variability $\tau^*$ of each species is defined as in Equation 3 of the main text: $\tau^* = \log_{10}(\sigma_2/\mu_2)$. The two requirements set up in Section 3.3 for representativeness in general also have to hold here:

1. Representativeness has to increase with the number of samples.

2. Representativeness has to decrease with increasing variability of the underlying distribution.

With RAND defined in this way, it is possible to test representativeness using the variability analysis following Rohrer and Berresheim (2006) and Kunz et al. (2008) (see Section 4.2) and the relative differences (see Section 4.3). The Kolmogorov-Smirnov test was positive for very few samples (less than fifty numbers, independent of $\tau^*$) and will not be further discussed. Its behaviour with aircraft data was subject of a sensitivity study, the results of which are shown in Sec. 3 of this supplement.

[Figure]

**Figure 1.** Representativeness score $R_{var}$ applied to RAND. Vertical lines indicate the values of $\tau^*$ of each species.

**2.1 Variability analysis**

The variability analysis (defined in Section 4.2 and Eq. 3) was applied in a simplified manner. As RAND is independent of time, $R_{var}$ is reduced to just a single value containing the absolute difference of variability of each species of RAND and the sample taken thereof: $R_{var} = |\nu_1 - \nu_2|$, where $\nu$ is the mean variability. Figure 1 shows a result. The exact result is a matter of
5  chance, as a random number generator is used. Similar to using $MOD_{CARIBIC}^{regular}$ and $MOD_{RANDPATH}$, a strong dependance on $\tau^*$ and a weak dependance on the number of samples is visible.

Similar to $R_{var}$ when using $MOD_{CARIBIC}^{regular}$ and $MOD_{RANDPATH}$, the variability analysis using RAND meets the two requirements necessary for describing representativeness, which were described in Section 3.3 and above. This result supports the finding that $R_{var}$ can be used as a statistic for describing representativeness.

10  ## 2.2 Relative differences

Similar to $R_{var}$, $R_{rel}$ is reduced to a simple relative difference when using RAND: $R_{rel} = |\mu_1 - \mu_2|/\mu_2$, where $\mu$ is the mean of the sample (index 1) and of the whole subset (index 2). Figure 2 shows a result when applying $R_{rel}$ to RAND. The dependance on $\tau^*$ is strong and linear. The result also depends on the number of samples, showing a slow increase with the number of samples. This dependance is sometimes disturbed by better values which are reached by chance when drawing from RAND.
15  Like for $MOD_{CARIBIC}^{regular}$ and $MOD_{RANDPATH}$, $R_{rel}$ passes both conditions for a valid description of representativeness: it depends on variability $\tau^*$ and on the number of samples. The latter is also being influenced by chance and generally much weaker.

[Figure]

**Figure 2.** Like Figure 1, but for $R_{rel}$.

The fact that $R_{rel}$ passes the two conditions for a description of representativeness can be understood with some theoretical considerations. The standard error of the mean is defined by

$$\sigma_{\overline{x}} = \frac{\sigma}{\sqrt{n}} \tag{1}$$

where $\sigma_{\overline{x}}$, the standard deviation of a sample, can be given by the following equation ($N$ being the number of samples):

$$\sigma_{\overline{x}} = \sqrt{\frac{1}{N}\sum_{i=1}^{N}(\overline{x}_i - \mu)^2} \tag{2}$$

For $N = 1$, this gives:

$$\sigma_{\overline{x}} = |\overline{x}_i - \mu| \tag{3}$$

Plugging Eq. 3 into Eq. 1 gives:

$$\frac{|\overline{x}_i - \mu|}{\mu} = \frac{\sigma}{\mu\sqrt{n}} = \frac{10^{1/\tau^*}}{\sqrt{n}} \tag{4}$$

and therefore

$$R_{rel} = \log_{10}\left(\frac{|\overline{x} - \mu|}{\mu}\right) = -0.5\log_{10}(n) + \frac{1}{\tau^*} \tag{5}$$

So ideally, $R_{rel}$ should depend inversely on $\tau^*$ and directly on the logarithm of the number of values. Figure 2 shows this is approximately true for RAND.

[Figure]

**Figure 3.** Flightroutes to Vancoucer, Canada, where each flight has been cut into 20 pieces and randomly chosen 30% of those pieces have been plotted. These are tested against the whole data from flights to Vancouver to give one point in Figure 4.

In the case of RAND, $R_{rel}$ and $R_{var}$ can both be used to describe representativeness as they pass the two conditions. Theoretical considerations make the finding plausible for $R_{rel}$. RAND can be considered a theoretical abstraction of MOD. The finding here therefore strongly supports that of Sections 5.2 and 5.3, where $R_{rel}$ and $R_{var}$ have also been found to be good descriptors of representativeness when using $MOD_{CARIBIC}^{regular}$ and $MOD_{RANDPATH}$ or $MOD_{RANDLOC}$. In the main text, we use $R_{rel}$ for final
5   results, as it more suitable to answer the question of representativeness for a climatology.

**3   Sensitivity study on the Kolmogorov-Smirnov test**

When applying the Kolmogorov-Smirnov test to $MOD_{CARIBIC}^{regular}$, $MOD_{RANDPATH}$ or $MOD_{RANDLOC}$, it returned almost only negative results. This indicates that $MOD_{CARIBIC}^{regular}$ is not representative of $MOD_{RANDPATH}$ in the definition of the Kolmogorov-Smirnov test. This behaviour was tested in a sensitivity study, the results of which are discussed here.
10   One of the most frequent destinations within the CARIBIC project is Vancouver, Canada (near $120\,°W$, $45\,°N$, see Figure 3), and only the subset of $MOD_{CARIBIC}^{regular}$ to this destination is considered in this example to minimize effects that may come from different flight routes. Parts of this reduced dataset were tested with the Kolmogorov-Smirnov test against the whole reduced dataset for all variables. To produce these partial datasets, each flight was cut into an increasing number of pieces (corresponding to a certain time) and different percentages of these pieces were used in testing. Figure 3 shows an example of
15   applying this method, by cutting each flight into 20 pieces and taking 30% of these by showing the corresponding flightpaths.

[Figure]

**Figure 4.** The Kolmogorov-Smirnov test applied to the flights to Vancouver, Canada, of $\text{MOD}^{\text{regular}}_{\text{CARIBIC}}$ and subsets of these flights. Dotted lines indicate those lengths in time and those percentages that were tested. 0 stands for a passing the Kolmogorov-Smirnov test, 1 for not passing.

Data was not binned in months. When applying the Kolmogorov-Smirnov test without binning in months, the result is a profile in HrelTP for each variable. The result can then be diplayed in similar way to Figures 7 and 8. This matrix of height versus species was calculated for each combination of number of pieces and percent of pieces. In each combination, all the profiles of the different variables were averaged to end up with one value betwween 1 and 0 characterizing the result of the test

5   for this combination of number of pieces and percent of pieces. The result can then give an impression of the strictness of the Kolmogorov-Smirnov test.

Figure 4 shows the result of the study. Independent of the number of pieces, the result is positive if all pieces are considered, as the definition of the test prescribes. But only when removing short pieces (shorter than $20\,\text{min}$) is the result also positive for less pieces, even though 70% percent of the data is still needed. When removing whole flights (at the top of the plot), more the

10  90% of the data has to be taken into account to achieve a positive result of the Kolmogorov-Smirnov test. This result is very similar also for other error probabilities $\alpha$, taking values of 0.001, 0.01, 0.05 (in the figure), 0.1 and 0.2. The area of failing increases only slightly with the error probability. This showcases the strictness of the test. The Kolmogorov-Smirnov test does not seem suitable to test a dataset measured with aircraft for representativeness of a larger dataset.

---

## Author Response (AR3)

**Reply to Editor Comments**

Dear Prof. Engel,

thank you for editing our manuscript and your constructive feedback. In the first part of this document we reply to your latest comments regarding the revised version of the manuscript.

Previously we had submitted the comprehensive feedback to the reviewers comments. The second part of this document repeats the detailed answers to the reviewers comments and includes the corresponding revised manuscript.

In response to your comments, we have revised Sec. 5. The first, major comment (October, 2016) only lead to a change in the wording in the last paragraph. You had suggested to reformulate the conclusion that implies that the results from the model data can also be applied to the CARIBIC measurements.

Your second comment in December 2016 motivated a revision of the method we use in Sec. 5 to compare model and measurement variability. This touches the main question raised by you and the referees, which was 'how well the model represents true atmospheric variability as measured by CARIBIC'. We now present a simpler, more general method for smoothing the measurement data before comparing its variability on longer time scales to that of the model data. The more general method provides an easier assessment of variability. The results show that the fraction of measurement variability reached by the model data is larger than 50%  for all species. We believe this to be sufficient for a meaningful conclusion about the CARIBIC measurements in Sec. 6. More detailed information is presented below and in Sec. 5.

On the following pages, we repeat the point-by-point answers to the comments and have attached a marked-up version of Sec. 5 of the manuscript, which shows all the differences that came about by considering your feedback.

Thank you very much and best regards,

Johannes Eckstein

Major Revisions (21. Oct. 2016)

1. *I do not find the answers to the reviewers questions sufficiently convincing. The major point mentioned by both reviewers was the underlying assumption that the models variability is representative of the atmospheric variability which CARIBIC samples. The main conclusion reached is that after smoothing the data to yield a similar 20-min (short term) variability as the model, the long term variability in this smoothed data set is still about twice as large as the model variability. I do not think that this supports the statement that "The representativeness evaluated from the model data alone is also valid for the real atmosphere and the measurements taken by CARIBIC". Please be more careful with the conclusion drawn.*

    We have reworded the paragraph. Before it read:

    'As is shown in Figure 6, the model reaches more than 50 % of the variability of the measurements. This ratio depends strongly on the species and is higher for longer time scales. This points at a high complexity of the model and justifies the assumption underlying this study: The representativeness evaluated from the model data alone is also valid for the real atmosphere and the measurements taken by CARIBIC .'

    We have expanded the paragraph, coming to a more careful conclusion and including some thoughts on why we believe this conclusion to be valid. The paragraph now reads:

    'As is shown in Figure 6, the model reaches more than 50 % of the variability of the measurements, depending on the species and time scale. In general, the model variability can be increased by using a run with a higher resolution, because a decrease in spatial resolution requires a decrease in the time step of the integration. The variability of the measurements in each bin of HrelTP (height relative to the tropopause) is also influenced by the choice of reference for HrelTP. For this study, HrelTP has been derived from model output fields from ECMWF at a resolution of 1◦ (≈ 110 km), while the measurement data have a much higher resolution (≈ 2.5 km, see Sec. 2.1). The highly variable measurements are then sorted into bins of coarsely resolved HrelTP, artificially increasing the variability of the measurements in each bin of HrelTP. To a lesser extent, this also affects MEAS_smoothed. Considering these complementing thoughts on the model and measurement variability, the fraction of CARIBIC variability reached by the model (more than 50 %) justifies the application of the representativeness evaluated from the model to MEAS_CARIBIC. '

2. *Second, the answers to the reviewers are insufficient. Please explain for every comment in detail how you have reacted, e.g. not by saying "the wording has been adopted", but by explaining how it has been changed.*
   *Third, please explain the "numerous other changes" made to the manuscript in detail, i.e. every change (beyond wording) needs to be clearly shown and explained. This includes the inclusion of two new appendices, only one of them motivated by a reviewer as far as I could see.*

    We have expanded the reply to the referees. This expanded version forms the second part of this document. The summarizing paragraph at the beginning of the document now gives a detailed list of all changes that go beyond wording recommendations, i.e. what was before summarized as 'numerous other changes'. The separate answers to the comments of the referees have been expanded. We now always state the old and new wording to make it easier to follow the comments.

Minor Revisions (19. Dec. 2016)

1. *I have one remaining major issue, which is related to the model-measurements comparision of variability and in particular to the use of Meas_smooth_CARIBIC, which by defintion has the same variability as Mod_sampled_CARIBIC on the 20 min time scale. It is thus not an observed variability, but a model variability. Therefore the term used here is misleading and should be eliminated. It has no relation to a measured variability.*
*In my view, the original question posed by both reviewers, if the model can represent the observed variability is thus not answered. This would only be achieved if an objective way of smoothing would be used for the measurements (e.g. use of 20 min averages). Right now the new section 5 in my understanding compares model variability on longer time frames with model varaiblity on the grid scale of about 200 km (which actually corresponds more to 15 min of flight time than 20min). The question posed by both referees was, however, how well the model represents true atmospheric variability as measured by CARIBIC. This question is not answered.*

> By using a more general smoothing method, we are now able to answer this question in Sec. 5.2 of the manuscript. Sec. 5.1 investigates the influence of the smoothing method on climatological mean values.
> We now use the same smoothing algorithm for all variables and heights. The length of each part of the time series that is smoothed corresponds to 200km or 20min of flight, which is roughly the resolution of the model output grid. This is different to before, when we smoothed the measurements until the 20min variability of the model data had been reached.
> Sec. 5.1 shows that the model resolution as such does not prevent a realistic climatology. This is a prerequisite for the investigation of the model climatologies. Fig. 5 shows the impact of the smoothing algorithm to be small for the considered species. The error induced in the climatologies reaches more than 5% only where the species  show strong gradients, e.g. $H_2O$ in the lower stratosphere. The influence is smaller than with the old smoothing method. This is consistent, as the smoothing is also weaker now than before.
> Sec. 5.2 compares the variability of the model data with the variability of the smoothed CARIBIC measurements for two time scales (30 days and 1 year, Fig. 6). The model reaches 50% or more of the variability for all species for both time scales. The variability of short-lived species remains better represented by the model (reaching 100%) as the model chemistry increases the variability of these species. The measurement variability is artificially increased due to the data used for the height relative to tropopause (see answer to Major Revisions, comment no. 1 (21. October 2016)). The fraction of measurement variability reached by the model data is therefore sufficient for our line of argument.
> To further disentangle the causes of the model deficiencies that become apparent in Sec. 5.2 is beyond the scope of this study. The results discussed here, presented in more detail in in Sec. 5, answer the question of the reviewers concerning the model's representation of atmospheric variability. The model's representation of the real atmosphere is sufficient. It allows us to apply the climatological representativeness investigated from the model datasets also to the CARIBIC measurement.

[revised manuscript text omitted]

**Figure 5.** Mean relative differences of $MEAS_{CARIBIC}^{smoothed}$ and $MEAS_{CARIBIC}$. $MEAS_{CARIBIC}^{smoothed}$ has been smoothed by interpolating between the 20 min mean values, the exact method being presented in App. **??**. The relative differences correspond to the error in the climatologies of $MOD_{CARIBIC}^{sampled}$ due to the coarse model resolution. $N_2O$, $C_2H_6$ and $C_3H_8$ are measured by air samples with a low measurement frequency and therefore not considered here.

errors in climatological mean values. Nevertheless, the model could have other deficiencies in the description of the different species. These are made visible in the following section by comparing model and measurement variability directly.

**5.2 Comparing model and measurement variability**

In this section, the variability of $MOD_{CARIBIC}^{sampled}$ is compared directly to that of $MEAS_{CARIBIC}^{smoothed}$. For this dataset,

[revised manuscript text omitted]

**Reply to Referee Comments**

Dear Editor, dear Referees,

thank you for your comments and remarks to the manuscript. We have considered the constructive criticism and feel that we have managed to improve the manuscript with your feedback.

We now devote a new section (now Sec. 5) to the comparison of the model and measurements. We compare the variability in the two datasets in order to justify the conclusion of the study, in which we apply the results from a model study to measurements. A newly added appendix (Sec. B) explains the method applied in Sec. 5.

The appendix contains two more sections (Sec. A1 and A2). One describes the influence of using a regional limit on the results, implemented as a limit in longitude corresponding to the Pacific Ocean (Sec. A1). In addition, the section on the influence of the pressure limit, which was previously part of the supplement, has been moved to the appendix (Sec. A2), as it is in structure very similar to Sec. A1.

We no longer include $NO_2$, as there are very few measurements.

Of course, the larger changes lead to numerous smaller changes in the manuscript, which are included in addition to those motivated directly by your comments. All changes are summarized here in order of appearance:

Abstract and introduction have been adapted due to the changes in the main body text, now also highlighting the new Sec. 5.

The description of the measurement frequency in Sec. 2.1 has been extended. In addition, the sentence describing the detrending of long-lived trace gases has been moved here, to make more clear that this method is applied to both, measurements and model data.

The description of the model run in Sec. 2.2 has been expanded, following the recommendations of both referees to include information on the boundary conditions and validation of the model.

In Sec. 3.1 (Representative for what parameter?), the limit in latitude of our analysis is now clearly defined at the beginning, as recommended by Referee 1. The appendices covering the limits in pressure and longitude are referenced.

The next Sec. 3.2 (Representative of which population?) describes the datasets used in the study. $MOD_{CARIBIC}$ has been renamed to $MOD^{regular}_{CARIBIC}$ to differentiate it from $MOD^{sampled}_{CARIBIC}$, thereby clearly marking the difference between the synthetic and flight path sampled data. $MOD^{sampled}_{CARIBIC}$ is now more often used, especially in the new Sec. 5 discussing model and measurement variability.

We now also include $MOD^3_{RANDPATH}$, which had so far been explained in all sections that use the dataset of reorganized random flights created from $MOD_{RANDPATH}$.

The paragraphs discussing the variability of $MOD_{CARIBIC}$ have been removed as Sec. 5 now covers this subject in a much more detailed manner.

As Sec. 5 discusses variability in detail, we also reconsidered Sec. 3.4, which discusses the variability used to sort results and test the statistical measures $R_{rel}$ and $R_{var}$. Figure 2 now shows relative standard deviation $\sigma_r$ (instead of its logarithm $\tau*$) of all datasets, not just of $MOD_{RANDPATH}$, as these datasets are also used in the new Sec. 5. This was also requested by Referee 2.

Sec. 4.1 mentions the other statistical tests which are now included in the study, as recommended by Referee 2. After the new Sec. 5, Sec. 6.1 describes the results of these additional tests.

The performance of the score using variability ($R_{var}$, results presented in Sec. 6.2) has been tested with $MOD^3_{RANDPATH}$, this is an extension of the previously submitted manuscript not directly motivated by the referees' comments. Both, Sec. 6.2 and 6.3 have been partly reworded, using the definition of $MOD^3_{RANDPATH}$ introduced in Sec. 3.2.

Changes in Sec. 6.4 (Representativeness uncertainty of the CARIBIC measurement data) come about by the changes in the names of datasets and the new material presented in Sec. 5. Referee 2 pointed out the importance of linking the thresholds for representativeness with the seasonality, which we now include here. This is highlighted especially in the discussion of separate climatologies displayed in Figure 10. Following a suggestion of Referee 2, the panels of Figure 10 are now labeled with letters from A to F.

Because $MOD^3_{RANDPATH}$ has been introduced above, Sec. 6.5 (Number of flights for representativeness) is now much shorter. In addition, we use a different figure that directly shows the relationship between the representativeness uncertainty and the number of flights for several species. It is therefore better suited to link a species' uncertainty to its seasonality, as suggested by Referee 2.

Due to the numerous changes of the manuscript, the conclusions have been adopted accordingly.

In the following, we present a point by point reply to all suggestions, followed by the marked up manuscript.

Best regards,

Johannes Eckstein

**Referee 1**

Major Comments

1. *However, the question arises if the model can be used as an appropriate tool for the question. I think this question has not been addressed sufficiently in the paper. How well can data from a course model resolution be representative of the state of the atmosphere as described here? The representation of the model climatology vs. flight track interpolation should depend on the models spatial and temporal resolution. If the grid or time span is too large (likely the case for global models), the model would not be able to represent the variability of the observations. A test would require to average the observations to the same model grid and then compare the variability.*

   We have now included a separate section (Sec. 5) that treats this question. We show the influence of the small scale variability on climatological mean values and discuss the differences between model and measurement variability on longer time scales. The section shows that the model reaches 50% to 100% of the variability of the measurement data, which have been smoothed to have the same small-scale variability as the model data. The ratio could be increased by a model run with a higher resolution, but is just as much influenced by the data used for binning the measurements, which has a much coarser resolution than the measurements themselves.

2. *Furthermore, I do not see any evaluation of the model. How well does the model represent the atmosphere? Especially water vapor is a gas that many models are not able to simulate appropriately, which is also the case for NOx and NOy. A discussion on how much this study depends on the performance of the model to represent chemical tracer should be added.*

   The new Sec. 5 also covers the differences between species. A detailed validation is beyond the scope of this study. Sec. 2.2, describing the model, has been expanded to include more references, e.g. to Hegglin et al. (JGR), 2010, who describe a validation of some aspects of the model, in addition to the validataion published by Jöckel et al. (GMD), 2016.

3. *Finally, little has been done to identify reasons for differences between the flight track comparison and the global comparison, based on the atmospheric character of different trace gases dependent on the region for instance. Depending on region, airmasses experience more pollution, convection, stratosphere/troposphere exchange. The Pacific experiences a lot of pollution from South East Asia in some seasons than the Atlantic. Since CARIBIC data do not cover the Pacific, what implication does that have of the representation of the data compared to a global average? I would suggest, plotting a lon/lat map for a certain altitude level, say 1 km below the tropopause. This may help explain why some tracers are representative and why others may be not. Certainty 35-70 degrees is a very large region that covers a lot of different airmasses reaching from the tropics to the polar regions.*

   We have included a section in the appendix that assesses the influence of a regional limit by excluding data taken at longitudes corresponding to the Pacific Ocean and comparing the results to the regular analysis (Sec. A1). The influence on climatological mean values is stronger for those species determined by source regions in Asia.

Minor Comments

1. *Page 3, Line 9. The assumption that species in the model show a similar variability has not been supported. A climatology of trace gases from the course model resolution is expected to show a much smaller variability than the observations. Wouldn't you expect a different result if you would run with a high model resolution spatial and temporal?*

   We now include a new section (Sec. 5) which treats the differences in model and measurement variability and the influence of the small scale processes on the climatological mean values, see also the answer to major comment 1 above.

2. *Page 4, line 15: Why is N2O5 counted twice, please explain.*

   $N_2O_5$ is measured by catalytic conversion to NO. One $N_2O_5$ molecule yields two NO molecules, this is why every N has to be counted. This is now explained in the manuscript in Sec. 2.2.

3. *Page 5: Line 6: is it +-4km (as stated above) or +- 4.25km?*

   This has been corrected here to +-4.25km. Heights are labeled with their centers, which corresponds to +-4km.

4. *Page 5, Line 17ff: Constraining the data to 35-75 degree N is not really removing different characteristics of tropical or polar airmasses and you would expect a larger variability. Earlier studies discussed differences in the characteristics of UTLS airmasses depending on the location with the jet stream and therefore with the height of the tropopause, which strongly varies with season. I think, constraining the comparison to 35-75 degrees N because of a good coverage of aircraft data would the better argument. There should be some discussion on the variability of the considered region.*

   We now discuss the questions of regional limits and coverage in more detail. The good coverage was also an argument for the limit in latitude and we now state so clearly in the text. The latitudinal limit is for sure not sufficient to exclude all influence of higher or lower latitudes, but is a first approximation. We do discuss data relative to the local tropopause, as all fields are presented with HrelTP (height relative to the tropopause) as vertical axis.

5. *Page 5, Line 23, if you define mid-latitude as 35-75deg, then please specify that here.*

   We now define mid-latitude more clearly, e.g. by specifying 'We consider monthly binned data in the height of ±4.25 km around the dynamical tropopause defined at the pressure at 3.5 PVU and in mid-latitudes with 75 °N < $\varphi$ < 35 °N.' in the first paragraph of Sec. 3.

6. *Page 6: Line 6-7: The temperature comparison for the data is taken from meteorological analysis. Are those the same that were used to nudge the model? That would explain the high correlation coefficient. Please clarify.*

   The temperature data for the statement is taken from CARIBIC aircraft measurements. This is used for ERA-Interim via the AMDAR network, which is used for nudging the model. So the two are not independent, but the high Pearson correlation coefficient mentioned here serves to indicate the usefulness of the interpolation and is not meant as validation of the model.

7. *Page 7, Line 7-8: HrelTP does not look very similar to me. Distributions in the lower two rows in Figure 1 are more often above the TP than the flight track interpolated data. What implications will this have for the analysis?*

> Both, the distribution and differences in HrelTP and the different trace gas climatologies, are influenced by the sampling pattern. We have now clarified this relationship in the text.

8. *Page 7: Line 18. The text describes that the variability of the model data if interpolated to the flight track is only 40-70% of the actual observed data. Further, it is discussed that the variability in the model cannot capture the small scale variability of the data. Then the assumption is made that the variability of the model is similar for all species. I do not follow this conclusion. Why is this the case?*

> This paragraph has been completely revised and Sec. 5 now covers this subject.

9. *Page 9: Line 19: How does the model represent CO2, N2O and CH4? If those are prescribed as fixed boundary conditions, certainly the model would not identify the variability that exists in the real data.*

> Boundary conditions are not fixed. For $CO_2$, $N_2O$ and $CH_4$, they are prescribed as latitude dependent monthly means. We have included a paragraph in the Sec. 2.2 on the boundary conditions of the chemical species in the model run.

10. *Page 13: I am not surprised about the different characteristics, since the different coverage of CARIBIC compared to the random distribution is very different, Figure 1 left column, the flight track sample more tropical air masses (being more concentrated in the south). Furthermore, the Pacific with different characteristic of tracers are not sampled by the CARIBIC data set. It would help to see for example a figure of CO at the altitude considered for example 1 km below the tropopause. A discussion on differences of the sampling location due to chemical characteristics that are different depending on sampling tropical or polar air masses, or characteristic longitudinal variability in different tracers would be helpful.*

> Whether the climatologies produced by the sampling pattern of CARIBIC are representative is just the question that we are investigating in this study. Regional differences are another, interesting subject, but are more difficult to investigate with CARIBIC data. As a first step, the influence of considering the Pacific ocean (or not) is now included as Sec. A2 of the appendix of the paper.

11. *Page 17: typo line 2 "while it is can be much"*

> The typo has been corrected by removing 'can be'.

12. *Page 17: Line 10: models usually have a poor representation of NO and NO2, especially in the UTLS it depends on lightning. Also convection is influencing NOx and can strongly vary with location, which is usually not well represented in models. Couldn't this be the reason why there is a larger uncertainty?*

> $NO_x$ production resulting from lightning activity is included in the model (Grewe et al., 2001). The geographical constraint of CARIBIC flight routes to flight corridors and thereby to the regions with high VMR of $NO_x$ has the stronger influence on representativeness.

13. *Line 14: How is the model representing H2O in the stratosphere?*

    In the stratosphere and mesosphere the chemical $H_2O$ tendencies (due to the methane oxidation) are calculated with the help of the chemical submodel MECCA (Sander et al., 2005). $H_2O$ in the lower stratosphere is one of the compounds discussed by Hegglin et al, (JGR), 2010.

14. *Line 20; C2H6 and C3H8 are considered short-lived species with lifetimes of a few weeks or so.*

    We have changed the description from 'rather long-lived' to 'moderately long-lived'.

15. *Sec. 5.5 I think, the question should be changes for extended to: What would be a better regional coverage improve the statistic? This could be easily addressed within this paper, since one could extend the coverage over the pacific region, but keep the number of flights the same.*

    The influence of the Pacific region on our analysis is now covered in the appendix (Sec. A1). A more detailed study of the influence of different regions could be performed in the future.

16. *Conclusions: Page 21: Line 14: Sentence is unclear.*

    The sentence has been reworded: From '$R_{rel}$ is more applicable for answering the question, asking for the representativeness of for a climatology. It is therefore used for the analysis.' to '$R_{rel}$ (describing the representativeness of a climatology) is better suited for answering the question and is therefore used in the remaining analysis. '.

**Referee 2**

Major Comments

1. *Global scale chemistry transport model*
   *There are two major concerns about using the EMAC model as a reference state of the atmosphere. First, the model description in the text is insufficient. It needs to be mentioned how the model was validated against other independent observations. For which species did the model perform well and for which not? Where is the model insufficient to reproduce variability on the scale given by the model resolution? This is especially important since one may suspect that the model will have difficulties reproducing vertical trace gas gradients in the UTLS region. Second, as shown in Figure 1 the model has only 3 levels in the UTLS region and output was only available every 12-hour. Therefore, the model misses large parts of the real variability (see also the CARIBIC comparison). How can it be justified that the model can still be assessed to analyse representativeness?*

   We have now included a separate section (Sec. 5) that treats this question. We show the influence of the small scale variability on climatological mean values and discuss the differences between model and measurement variability on longer time scales. The section shows that the model reaches 50% to 100% of the measurement data that have been smoothed to have the same small-scale variability as the measurements. The ratio could be increased by a model run with a higher resolution, but is just as much influenced by the data used for binning the measurements, which has a much coarser resolution than the measurements themselves.

2. *Sampling strategy*
   *Several choices seem to be arbitrary. I especially don't understand why the temporal domain is not sampled as a whole. Both sampling patterns RANDPATH and RANDLOC only sample 12 and 8 days per month, respectively. It would seem more appropriate to sample daily but on the other hand with a more realistic pattern that resembles that of the CARIBIC flights (i.e., on great arcs between major airports in the northern hemisphere, leaving out transpacific flights, since this region is never covered by CARIBIC). In that case the RANDPATH sampling could be viewed as the maximal achievable sampling pattern by commercial aircraft and RANDLOC could still be seen as sampling the northern hemispheric UTLS region as a whole.*

   The alternative approach for creating RANDPATH proposed here may be more realistic, but the results would not be much different. The same is true for sampling the temporal domain. This is probable, as even RANDPATH and RANDLOC yield very similar climatologies, despite the differences in their sampling statistics.

3. _Selected statistical measures_
   _Again there seem to be arbitrary choices concerning the statistical estimators and tests. If the Komogorov-Smirnov test turned out to be too strict because it requires similarity of the whole distribution, why did you not select other statistical tests that only evaluate one statistical parameter at a time (e.g., Mann-Whitby test for the mean and Levene's or Brown–Forsythe test for variance, all are non-parametric tests suited for atmospheric trace gas observations). Furthermore, the results need to be discussed together with observed seasonality of the trace species as is mentioned by the authors themselves on page 17, line 1, but than dropped without further reasoning 3 lines later. The relative difference does not contain much information in itself and as stated correctly depends on the lifetime of a species._

   We have checked all the tests proposed here and find that they do not provide more detailed answers than the Kolmogorov-Smirnov test used for the original manuscript. Short comments on this are included in the text. In discussing results, we now more explicitly state the need to consider the seasonality and do so when considering climatologies of individual trace gases in Figure 10, formerly Figure 8.

**Minor Comments**

1. _P1,L11: "formulated above". Not clear from the context where this was formulated_

   The sentence has been reformulated: From 'In contrast, the variability based scores pass the general requirements for representativeness formulated above. ' to 'In contrast, the two scores based on either variability or relative differences show the expected behaviour and thus appear applicable for investigating representativeness. '.

2. _P3,L28ff: Although no details on the measurement techniques are needed here, it would still be interesting to learn something about the overall uncertainties of the measurements and how these compare to the later discussion of representativeness._

   It is difficult to give an overall uncertainty of the measurements, as these are taken by many different instruments. For example, the accuracy of acetone measurements is typically +-15%, which is mainly determined by the accuracy of the calibration gas standard and the reproducibility of the calibration. The relative precision becomes smaller for higher mixing ratios. At 1000 pptV, it is ~+- 3%, but it becomes +-25% at 200 pptV. For $O_3$, the precision is in the order of 0.3-1%. Since the instruments have such different characteristics, we have decided not to include too much detail on this in the manuscript.

3. _P4,L10: Model output every "eleven hours"? Did you mean 12 hours?_

   Model output for this model run was saved every eleven hours in order to be able to reproduce mean daily cycles.

4. _P4,L9ff: Additional information on emissions used in EMAC and vertical resolution in the UTLS region would be useful here._

   The vertical resolution is displayed in Figure 1 and corresponds to about 1.5km in the UTLS.

5. _Sec. 3.1: It should be more prominently mentioned in the first paragraph of this Sec. that you restrict the analysis to the latitude region 35N to 75N. Details follow towards the end of the Sec. and can remain there, but it would be good to make this important detail clear from the beginning. It should also be stated in the abstract._

   We now state these limits at the beginning of Sec, 3.1 as well as in the abstract.

6. *Table 1: For RANDPATH it is an adjusted Gaussian distribution, as mentioned in the text.*

   Wording has been adopted, changing 'gaussian' to 'adjusted gaussian' in the Table 1.

7. *Table 1 and elsewhere: "Uniform" or "rectangular" distribution should be used instead of "even".*

   Wording has been adopted, changing 'even' to 'uniform' in Table 1 and the description in Sec. 3.2.

8. *P6,L6f: The good correlation for temperature is not a big surprise, given the strong vertical stratification in the UTLS and the assumably large number of measurements. Since this is one of the few pieces of model validation mentioned, one could add a scatter plot to the supplement.*

   A validation of the model is not the focus of this study. It has been done elsewhere, e.g. Hegglin et al. (JGR), 2010 discuss a validation of some aspects of the model. The text now gives a reference to this study in Sec. 2.2.

9. *P6,L9f: It is not clear to me why the limited vertical model resolution is the reason you cannot compare CARIBIC directly to EMAC. The random sampling is still done using vertical interpolation to specific pressure levels. Would't the same argument apply to the random sampling strategy as well and could one not simply drop it and do the analysis of representativeness on discreet model levels instead?*

   Sampling on the discrete pressure levels would give rather poor results. The pressure would be limited to certain values only, which - in addition - are close to the limits CARIBIC ever reaches.

10. *P6,L20f: Why did you choose these cut-off values instead of simply using the standard deviation as a criterion (i.e., redistribute values outside +/- 2 sigma). I don't assume this would change much, but would seem statistically more sound. Alternatively, one could have sampled directly from the observed CARIBIC distribution.*

    We used these cut-off values as these correspond to the upper and lower limit of the CARIBIC measurements. The lower boundary was set to exclude ascents and descents of the aircraft.

11. *P7,L7ff: I don't agree with the statement that the distribution "is very similar for all datasets". There is a strong offset to higher HrelTP in both random sampling strategies. What is the actual mean HrelTP for all three samples?*

    The mean of HrelTP for the different datasets is now stated in Sec. 3.2. Both, the distribution and differences in HrelTP and the different trace gas climatologies, are influenced by the sampling pattern. We have now clarified this relationship in the text.

12. *p7,L11f: This requires some further justification (see major comment above). Without being aware of the details of Jöckel et al 2015, it seems a bit hard to believe that the model performs equally well for the very different set of species analysed here. There should be additional discussion of the species for which this may not be justified.*

    We have included a new, separate section (Sec. 5) that covers this subject, comparing model and measurement variability. The paragraphs you are referring to have therefore been removed.

13. *p9,l19: How was the mean tau\* calculated? As the mean over all monthly tau\* or as tau\* of the mean mu and sigma?*

> This is now clearly stated in the text: As $\tau^*$ (logarithm of the relative standard deviation) of the mean of $\mu$ and standard deviation $\sigma$ using the whole time series.

14. *Figure2: It would be interesting to add CARIBIC observed tau\* in the figure (where available).*

> Figure 2 now includes all $\tau^*$ (logarithm of the relative standard deviation) of all the relevant datasets. In addition, we have modified the figure to show relative standard deviation, $\sigma_r$, which makes the figure easier to understand.

15. *Figure3 and others: The y-axis if often titled "variability". It would be useful to give a more concrete title, since the manuscript is dealing with all kinds of variability. This could reduce confusion. In this specific case I assume this is relative standard deviation?*

> The y-axis is now titled $\sigma_r$, the relative standard deviation.

16. *p12,l18: "The differences are small, mostly below an absolute value of 0.15." But this means that the absolute difference between both samples is 1.4 times larger than the value of the reference (or am I mistaken). I am not sure that I would call this small! In general using the log scaled relative difference seems a bit odd and only confuses. Why not use the relative difference as is?*

> The value 0.15 refers to the absolute values of which the logarithm has not been taken. 0.15 means there is 15% percent difference between the fields.

17. *p12,l29f: "A similar analysis has also been performed with data from a random number generator, leading to equivalent results." Are you referring to the RANDLOC sample here?*

> We are referring to the study of data created with a random number generator. It is documented in the supplement to the paper. The sentence has been reworded. It now states: 'These methods have also been applied to data not from an atmospheric model but from a random number generator, leading to equivalent results. These are presented as supplementary material to the article.'

18. *p13,l13: At least repeat the result of the sensitivity study here. The supplement should not be a paper on its own.*

> The reference in the following sentence has been made clearer by stating: 'This is also the result of a sensitivity study, which is discussed as supplementary material to this text. '

19. *p16,l5: Not clear which correlation is referred to here.*

> The reference to the correlation coefficient (with the number of samples) has been removed as it is not necessary for the argument.

20. *p16,l6: What is an "uncertainty error"? I think the use of representativeness uncertainty would in general work better.*

> The wording has been adopted, changing 'uncertainty error' to a plain 'uncertainty' here, then introducing the term 'representativeness uncertainty' in the sentence.

21. *p16,l8-13: This description is completely confusing. I don't understand what is done and why. Please improve the description.*

> The wording of this paragraph has been changed, please also refer to the highlighted differences at the end of this document. The paragraph now reads:
> 'In order to asses the uncertainty for accepting CARIBIC measurement data to create a climatology, model data have to contain the same amount of data as $MEAS_{CARIBIC}$ , which is why $MOD^{sampled}_{CARIBIC}$ (see Sec. 2) will be used in the following. In addition, $MOD_{RANDLOC}$ (see Table 1) was used as reference, as it has a random sampling pattern and represents the full model state, independent of the sampling pressure. The limits in pressure where again set to 180 hPa < p < 280 hPa. The resulting $R_{rel}$ is shown in Figure 9. Using different wording, $R_{rel}$ in this formulation can also be considered the sampling error of the measurements. '

22. *p17,l10ff: Since the discussion on NOx is along the EMAC results, it would be interesting to know how NOx sources in the UTLS are treated in the model. Does the model include a realistic representation of lightning NOx? Has this been analysed in previous studies?*

> $NO_x$ production resulting from lightning activity is calculated with the help of the EMAC submodel LNOX (Grewe et al., 2001).

23. *p17,l33f: The representativeness uncertainty of 5 [% for long-] lived trace gases is huge considering their atmospheric abundance. It is much larger than their seasonal variability. This aspect needs to be considered in the analysis and discussed along with the results.*

> We now more clearly state the importance of the seasonal cycle. 'Less than 5%' has been reworded to be more meaningful for these substances by stating: '…all have representativeness uncertainties of less than 0.4 %, which is lower than their seasonal variability.' In addition, the color scale of the corresponding figure has been changed to highlight lower values more prominently.

24. *p18,l20ff: Finally there is some discussion using species specific thresholds, but again these thresholds are chosen without any justification. They should be related to seasonal variability.*

> The thresholds are now related to seasonal variability: 0.03% for $CO_2$ and 15% for $O_3$, one third and one fourth of the seasonal variability, respectively.

25. *p18,l32: Was it ever shown that R_rel "increases linearly"? May be an increasing relationship, but linearly?*

> A linear relationship has been shown to exist by the significant Pearson correlation coefficient in Sec. 6.3.

26. *Figures: It would be easier to follow the discussion of the figures if sub-panels would be labelled by letters (which is Copernicus style). For example discussion of Figure 8 on page 18.*

> We have included labels from A to F to the panels of Figure 10, formerly Figure 8.

27. *Figure 1 in supplement: Please explain black line in legend and add fit as additional line to the plot. Indicate which species are behind each point. Is the given fit applied to log(meas) and log(model)? Is it just my impression or does the model actually capture less of the variability for species that have a small relative variability? How could this be explained? I would have expected the opposite.*

This part of the supplement has been removed as it is now covered in the new Sec. 5 on model and measurement variability.

**Technical Comments**

1. *P1,L2: It is "representative of" not "representative for".*

This has been changed where 'representative for' stands for representative of a population.

2. *P5,L6 and elsewhere: "Data" is always plural. Change to "Data were*

This has been changed everywhere in the manuscript.

**Literature**

- Grewe, V., Brunner, D., Dameris, M., Grenfell, J. L., Hein, R., Shindell, D., and Staehelin, J.: Origin and variability of upper tropospheric nitrogen oxides and ozone at northern mid-latitudes, Atmos. Environ., 35, 3421–3433, 10.1016/S1352-2310(01)00134-0, 2001

- Hegglin, M. I., Gettelman, A., Hoor, P., Krichevsky, R., Manney, G. L., Pan, L. L., Son, S.-W., Stiller, G., Tilmes, S., Walker, K. A., Eyring, V., Shepherd, T. G., Waugh, D., Akiyoshi, H., Añel, J. A., Austin, J., Baumgaertner, A., Bekki, S., Braesicke, P., Brühl, C., Butchart, N., Chipperfield, M., Dameris, M., Dhomse, S., Frith, S., Garny, H., Hardiman, S. C., Jöckel, P., Kinnison, D. E., Lamarque, J. F., Mancini, E., Michou, M., Morgenstern, O., Nakamura, T., Olivié, D., Pawson, S., Pitari, G., Plummer, D. A., Pyle, J. A., Rozanov, E., Scinocca, J. F., Shibata, K., Smale, D., Teyssèdre, H., Tian, W., and Yamashita, Y.: Multimodel assessment of the upper troposphere and lower stratosphere: Extratropics, Journal of Geophysical Research: Atmospheres, 115, doi:10.1029/2010JD013884, 2010.

- Jöckel, P., Tost, H., Pozzer, A., Kunze, M., Kirner, O., Brenninkmeijer, C. A. M., Brinkop, S., Cai, D. S., Dyroff, C., Eckstein, J., Frank, F., Garny, H., Gottschaldt, K.-D., Graf, P., Grewe, V., Kerkweg, A., Kern, B., Matthes, S., Mertens, M., Meul, S., Neumaier, M., Nützel, M., Oberländer-Hayn, S., Ruhnke, R., Runde, T., Sander, R., Scharffe, D., and Zahn, A.: Earth System Chemistry integrated Modelling (ESCiMo) with the Modular Earth Submodel System (MESSy) version 2.51, Geoscientific Model Development, 9, 1153–1200, doi:10.5194/gmd-9-1153-2016, 2016.

- Sander, R., Kerkweg, A., Jöckel, P., and Lelieveld, J.: Technical note: The new comprehensive atmospheric chemistry module MECCA, Atmos. Chem. Phys., 5, 445–450, doi:10.5194/acp-5-445-2005, 2005

[revised manuscript text omitted]
_{CARIBIC}^{sampled}}$. Right panel: Mean relative differences of $\mathrm{MEAS_{CARIBIC}^{smoothed}}$ and $\mathrm{MEAS_{CARIBIC}}$. $\mathrm{MEAS_{CARIBIC}^{smoothed}}$ has been smoothed to have the same $20\,\mathrm{min}$ variability as $\mathrm{MOD_{CARIBIC}^{sampled}}$, using the smoothing number from the left hand panel. The relative differences correspond to the error in the climatologies of $\mathrm{MOD_{CARIBIC}^{sampled}}$ due to the coarse model resolution. $N_2O$, $C_2H_6$ and $C_3H_8$ are measured by air samples with a low measurement frequency and therefore not considered here.

model could have other deficiencies in the description of the different species. These are made visible in the following section by comparing model and measurement variability directly.

**5.2 Comparing model and measurement variability**

In this section, the variability of $\mathrm{MOD_{CARIBIC}^{sampled}}$ is compared directly to that of $\mathrm{MEAS_{CARIBIC}^{smoothed}}$. For this dataset, $\mathrm{MEAS_{CARIBIC}}$
5  has been altered in such a way to reproduce the $20\,\mathrm{min}$ variability of $\mathrm{MOD_{CARIBIC}^{sampled}}$, see the preceeding section. As this study argues completely within the model world, it is important that the model has similar values for the variability, which is used as an indicator of the underlying complexity. If the model cannot reproduce the measurement variability at all, it is not plausible why conclusions on representativeness drawn from model data should also be true for the real atmosphere.

As has been discussed in Sec. 4.2, variability depends on the time scale for which it is considered. In order to evaluate the
10  model performance, we compare $\sigma_r$ on time scales of $30\,\mathrm{d}$ and $1\,\mathrm{a}$. $30\,\mathrm{d}$ variability includes data from typically 4 flights, so this is a measure for the atmospheric variabilty on the global, large scale dynamics. $1\,\mathrm{a}$ variability gives a good impression of the annual cycle, as it includes data from many flights and different years. Figure 6 shows $\sigma_r^{\mathrm{MOD}}/\sigma_r^{\mathrm{MEAS}}$ for time scales of $30\,\mathrm{d}$ (left) and $1\,\mathrm{a}$ (right), using the datasets $\mathrm{MOD_{CARIBIC}^{sampled}}$ and $\mathrm{MEAS_{CARIBIC}^{smoothed}}$

[Figure]

**Figure 6.** $\sigma_r^{MOD}/\sigma_r^{MEAS}$ given in percent for time scales of $30\,d$ (left) and $1\,a$ (right), where MOD stands for $MOD_{CARIBIC}^{sampled}$ and MEAS stands for $MEAS_{CARIBIC}^{
[revised manuscript text omitted]
_{CARIBIC}$ (CARIBIC measurements) and $MOD_{CARIBIC}$ in each month. $\sigma_r^{MOD_{CARIBIC}}$ and $\sigma_r^{MEAS_{CARIBIC}}$ were calculated in each month. Figure ?? shows the correlation of $\sigma_r^{MOD_{CARIBIC}}$ and $\sigma_r^{MEAS_{CARIBIC}}$. Monthly variability $\sigma_r$ of $MOD_{CARIBIC}$ over $MEAS_{CARIBIC}$. Colorcoding corresponds to the variability $\tau^*$ of each species. Data closer to the tropopause is plotted as larger circles.~~

$_{CARIBIC}^{regular}$.

**2 Calculating representativeness from random numbers**

All three methods to investigate representativeness (Kolmogorov-Smirnov test, variability analysis and relative differences) have also been applied to data created with a random number generator. The results of this study are  presented here.

To produce the random numbers, 20 sets of $10^8$ numbers were taken from a normal distribution. These 20 sets are referred to as species, well aware of the fact that they are purely artificial. From species to species, the standard deviation $\sigma$ was set to vary from $10^{-3}$ to $10^3$, values of the exponent  increasing linearly. 20 mean values $\mu$ (increasing from $10^4$ to $10^8$, with a linear increase in the exponent) where distributed randomly  to the 20 species. This results in 20 species with different values for $\sigma$ and $\mu$. The statistics of each species will be indexed by the number 2. For short, this dataset will be called RAND.

3000 samples were taken from each of the 20 species. For each sample, 20 numbers were first randomly drawn from each species. These new numbers and all those that had been drawn before then make up this one sample. So the size increases by 20 for each sample. This way, the relationship of the representativeness score with the sample size is directly accessible.  Samples are indexed by the number 1.

The variability $\tau^*$ of each species  is defined as in Equation  3 of the main text: $\tau^* = \log_{10}(\sigma_2/\mu_2)$. The two requirements set up in Section 3.3 for representativeness in general also have to hold here:

1. Representativeness has to increase with the number of samples.

2. Representativeness has to decrease with increasing variability of the underlying distribution.

With RAND defined in this way, it is possible to test representativeness using the variability analysis following Rohrer and Berresheim (2006) and Kunz et al. (2008) (see Section 4.2) and the relative differences (see Section 4.3). The Kolmogorov-Smirnov test was positive for very few samples (less than fifty numbers, independent of $\tau^*$) and will not be further discussed. Its behaviour with aircraft data was subject of a sensitivity study, the results of which are shown in Sec. 3 of this supplement.

**2.1 Variability analysis**

The  variability analysis (defined in Section 4.2 and Eq. 3) was applied in a simplified manner. As RAND is independent of time, $R_{var}$ is reduced to just a single value containing the absolute difference of variability of each species of RAND and the sample taken thereof: $R_{var} = |\nu_1 - \nu_2|$, where $\nu$ is the mean variability. Figure 1 shows a result. The exact result is a matter of chance, as a random number generator is used. Similar to using $MOD_{CARIBIC}$  regular and $MOD_{RANDPATH}$, a strong dependance on $\tau^*$ and a weak dependance on the number of samples is visible.

Similar to $R_{var}$ when using $MOD_{CARIBIC}$  regular and $MOD_{RANDPATH}$, the variability analysis using RAND meets the two requirements necessary for describing representativeness, which were described in Section 3.3 and above. This result supports the  finding that $R_{var}$ can be used as a statistic for describing representativeness.

[Figure]

**Figure 1.** Representativeness score $R_{var}$ applied to RAND. Vertical lines indicate the values of $\tau^*$ of each species.

**2.2 Relative differences**

Similar to $R_{var}$, $R_{rel}$ is reduced to a simple relative difference when using RAND: $R_{rel} = |\mu_1 - \mu_2|/\mu_2$, where $\mu$ is the mean of the sample (index 1) and of the whole subset (index 2). Figure 2 shows  a result when applying $R_{rel}$ to RAND. The dependance on $\tau^*$ is strong and linear. The result also depends on the number of samples, showing a slow increase with the

5  number of samples. This dependance is sometimes disturbed by better values which are reached by chance when drawing from RAND.

Like for $MOD_{CARIBIC}$  regular and $MOD_{RANDPATH}$, $R_{rel}$ passes both conditions for a valid description of representativeness: it depends on variability $\tau^*$ and on the number of samples. The latter is also being influenced by chance and generally much weaker.

10  The fact that $R_{rel}$ passes the two conditions for a description of representativeness can be understood with some theoretical considerations. The standard error of the mean is defined by

$$\sigma_{\overline{x}} = \frac{\sigma}{\sqrt{n}} \tag{1}$$

where $\sigma_{\overline{x}}$, the standard deviation of a sample, can be given by the following equation ($N$ being the number of samples):

$$\sigma_{\overline{x}} = \sqrt{\frac{1}{N}\sum_{i=1}^{N}(\overline{x}_i - \mu)^2} \tag{2}$$

15  For $N = 1$, this gives:

$$\sigma_{\overline{x}} = |\overline{x}_i - \mu| \tag{3}$$

[Figure]

**Figure 2.** Like Figure 1, but for $R_{rel}$.

Plugging Eq. 3 into Eq. 1 gives:

$$\frac{|\overline{x}_i - \mu|}{\mu} = \frac{\sigma}{\mu\sqrt{n}} = \frac{10^{1/\tau^*}}{\sqrt{n}} \tag{4}$$

and therefore

$$R_{rel} = \log_{10}\left(\frac{|\overline{x} - \mu|}{\mu}\right) = -0.5\log_{10}(n) + \frac{1}{\tau^*} \tag{5}$$

5 So ideally, $R_{rel}$ should depend inversely on $\tau^*$ and directly on the logarithm of the number of values. Figure 2 shows this is approximately true for RAND.

In the case of RAND, $R_{rel}$  and $R_{var}$ can both be used to describe representativeness as  they pass the two conditions. Theoretical considerations make the finding plausible for $R_{rel}$. RAND can be considered a theoretical abstraction of MOD. The finding here therefore strongly supports that of Sections 5.2 and 5.3, where $R_{rel}$ and $R_{var}$

10 have also been found to be good descriptors of representativeness when using $\text{MOD}_{\text{CARIBIC}}^{\text{regular}}$ and $\text{MOD}_{\text{RANDPATH}}$ or $\text{MOD}_{\text{RANDLOC}}$. In the main text, we use $R_{rel}$ for final results, as it more suitable to answer the question of representativeness for a climatology.

**3 Sensitivity study on the Kolmogorov-Smirnov test**

When  applying the Kolmogorov-Smirnov test to $\text{MOD}_{\text{CARIBIC}}^{\text{regular}}$, $\text{MOD}_{\text{RANDPATH}}$ or $\text{MOD}_{\text{RANDLOC}}$,

15  it returned almost only negative results. This indicates that $\text{MOD}_{\text{CARIBIC}}^{\text{regular}}$ is not representative of $\text{MOD}_{\text{RANDPATH}}$ in the definition of the Kolmogorov-Smirnov test. This behaviour was tested in a sensitivity study, the results of which are  discussed here.

[Figure]

MOD$_{CARIBIC}$, flights to/from Vancouver

**Figure 3.** Flightroutes to Vancoucer, Canada, where each flight has been cut into 20 pieces and randomly chosen 30% of those pieces have been plotted. These are tested against the whole data from flights to Vancouver to give one point in Figure 4.

One of the most frequent destinations within the CARIBIC project is Vancouver, Canada (near $120\,°\mathrm{W}$, $45\,°\mathrm{N}$, see Figure 3), and only the subset of MOD$_{CARIBIC}$ $_{CARIBIC}^{regular}$ to this destination is considered in this example to minimize effects  that may come from different flight routes. Parts of this reduced dataset were tested with the Kolmogorov-Smirnov test against the whole reduced dataset for all variables.

5  To produce these partial datasets, each flight was cut into an increasing number of pieces (corresponding to a certain time) and different percentages of these pieces were used in testing. Figure 3  shows an example of applying this method, by cutting each flight into 20 pieces and taking 30% of these by showing the corresponding flightpaths.

Data was not binned in months. When applying the Kolmogorov-Smirnov test without binning in months, the result is a

[revised manuscript text omitted]